# Transient hepatic reconstitution of trophic factors enhances aged immunity

Mirco J. Friedrich[1,2,3,4,5], Julie Pham[1,2,3,4,5], Jiakun Tian[2,6], Hongyu Chen[2,6], Jiahao Huang[2,6], Niklas Kehl[2], Sophia Liu[2,7], Blake Lash[1,2,3,4,5], Fei Chen[2,8,9], Xiao Wang[2,6,10], Rhiannon K. Macrae[1,2,3,4,5] & Feng Zhang[1,2,3,4,5 ✉]

Ageing erodes human immunity, in part by reshaping the T cell repertoire, leading to increased vulnerability to infection, malignancy and vaccine failure[1–3]. Attempts to rejuvenate immune function have yielded only modest results and are limited by toxicity or lack of clinical feasibility[1,3–5]. Here we show that the liver can be transiently repurposed to restore age-diminished immune cues and improve T cell function in aged mice. These immune cues were found by performing multi-omic mapping across central and peripheral niches in young and aged animals, leading to the identification of Notch and Fms-like tyrosine kinase 3 ligand (FLT3L) pathways, together with interleukin-7 (IL-7) signalling, as declining with age. Delivery of mRNAs encoding Delta-like ligand 1 (DLL1), FLT3L and IL-7 to hepatocytes expanded common lymphoid progenitors, boosted de novo thymopoiesis without affecting haematopoietic stem cell (HSC) composition, and replenished T cells while enhancing dendritic cell abundance and function. Treatment with these mRNAs improved peptide vaccine responses and restored antitumour immunity in aged mice by increasing tumour-specific CD8⁺ infiltration and clonal diversity and synergizing with immune checkpoint blockade. These effects were reversible after dosing ceased and did not breach self-tolerance, in contrast to the inflammatory and autoimmune liabilities of recombinant cytokine treatments[6,7]. These findings underscore the promise of mRNA-based strategies for systemic immune modulation and highlight the potential of interventions aimed at preserving immune resilience in ageing populations.

Ageing has a profound effect on the immune system, including the T cell repertoire, leading to reduced immune resilience[1–3]. Central to this decline in humans and most other mammals is the involution of the thymus. Thymic involution curtails naive T cell output, contracts T cell receptor (TCR) repertoire diversity and blunts primary responses, whereas peripheral T cells accrue dysfunctional states that heighten susceptibility to infection, vaccine failure and cancer.

Efforts to counter immune ageing have primarily focused on reversing thymic involution through hormones[8], cytokines[9], small molecules[10] and heterochronic parabiosis[11], or by directly modulating haematopoiesis[12]. Although these strategies have provided valuable insights into immune ageing, they have been limited by effect size, toxicity or clinical feasibility[4,5].

Here we describe an approach for reconstituting thymus-derived factors in the liver to address age-related immune decline (Fig. 1a). We first identified signalling pathways in the thymus and peripheral blood T cells that decline with age. We then delivered mRNAs encoding these factors (DLL1, FLT3-L and IL-7) to the liver using lipid nanoparticles (LNPs). We found that this approach significantly improved immune response in ageing mice in both vaccination and cancer immunotherapy

models with no adverse side effects or evidence of increased auto-immunity. These results highlight the potential of this approach to improve immune function and, more broadly, to use the liver as a transient 'factory' for replenishing factors that decline with age.

## T cell support signals decline with age

A functional T cell repertoire depends on pro-survival and trophic cues: cytokines, hormones and self-peptide MHC, which are abundant in healthy individuals but diminish with age, contributing to immune decline[13–15].

To nominate potentially immune restorative cues, we profiled trophic signalling across the mouse lifespan using single-cell RNA sequencing (scRNA-seq) or TCR-seq of peripheral blood T cells and spatially resolved Slide-TCR-seq on the thymus, the principal organ of T cell production[16]. After quality control, the dataset comprised approximately 97,000 circulating T cell transcriptomes (CD4⁺ and CD8⁺) and 47 spatial arrays covering approximately 1.26 million thymic positions spanning the day of birth to 90 weeks of age (Extended Data Fig. 1a–h). Ageing shifted T cell states, with loss of naive (*Tcf7*, *Sell* and *Ccr7*) and

[1]Howard Hughes Medical Institute, Cambridge, MA, USA. [2]Broad Institute of MIT and Harvard, Cambridge, MA, USA. [3]McGovern Institute for Brain Research at MIT, Cambridge, MA, USA. [4]Department of Brain and Cognitive Science, Massachusetts Institute of Technology, Cambridge, MA, USA. [5]Department of Biological Engineering, Massachusetts Institute of Technology, Cambridge, MA, USA. [6]Department of Chemistry, Massachusetts Institute of Technology, Cambridge, MA, USA. [7]Ragon Institute of MGH, MIT and Harvard, Cambridge, MA, USA. [8]Harvard Stem Cell Institute, Cambridge, MA, USA. [9]Department of Stem Cell and Regenerative Biology, Harvard University, Cambridge, MA, USA. [10]Stanley Center for Psychiatric Research, Broad Institute of MIT and Harvard, Cambridge, MA, USA. ✉e-mail: zhang@broadinstitute.org

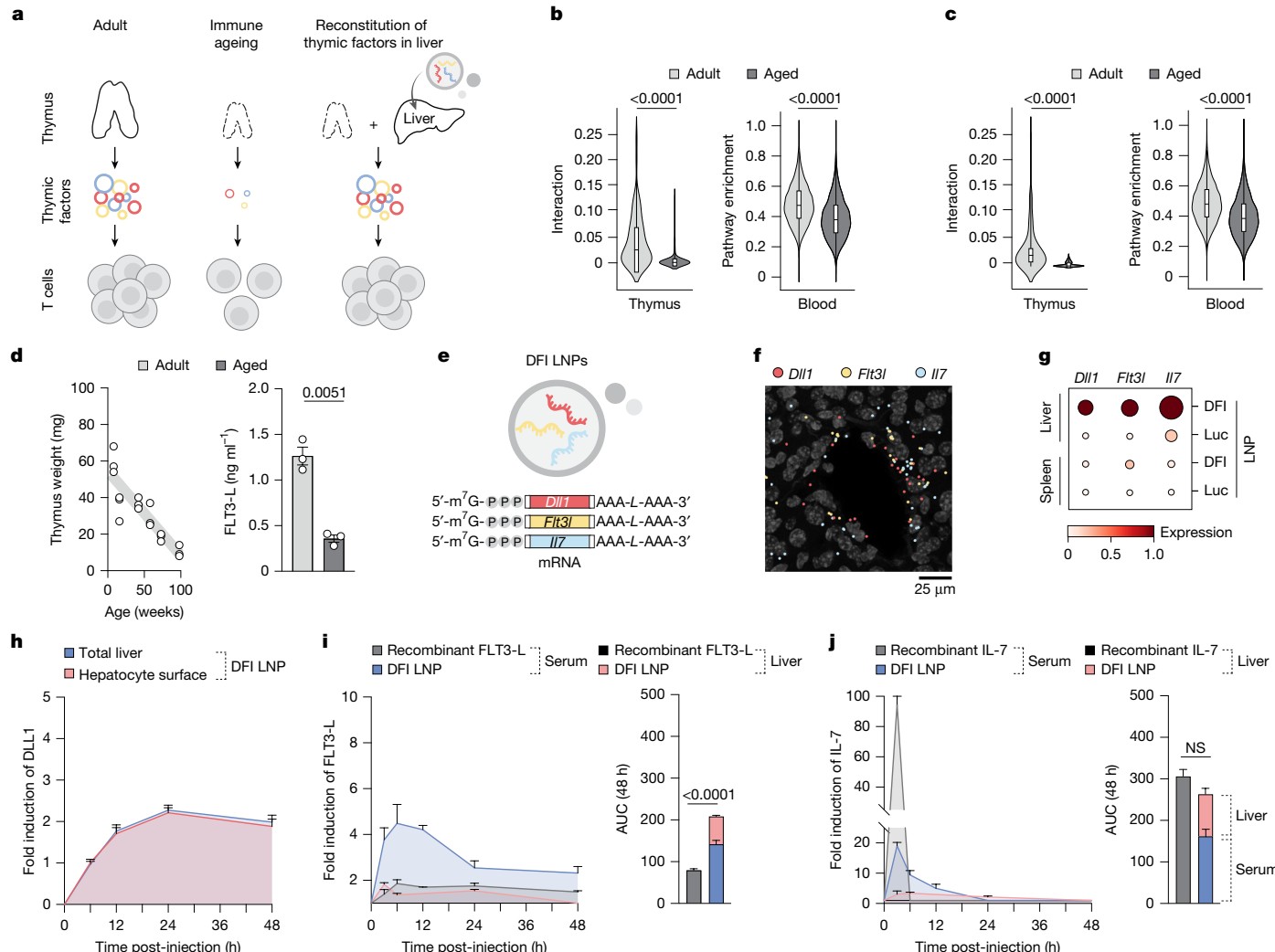

**Fig. 1 | Hepatic reconstitution of declining T cell signalling factors to restore immune signalling in ageing. a**, Overview of the approach to restore age-declining immune trophic cues by hepatic expression of *Dll1*, *Flt3l* and *Il7* mRNAs. **b**, Spatial ligand–receptor interactions between thymic cortical epithelial cells (cTECs) and thymocytes decline with age (left), and ssGSEA shows reduced Notch pathway activity in circulating T cells (right). *n* = 47 spatial arrays and 96,683 blood T cell transcriptomes across 21 ages. Data are represented as violin plots with median + interquartile range. Statistical significance was determined by Mann–Whitney tests. **c**, cTEC–T cell IL-7 interaction (left) and downstream pathway in circulating T cell (right) activities are likewise diminished with age. *n* = 47 spatial arrays and 96,683 blood T cell transcriptomes across 21 ages. Data are represented as violin plots with median + interquartile range. Statistical significance was determined by Mann–Whitney tests. **d**, Thymus weight decreases with age (*n* = 18; 3 per timepoint). Interstitial FLT3-L levels are reduced in aged thymus by ELISA (*n* = 3 per group). Data are mean ± s.e.m.; statistical significance was determined by a two-tailed unpaired Student's *t*-test. **e**, mRNA (DFI; *Dll1*, *Flt3l* and *Il7*) constructs formulated in

SM-102 LNPs. **f**, Representative RIBOmap images 6 h post-DFI show robust ribosome-bound transcripts in the liver. A representative image from three imaged DFI-treated animals is shown. **g**, Single-cell quantification: translating *Dll1*, *Flt3l* and *Il7* in the liver and spleen by RIBOmap (*n* = 1 for Luc and *n* = 3 for DFI). **h**, Immunofluorescence of DLL1 protein over 0–48 h after 5 μg DFI reveals transient induction in total liver and hepatocyte surface (phalloidin co-stain). Fold induction from baseline (0 h) is shown. *n* = 32 fields of view from *n* = 3 animals per time point per condition. Data are mean ± s.e.m. **i**, ELISA for FLT3-L levels in serum and the liver after 10 μg recombinant FLT3-L or 5 μg DFI at 3–48 h. Liver concentrations were normalized to liver weight; fold change from 0 h is shown. *n* = 3 animals per time point per compartment per condition. Data are mean ± s.e.m.; area under the cover (AUC) over 48 h compared by a two-tailed unpaired Student's *t*-test. **j**, ELISA for IL-7 in serum and the liver after 10 μg recombinant IL-7 or 5 μg DFI at 3–48 h with liver normalization and fold change from 0 h. *n* = 3 animals per time point per compartment per condition. Data are mean ± s.e.m.; 48-h AUC compared by a two-tailed unpaired Student's *t*-test. NS, not significant.

stem-like populations (*Bcl11b*, *Lef1*, *Id3* and *Sox4*), and expansion of (virtual) memory (*Cd44*, *Eomes*, *Gzmb* and *Tbx21*) and exhaustion-like phenotypes (*Pdcd1*, *Havcr2*, *Ctla4*, *Lag3*, *Tigit*, *Gzmk* and *Entpd1*), consistent with previous mouse and human studies[17,18] (Extended Data Figs. 1e,h,i and 2).

Using spatial proximity to infer bona fide cell–cell interactions (rather than co-regulation alone), we used permutation-based null models to identify age-dependent receptor–ligand pairs between thymocytes and thymic epithelial cells (TECs). Cortical TEC–T cell signalling declined markedly with age, whereas medullary TEC–T cell

interactions were relatively preserved (Extended Data Fig. 3a,b). Integrating spatially informed interaction analysis with single-sample Gene Set Enrichment Analysis (ssGSEA) of peripheral blood-circulating T cells revealed age-linked attenuation of Notch1/3 and IL-7 signalling (Fig. 1b,c and Extended Data Fig. 3c,d). Although thymic stromal cells primarily act locally to support T cell maturation and selection, several of their soluble products accumulate in the thymus, enter the circulation and contribute to systemic immune homeostasis[13,19]. Consistently, intra-thymic as well as circulating T cells from aged mice showed reduced expression of Notch, IL-7 and FLT3-L target genes and diminished

downstream activity (Fig. 1b,c and Extended Data Fig. 3e,f). Soluble FLT3-L, produced in part by intrathymic fibroblasts and required to sustain T cell function in aged and post-transplant settings, also declined in thymic homogenates[20,21] (Fig. 1d and Extended Data Fig. 3g).

These findings motivated us to transiently reconstitute age-diminished immune signalling using three immune trophic factors — DLL1 (to activate Notch), FLT3-L and IL-7 — in aged hosts. We selected DLL1 over DLL4 based on a more favourable safety and immunological profile: DLL4 is linked to angiogenesis and vascular remodelling and can overly restrict lymphoid differentiation, whereas we hypothesized that DLL1 supports T cell development without suppressing B cell output[22,23].

## Reconstituting immune cues in the liver

To bypass the structural and functional constraints of the involuted thymus, we reconstituted the identified signalling pathways ectopically in the liver. The unique haemodynamics and anatomical features of the liver enable priming and maintenance of adaptive immunity, and its protein-synthesis capacity is preserved even at advanced ages, making it a suitable site to modulate circulating T cells[24–26].

We chose mRNA delivery over recombinant proteins because recombinant cytokines clear rapidly, necessitating frequent high-dose administrations that often result in significant toxicity[6,7]. By contrast, mRNA allows for more controlled, transient protein production, and recent advances have further optimized mRNA stability and functionality while minimizing immune-related side effects for in vivo applications[6,27–29]. In addition, canonical Notch ligands are transmembrane proteins and require cell–cell contact, precluding soluble delivery. We therefore packaged mRNAs encoding DLL1 (*Dll1*), FLT3-L (*Flt3l*) and IL-7 (*Il7*) (collectively, DFI), or firefly luciferase (Luc) or GFP controls, in an SM-102 LNP formulation (Fig. 1e, Extended Data Fig. 4a–e and Supplementary Fig. 1a–e). All mRNAs were m$^1\Psi$ modified and 5′-m$^7$GpppNm capped. Primary hepatocytes expressed DLL1 on the surface and secreted biologically active IL-7 and FLT3-L into the supernatant after transfection (Extended Data Fig. 4a–c). Systemic LNP administration predominantly targeted the liver with minimal translation in other tissues (Fig. 1f,g, Extended Data Fig. 4d,e and Supplementary Fig. 1f,g). In situ profiling of ribosome-bound (rather than endosome-trapped) transcripts (RIBOmap) confirmed robust translation of all three DFI mRNAs in hepatocytes in vivo (Fig. 1f and Extended Data Fig. 4d,e), indicating hepatic expression and release with negligible off-target translation[30].

We next assessed pharmacokinetics and safety. *Dll1* mRNA induced sustained DLL1 on the surface of hepatocytes for approximately 48 h (ref. 31) (Fig. 1h and Extended Data Fig. 4f). mRNA-encoded FLT3-L achieved comparable serum half-life but higher single-dose systemic levels than recombinant FLT3-L (Fig. 1i). Intravenous recombinant IL-7 produced a sharp serum spike, whereas mRNA-encoded IL-7 yielded lower-amplitude, sustained levels with an approximately tenfold lower peak over 24 h (Fig. 1j). Because secreted IL-7 binds to heparan-sulfate proteoglycans, a substantial fraction of mRNA-delivered IL-7 remained in the hepatic extracellular matrix up to 24 h after serum levels normalized (Fig. 1j), potentially providing a compartmentalized source of IL-7 for cells traversing the sinusoids, where they also encounter hepatocyte-bound DLL1 (refs. 32,33).

Despite prolonged bioavailability of the three factors, we observed no changes in body weight, transaminases or liver function, and only minimal hepatic inflammation on histopathology after 4 weeks of DFI mRNA–LNPs in aged mice (Extended Data Fig. 4g–l), consistent with previous studies of SM-102 formulations[6,34]. By contrast, recombinant FLT3-L and IL-7 administered on the same schedule (two doses per week for 28 days) induced marked elevations of GM-CSF, IL-10, IL-12 and IL-1β, underscoring the greater risk of systemic inflammation with recombinant cytokines (Supplementary Tables 2 and 3 and Supplementary Fig. 1h,i).

## DFI mitigates immune ageing phenotypes

Ageing is accompanied by a loss of naive T cells and accumulation of (virtual) memory and exhaustion-like states, most pronounced in CD8$^+$ T cells and driven by thymic involution, chronic antigen exposure and inflammation[1,17,18]. We observed these shifts in our longitudinally profiled cohorts (Fig. 2a and Extended Data Figs. 1i and 2b,c). This bias towards the memory phenotype in ageing compromises responses to new antigens. We therefore asked whether DFI rebalances naive T cell representation.

Aged mice (72 weeks old) received DFI mRNA–LNPs or Luc mRNA–LNPs twice weekly for 28 days. DFI (but not individual factors) increased both frequency and absolute number of circulating naive (CD44$^-$CD62L$^+$) CD4$^+$ and CD8$^+$ T cells (Fig. 2b and Extended Data Fig. 5). Repeated administration of recombinant IL-7 has been shown to promote the proliferation of mature T cell subsets[35–37]. In line with this, *Il7* mRNA alone expanded effector-memory (CD44$^+$CD62L$^-$) cells; however, full DFI did not increase memory subsets (Fig. 2c and Extended Data Fig. 5c–e). Consequently, the naive-to-memory ratio rose in the blood and spleen of DFI-treated animals, indicating complementary or moderating activity of the three signals when delivered in combination (Fig. 2d and Extended Data Fig. 5b).

To delineate the origin of these DFI-induced shifts in the T cell compartment of aged mice, we next analysed the spleen, peripheral blood, bone marrow and thymus of adult and aged mice. Deep bulk V(D)J sequencing showed no increase in clonality following DFI conditioning (Fig. 2e), arguing against homeostatic proliferation and favouring enhanced thymic output as a driver of the observed increase in naive T cell counts. Although aged thymuses displayed reduced mass and cellularity relative to young controls, 28 days of DFI partially restored both (Extended Data Fig. 6a–c). Although overall distributions of double-negative (DN), double-positive or single-positive thymocytes were preserved, DFI selectively expanded early DN1–DN3 thymocytes — the stages of early T lineage commitment — and induced *Rag2* in thymocytes of *Rag2*–eGFP mice within 12 h, followed by increased mature single-positive CD4$^+$ and CD8$^+$ thymocytes towards youthful levels after 28 days of treatment (Fig. 2f and Extended Data Figs. 6d–f and 7a–c). Consistent with these data, TCR excision circles in peripheral blood were elevated (Fig. 2g), and *Nur77* expression was elevated in circulating T cells (Extended Data Fig. 7d–f), indicative of an increase in recent thymic emigrants.

Because thymic output depends on progenitor supply, we examined haematopoiesis. With ageing, HSCs expand in number but increasingly adopt a myeloid-biased differentiation program[12,38,39]; we recapitulated HSC expansion and CD150$^{high}$ myeloid-biased HSC enrichment with age, which DFI did not reverse (Extended Data Fig. 8a–d). Multipotent progenitors were largely unchanged aside from an increase in lymphoid-primed multipotent progenitors after DFI (Extended Data Fig. 8b). By contrast, DFI robustly expanded common lymphoid progenitors (CLPs), which are markedly depleted with age, in bone marrow and increased the number of circulating CCR9$^+$ (but not CCR7$^+$) CLPs, which are associated with preferential thymus homing[38,40] (Extended Data Figs. 8e–i and 9a–c). CCR9 surface levels on CLPs rose from the bone marrow to the thymus in DFI-treated mice, mirroring the dynamics found in young animals and absent in aged controls (Extended Data Fig. 9c–e), consistent with a peripheral priming of circulating CLPs that potentially augments thymic entry. Together, these data support a model in which liver-expressed DLL1, together with systemic FLT3-L and IL-7, boosts CLP production, survival and recruitment to the thymus, thereby increasing thymopoiesis. Thus, DFI enhances naive T cell output by amplifying committed lymphoid progenitors and facilitating intrathymic maturation, without reprogramming HSC composition.

Beyond T cells, DFI mitigated ageing-associated phenotypes in antigen-presenting and B cell compartments. DFI preferentially expanded splenic conventional type 1 dendritic cells (cDC1s), which

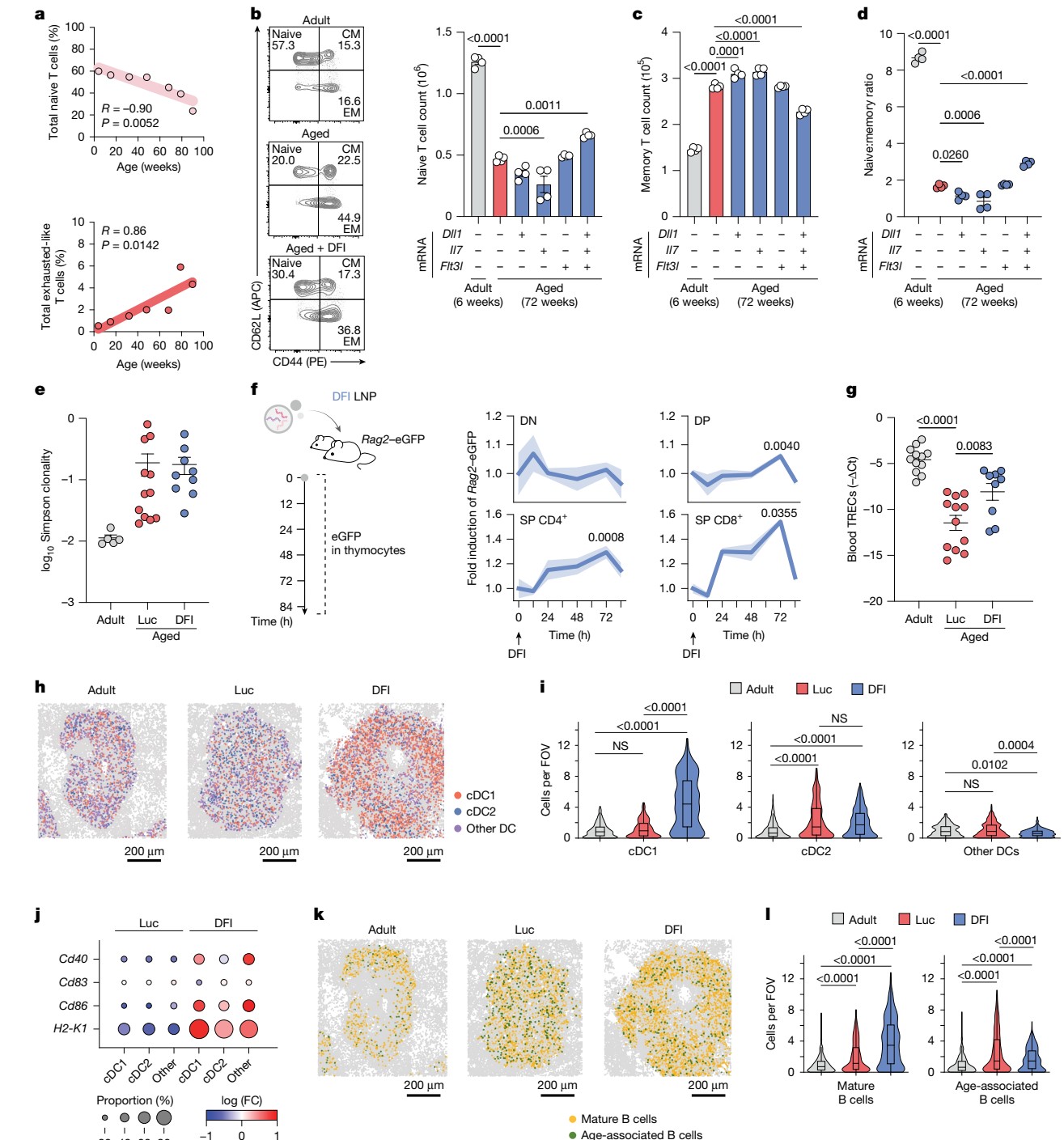

**Fig. 2 | DFI treatment mitigates ageing-associated immune features. a**, Age-related decline in circulating naive T cells (*Tcf7*, *Sell* and *Ccr7*) and accumulation of exhausted-like T cells (*Pdcd1*, *Ctla4*, *Lag3*, *Tigit*, *Entpd1* and *Tnfrsf9*) in mouse peripheral blood across age. Linear regression (red lines) shows Pearson's *R* and significance versus age. **b**–**d**, Flow cytometry of spleens from adult (6 weeks) and aged (72 weeks) mice treated twice weekly with DFI or control (Luc) mRNA–LNPs for 28 days. Absolute numbers of naive T cells (CD44⁻CD62L⁺; **b**) and memory T cells (CD44⁺CD62L⁻/⁺; **c**), and naive-to-memory T cell ratio (**d**) are shown. *n* = 4 per group. Data are mean ± s.e.m.; one-way ANOVA with Dunnett's post-hoc test. CM, central memory; EM, effector memory. **e**, TCR repertoire diversity from bulk V(D)J sequencing (*n* = 5 adult, 12 aged + Luc and 9 aged + DFI). Data are mean ± s.e.m. **f**, *Rag2*–eGFP tracing of thymocyte maturation (DN, double-positive (DP), single-positive (SP) CD4⁺ and SP CD8⁺) demonstrating increased thymopoiesis after DFI treatment. *n* = 3 mice per time point. Negative control refers to *Rag2*–eGFP⁻/⁻. Data are mean ± s.e.m. (error bands); repeated-measures two-way ANOVA with Dunnett's post-hoc test was used.

**g**, Quantification of TCR excision circles (TRECs) in peripheral blood indicating recent thymic emigrants (*n* = 12 adult, 12 aged + Luc and 9 aged + DFI); data are mean ± s.e.m.; one-way ANOVA with Tukey post-hoc test. **h**, Representative spatial STARmap projections of splenic dendritic cell (DC) subtypes from adult, Luc-treated and DFI-treated mice. **i**, Quantification of cDC1, cDC2 and other DC subsets per field of view (FOV; adult = 404, Luc = 386 and DFI = 408 FOVs from one animal per condition); data are represented as violin plots with median + interquartile range; one-way ANOVA with Tukey post-hoc test. **j**, Differential expression of activation markers (*Cd40*, *Cd83*, *Cd86* and *H2-K1*) in the indicated DC subsets. FC, fold change. **k**, Spatial STARmap projections of splenic B cell subtypes. **l**, Relative abundance of mature (*Cd19⁺*, *Ms4a1⁺*, *Cd22⁺* and *Cd40⁺*) and age-associated (*Cd19⁺*, *Ms4a1⁺*, *Itgax⁺* and *Tbx21⁺*) B cells per FOV (adult = 404, Luc = 386 and DFI = 408 FOVs from one animal per condition); data are represented as violin plots with median + interquartile range; one-way ANOVA with Tukey post-hoc test.

are critical[20] for antigen cross-presentation and decline in number and co-stimulatory function with age, probably by sustained FLT3-L-dependent differentiation from FLT3[+] progenitors[20,41,42] (Fig. 2h,i and Supplementary Fig. 2a–c). In situ-seq of spleens by STARmap revealed cDC1 enrichment within periarteriolar lymphoid sheaths and a concomitant upregulation of *H2-K1* and co-stimulatory molecules (CD40, CD83 and CD86), consistent with improved priming capacity[43] (Fig. 2j and Supplementary Fig. 2a–c). In addition, we observed a reduced abundance of splenic age-associated B cells (*CD19*[+], *Ms4a1*[+], *Itgax*[+] and *Tbx21*[+]) and an expansion of mature follicular-like B cells (*CD19*[+], *Ms4a1*[+], *Cd22*[+] and *CD40*[+]) localized to B cell follicles following DFI treatment[44,45] (Fig. 2k,l and Supplementary Fig. 2a–e).

## DFI strengthens vaccine responses

On the basis of the above findings, we next tested whether DFI could restore adaptive immune function in aged animals. As a test case, we looked at vaccine response, a T cell-mediated process known to be blunted by ageing. In an adjuvanted ovalbumin (OVA) prime–boost model in adult (6 weeks) and aged (72 weeks) mice, aged cohorts showed fewer OVA-specific CD8[+] T cells in the spleen and blood upon vaccination, impaired antigen-driven proliferation and reduced IL-2 and IFNγ production on recall, approaching levels of sham-vaccinated mice (Supplementary Fig. 3a–d). These deficiencies align with the impaired T cell-mediated vaccine responses commonly observed in older animals and humans[1,46]. Pre-conditioning aged mice with individual factors or the full DFI combination before vaccination showed that *Il7* mRNA alone increased total T cell counts, whereas only DFI increased both total T cells and the frequency of OVA-specific CD8[+] cells, yielding approximately twofold more antigen-specific CD8[+] T cells in the spleen and a similar increase in blood[35–37] (Fig. 3a–d and Supplementary Fig. 3e,f).

In addition to the reduced generation of new immune cells, ageing is characterized by the progressive shift to dysfunctional states by existing cells[2,17]. Aged mice were shown to accumulate PD1[high]CD62L[−] T cells, which we confirmed using scRNA-seq and flow cytometry in our longitudinal cohorts[12,47] (Fig. 2a and Extended Data Fig. 1e,i). This phenotype expanded further with repeated vaccination, unlike in adults (Fig. 3f). DFI conditioning preserved a higher naive fraction post-vaccination and reduced PD1[high]CD62L[−] cells (with a reduced effect from IL-7 alone), together producing a more balanced T cell composition (Fig. 3e–g and Supplementary Fig. 3g). Functionally, CD8[+] T cells from DFI-treated mice generated higher levels of IL-2 and IFNγ upon antigen-specific restimulation, whereas OVA-specific CD4[+] frequencies and cytokines were largely unchanged (Fig. 3h and Supplementary Fig. 4a–e), suggesting that DFI preferentially supports CD8[+] T cell responses.

In a longitudinal vaccination study across the lifespan of C57BL6/J mice, we found that DFI treatment increased vaccine-induced T cell counts in aged mice to levels comparable with those seen in much younger mice, effectively rejuvenating their response by approximately 24 weeks (Fig. 3i). Together, the full DFI combination, but not its single components, counteracts age-related defects in CD8[+] vaccine responses, consistent with the broader principle that aged immunity retains the capacity to mount robust immune responses when sufficiently stimulated[48].

## DFI rejuvenates antitumour responses

We next tested whether DFI improved responsiveness to tumour immunotherapy, which also diminishes with ageing. In B16-OVA melanoma and MC38-OVA colon carcinoma, we found that aged mice showed faster tumour progression and poorer survival than adults; immune checkpoint inhibition (ICI) with PDL1 blockade that controlled tumours in adults conferred little benefit in aged cohorts (Supplementary Fig. 5a–h), mirroring reports of age-dependent efficacy of PD1 or CTLA4 ICI[48–50].

Pre-conditioning aged hosts with DFI for 28 days followed by a 72-h washout to control for direct antitumour effects improved endogenous control of MC38-OVA, increasing spontaneous rejection rates and prolonging survival (Supplementary Fig. 5e–h). In the more aggressive B16-OVA model, DFI drove complete rejection in 40% of aged mice, whereas all controls succumbed within 3 weeks despite anti-PDL1 treatment (Fig. 4a–d). Co-administration at treatment onset similarly delayed progression and improved survival over PDL1 blockade alone in aged animals with established tumours (Fig. 4e–h).

In young mice, ICI confers antitumour immunity primarily through expansion and activation of tumour-specific CD8[+] T cells. Ageing impairs this process, and elderly mice and humans show diminished CD8[+] T cell infiltration and intratumoural effector function. To probe the underlying mechanism of the observed antitumour effects, we analysed tumour-infiltrating lymphocytes (TILs) on day 12 post-implantation, before significant tumour size divergence (Fig. 4i,j). Total TIL numbers were unchanged (Fig. 4k), but DFI increased the frequency of intratumoural CD8[+] T cells and consequently lowered the CD4:CD8 ratio (Fig. 4l); SIINFEKL-loaded tetramers confirmed intratumoural enrichment of antigen-specific CD8[+] cells (Fig. 4m and Supplementary Fig. 5i). In parallel, naive CD8[+] cells expanded systemically (Supplementary Fig. 5j,k), suggesting that rejuvenated peripheral pools contribute to the TIL compartment.

scRNA–V(D)J profiling and CITE-seq using oligo-conjugated SIINFEKL–H-2K[b] tetramers resolved eight canonical CD8[+]–CD4[+] TIL states and delineated tumour antigen-specific T cells (Fig. 4n,o and Supplementary Fig. 5l–o). With DFI, both tumour-specific and bystander clonotypes showed lower expression of exhaustion-associated genes (*Havcr2*, *Gzmk*, *Lag3*, *Pdcd1* and *Tigit*) and moderately higher *Entpd1* (CD39) expression, consistent with increased tumour-antigen engagement or tissue residency rather than terminal dysfunction[51] (Fig. 4p). The naive-like fraction among tetramer[+]CD8[+] TILs rose approximately 1.8-fold (Fig. 4q and Supplementary Fig. 5o), and clonal diversity (as assayed by Shannon index) increased in both tetramer[+] and bystander repertoires (Fig. 4r and Supplementary Fig. 5p), indicating broader recruitment and intratumoural TCR repertoire breadth.

Together, DFI conditioning alone boosted endogenous antitumour control and synergized with PDL1 blockade to enhance therapeutic efficacy in aged hosts. Alongside recent work with mRNA-encoded cytokines[6], these data highlight liver-encoded delivery of immune modulators as a strategy to mitigate age-related immune dysfunction and overcome components of immunotherapy resistance.

## Immunological safety of DFI treatment

To define durability and safety of DFI, aged mice received DFI or control mRNA–LNPs for 28 days and were then observed off-treatment for 28 days (Supplementary Fig. 6a). Thymic output rose during dosing, as evidenced by elevated TCR excision circles, and returned to baseline thereafter (Supplementary Fig. 6b,c), consistent with the short half-life of mRNA and the rapid decline of thymocyte *Rag2* expression after a single dose (72 h or less; Fig. 2f and Extended Data Fig. 7c). STARmap of whole spleens showed that, despite waning thymic export, splenic T cell and dendritic cell numbers remained modestly elevated at day 28 post-cessation (Supplementary Fig. 6d–g), indicating partial persistence of peripheral remodelling. However, when vaccination occurred 4 weeks after stopping DFI, previously observed benefits such as higher vaccine-specific T cells, fewer exhaustion-like cells and favourable naive–effector-memory balance were no longer evident (Supplementary Fig. 6h–k). Thus, immune enhancement by our approach is largely confined to the dosing window and reverses upon withdrawal.

We next assessed whether DFI would exacerbate autoimmune disease. In NOD mice, which are prone to type 1 diabetes due to frequent generation of autoreactive TCRs, 4 weeks of DFI or control mRNA was followed by 6-month monitoring of glycaemia, glycosuria and

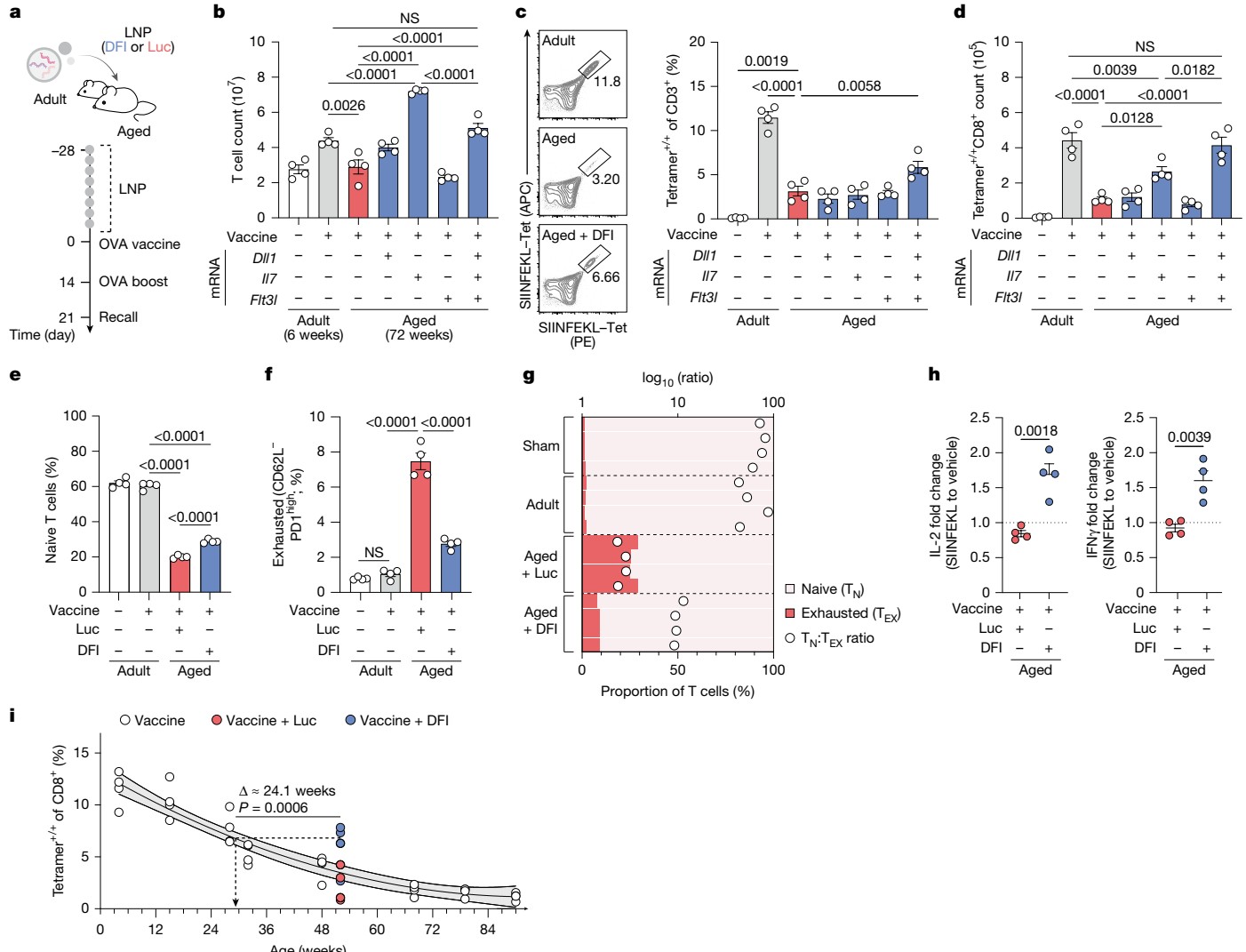

**Fig. 3 | Hepatic DFI reconstitution enhances vaccine-induced T cell responses in aged hosts. a**, Experimental design. Aged mice received DFI or control (Luc) mRNA–LNPs twice weekly for 28 days, then were immunized with adjuvanted OVA. **b**, Total splenic T cells (live CD45+CD3+) per spleen (*n* = 4 per group). Data are mean ± s.e.m.; one-way ANOVA with Tukey post-hoc test. **c**, Frequency of SIINFEKL–H-2K(b) tetramer+CD8+ T cells in the spleen (*n* = 4 per group). Data are mean ± s.e.m.; one-way ANOVA with Tukey post-hoc test. **d**, Absolute number of SIINFEKL–H-2K(b) tetramer+CD8+ T cells per spleen (*n* = 4 per group). Data are mean ± s.e.m.; one-way ANOVA with Tukey post-hoc test. **e**, Frequency of naive T cells (CD44−CD62L+) in the spleen (*n* = 4 per group). Data are mean ± s.e.m.; one-way ANOVA with Tukey post-hoc test. **f**, Frequency of exhausted-phenotype T cells (CD62L−PD1high) in the spleen (*n* = 4 per group). Data are mean ± s.e.m.; one-way ANOVA with Tukey's post-hoc test. **g**, Proportions of naive ($T_N$) and exhausted ($T_{EX}$) T cells and the $T_N$:$T_{EX}$ ratio after repetitive OVA or sham (vehicle) vaccination (*n* = 4 per treatment). **h**, Functional antigen recall: fold change in intracellular cytokines following 6-h SIINFEKL restimulation versus DMSO vehicle. IL-2 (left) and IFNγ (right) mean fluorescence intensity (*n* = 4 per group). Dotted lines indicate no change (fold change = 1.0); Data are mean ± s.e.m.; two-tailed unpaired *t*-tests. **i**, Age–response modelling. In a longitudinal cohort covering increasing ages, SIINFEKL–H-2K(b) tetramer frequencies were fit with a quadratic (second-order) polynomial regression with 95% confidence bands (d.f. = 29, $R^2$ = 0.9116). This standard curve was used to estimate an 'immunological vaccination response age' for numerically 52-week-old mice preconditioned with either DFI or Luc mRNA–LNPs before immunization (*N* = 40 total; *n* = 4 per age and treatment). The *P* value comparing the estimated vaccination response ages of the two treatment groups was calculated using a two-tailed unpaired *t*-test.

disease onset (Fig. 5a). In addition, we quantified frequencies of CD8+ T cells specific to the NRP-V7 mimotope, a known target in NOD mice[52] (Extended Data Fig. 10a). DFI did not alter blood glucose levels, diabetes onset or autoreactive T cell frequencies versus control littermates (Fig. 5b,c), whereas NOD.Cg-Tg(TcraTcrbNY8.3)1Pesa/DvsJ (NY8.3) NOD mice with a genetically encoded NRP-V7-restricted TCR uniformly developed diabetes within 8 weeks, providing a positive-control threshold. We did not observe any treatment-related adverse effects over the course of this 6-month study.

To test safety in a model with intact central tolerance, we used Act-mOVA mice, which constitutively express germline-encoded OVA[53] (Fig. 5d). This allowed us to use the same potent model antigen as in

our previous vaccination experiments; however, in this case, OVA was subject to central tolerance. Despite enhancing responses to exogenous OVA in aged wild-type mice, DFI neither induced OVA-specific CD4+ or CD8+ T cells after 4 weeks (Fig. 5e,f) nor broke humoral tolerance after adjuvanted OVA challenge in Act-mOVA mice, in contrast to wild-type controls (Fig. 5g).

Finally, in experimental autoimmune encephalitis, a polyclonal, antigen-driven central nervous system autoimmunity model, DFI was administered every 3 days following experimental autoimmune encephalitis induction by $MOG_{35-55}$ peptide immunization and thereafter until symptom onset. Aged mice showed delayed disease onset, consistent with impaired T cell priming and previous

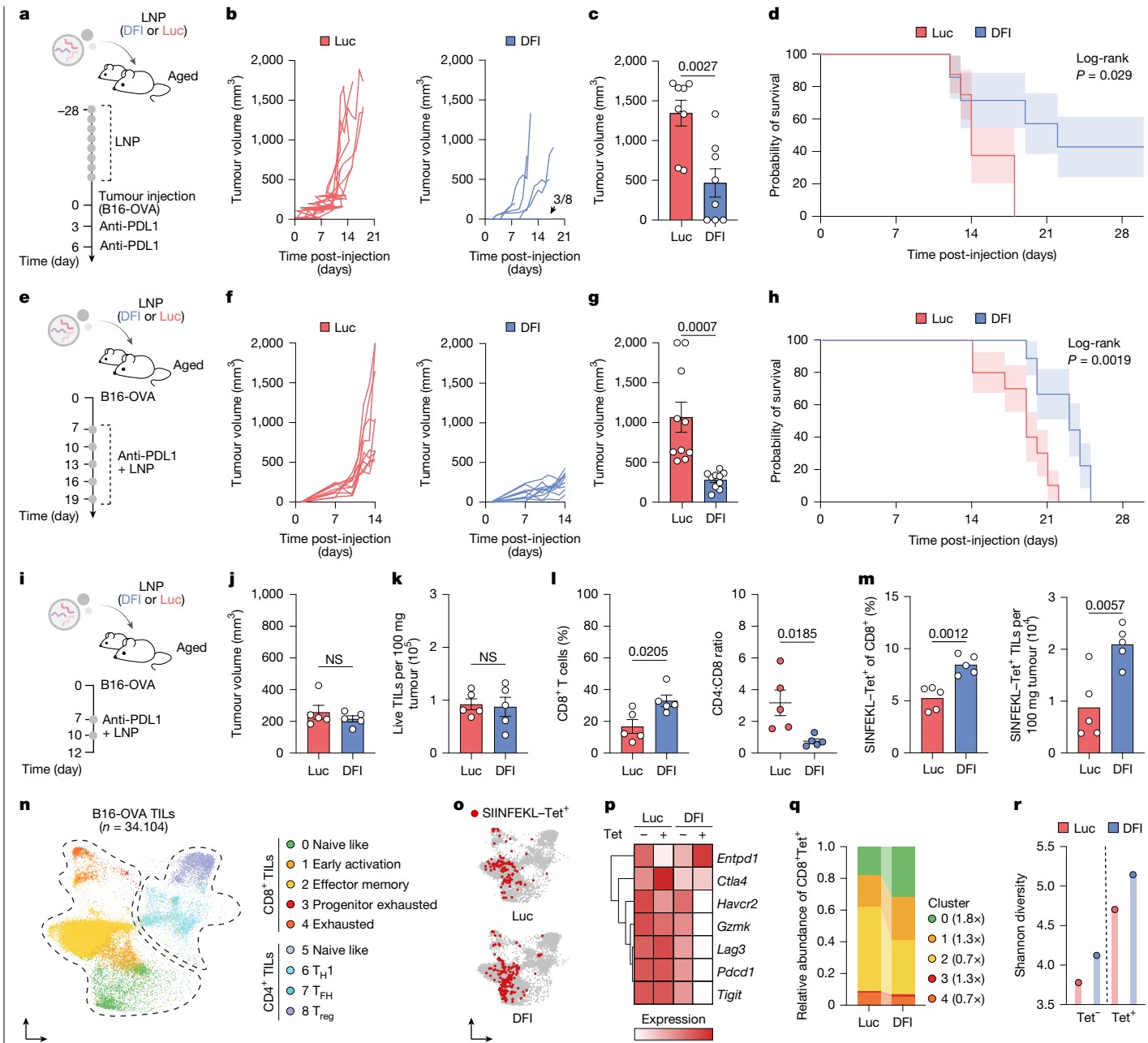

**Fig. 4 | Hepatic DFI reconstitution enhances antitumour T cell responses in aged hosts. a**, Experimental design. Aged mice received DFI or control (Luc) mRNA–LNPs for 28 days, followed by subcutaneous B16-OVA tumour challenge and two doses of anti-PDL1 checkpoint blockade. **b**, Tumour growth over time in the B16-OVA model (*n* = 8 per group). Three out of eight animals did not establish a measurable tumour (black arrow). **c**, Tumour size on day 12 (mean ± s.e.m.; two-tailed unpaired *t*-test). **d**, Kaplan–Meier survival analysis for the B16-OVA cohort (*n* = 8 per group; log-rank test). **e**, Design of adjuvant DFI or Luc LNP therapy combined with anti-PDL1 checkpoint blockade in established B16-OVA tumours. **f**, Tumour growth trajectories during combination therapy (*n* = 10 per group). **g**, Tumour size on day 14 (mean ± s.e.m.; two-tailed unpaired *t*-test). **h**, Kaplan–Meier survival curves for combination treatment (*n* = 10 per group; log-rank test). **i**, Schematic of a parallel adjuvant DFI + anti-PDL1 experiment terminated at day 12 for immune profiling with matched

tumour sizes. **j**, Tumour size at day 12 (*n* = 5 per group). **k**, Absolute counts of live TILs in explanted tumours (*n* = 5 per group). **l**, CD8[+] T cell frequency (left) and CD4:CD8 ratio (right) among TILs (*n* = 5 per group). **m**, Relative (left) and absolute (right) numbers of tumour-specific SIINFEKL–H-2K(b)[+] TILs (*n* = 5 per group). For **j**–**m**, data are mean ± s.e.m.; statistical significance was determined by a two-tailed unpaired Student's *t*-test. **n**, Uniform manifold approximation and projection (UMAP) of scRNA-seq of CD45[+]CD3[+] TILs (*n* = 34,104 cells). $T_{FH}$, follicular helper T; $T_H$, helper T; $T_{reg}$, regulatory T. **o**, CITE-seq feature plot showing CD8[+]SIINFEKL–H-2K(b)[+] (Tet[+]) TILs. **p**, Expression of canonical markers of tumour recognition and exhaustion in Luc-treated versus DFI-treated TILs. **q**, Relative abundance of transcriptional clusters among CD8[+]Tet[+] clonotypes. All comparisons were by two-tailed unpaired *t*-tests (**o**–**q**). **r**, Shannon diversity of bystander (Tet[−]) and tumour-specific (Tet[+]) TCR clonotypes across treatments.

reports, but worse subsequent clinical deterioration[54,55] (Extended Data Fig. 10b–d). In line with our earlier vaccination experiments, DFI increased peripheral, but not central nervous system-infiltrating MHC class III-A[b]–MOG-specific CD4[+] T cells after induction (Extended

Data Fig. 10e,f) and did not worsen clinical scores or spinal cord inflammation and demyelination (Extended Data Fig. 10d,g,h). In adults, DFI neither increased MOG-reactive T cells in the periphery nor aggravated disease (Extended Data Fig. 10c,e–h), indicating

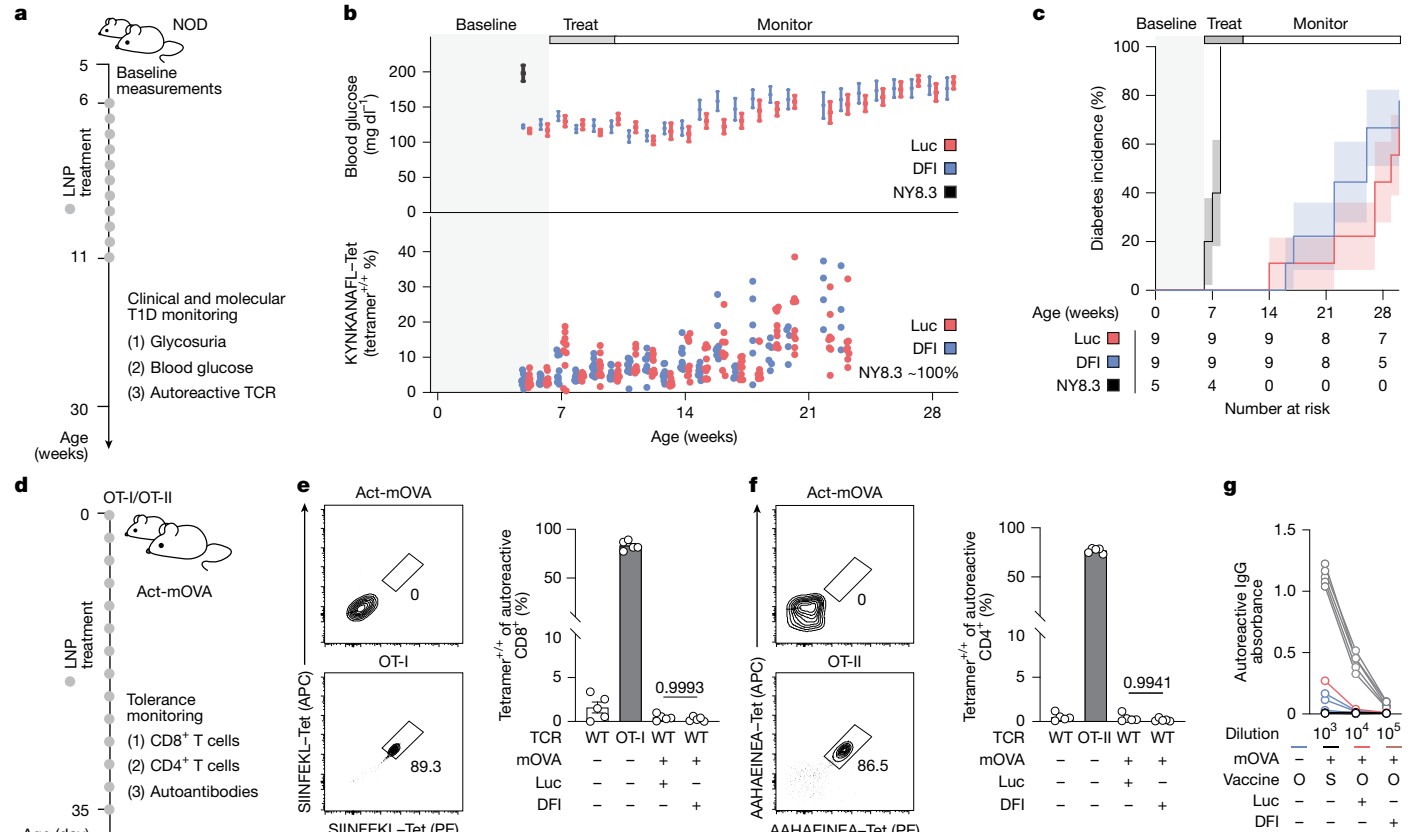

**Fig. 5 | Immunological safety assessment of DFI. a**, Experimental design to evaluate autoimmune risk in NOD mice receiving DFI or control (Luc) mRNA–LNPs. NY8.3 mice, carrying an autoreactive TCR specific for NRP-V7 (KYNKANAFL), served as positive controls. Type 1 diabetes (T1D) onset was defined by blood glucose > 200 mg dl$^{-1}$ in two consecutive measurements or by glucosuria. **b**, Longitudinal monitoring of blood glucose (top) and frequency of NRP-V7-specific TCRs in peripheral blood (bottom) of NOD mice treated as in panel **a**. NY8.3 ($n = 5$), NOD + Luc ($n = 9$) and NOD + DFI ($n = 9$). Data are mean ± s.e.m. **c**, Kaplan–Meier analysis of cumulative T1D incidence in NY8.3 ($n = 5$), NOD + Luc ($n = 9$) and NOD + DFI ($n = 9$) groups. Statistical significance was tested using log-rank (Mantel–Cox) tests. **d**, Experimental set-up to test antigen-specific T cell tolerance in Act-mOVA mice. OT-I (CD8$^+$, OVA$_{257-264}$) and OT-II (CD4$^+$, OVA$_{329-337}$) transgenic mice served as positive controls for

OVA-targeted responses. **e**, Frequencies of SIINFEKL–H-2K(b) tetramer$^+$CD8$^+$ T cells in peripheral blood of Act-mOVA mice treated with Luc or DFI mRNA for 28 days. $n = 5$ per group. Data are mean ± s.e.m.; one-way ANOVA with Tukey's post-hoc test; $P$ values are shown for Act-mOVA comparisons. WT, wild type. **f**, Frequencies of AAHAEINEA–I-A(b) tetramer$^+$CD4$^+$ T cells in the peripheral blood of Act-mOVA mice treated as in panel **e**. $n = 5$ per group. Data are mean ± s.e.m.; one-way ANOVA with Tukey's post-hoc test; two-tailed $P$ values are indicated for Act-mOVA comparisons. **g**, Serum anti-OVA IgG ELISA (OD$_{450-570}$) in Act-mOVA mice after Luc or DFI mRNA treatment followed by OVA–complete Freund's adjuvant prime and OVA–incomplete Freund's adjuvant boost (O) or sham vaccination (S). Wild-type ($n = 5$) vaccinated mice served as positive controls. Data are mean ± s.e.m.

correction of age-related deficits rather than indiscriminate immune amplification.

Together with the vaccination data, these results show that DFI transiently augments antigen-specific immunity in aged hosts while preserving self-tolerance across several models. The effects are reversible and temporally restricted, supporting liver-encoded systemic immune modulation as a novel strategy to improve immunity in older individuals.

## Discussion

Here we have showed that immune function can be improved by repurposing the liver as a platform for ectopic production of immune factors. Our results show that delivery to the liver of mRNA encoding DLL1, IL-7 and FLT3-L in combination successfully enhanced immune function in aged mice. This approach provides a scalable alternative to more invasive methods of immune rejuvenation, such as thymic transplantation[56] or HSC manipulation[12], which are clinically challenging. In addition, our disease-agnostic strategy holds potential for synergizing with other immunostimulatory treatments, as we observed in the context of ICI treatment.

Although the biological roles of IL-7 and FLT3-L are well established, with both factors known to support thymopoiesis, dendritic cell expansion and peripheral T cell homeostasis[20,21,35–37,41,42], combining them with DLL1 and delivering them to hepatocytes enables the creation of a transient, rejuvenated immune milieu in aged mice, distinct from conventional approaches based on recombinant protein infusion, which lack spatial control and tend to require chronic dosing or come with high toxicity. By using mRNA, we mitigated the systemic inflammation and autoimmune sequelae typically associated with recombinant cytokine-based therapies. However, the transient nature of mRNA delivery necessitates repeated administrations to sustain therapeutic effects. The long-term consequences of continuous exposure to these factors, especially in aged individuals should be analysed through extensive long-term safety studies.

Ageing affects the immune system beyond T cell biology, encompassing epigenetic[57] and metabolic[58] remodelling in lymphocytes, dysfunction in myeloid subsets[59] and structural alterations in stromal networks[60]. Future studies may therefore uncover additional strategies and factors to holistically target the hallmarks of immune ageing[1].

In summary, this study demonstrates that transiently repurposing the liver to express and secrete various therapeutic proteins could be

a generalizable approach to engineering physiological processes. By mimicking specific signalling niches within the liver or secreting proteins for systemic circulation to restore homeostatic signals throughout the body, this strategy has the potential to improve health outcomes and address a wide range of human diseases and conditions.

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

## Methods

### Ethical statement

All experiments were performed in compliance with all relevant ethical regulations as approved by the Institutional Biosafety Committee (IBC) of the Broad Institute (protocol #IBC-2017-00146). All animal experiments were approved by the Institutional Animal Care and Use Committee of the Broad Institute (protocol ID 0017-09-14-2). Animal maintenance complied with all relevant ethical regulations and were consistent with local, state and federal regulations as applicable, including the National Institutes of Health (NIH) Guide for the Care and Use of Laboratory Animals.

### Plasmid construction

G-blocks encoding the human α-globin 5′ untranslated region (UTR) and 3′ FI element were synthesized de novo by IDT. In vitro transcription vectors were cloned by inserting UTRs into a pET45 vector via Gibson assembly using Gibson Assembly Master Mix (E2611L, NEB) and transformation into chemically competent Stbl3 cells. A hard-coded A30LA70 polyA tail was added by PCR and ligation using the KLD enzyme mix (New England Biolabs). Subsequent coding sequences were inserted by digestion with NcoI and XhoI and Gibson assembly. Plasmid sequences were verified via next-generation sequencing, long-read sequencing (Primordium Labs) and PCR to verify the length of polyA tails.

### In vitro transcription and LiCl purification of mRNA

Plasmids were linearized, and a T7-driven in vitro transcription reaction (Life Technologies) was performed to generate mRNA with 101 nucleotide long polyA tails. The 5′ UTR and the 3′ FI elements contained sequences from the human α-globin gene. Capping of mRNA was performed in concert with transcription through addition of a trinucleotide cap1 analogue CleanCap, and m1Ψ-5′-triphosphate (TriLink) was incorporated into the reaction instead of uridine-5′-triphosphate (UTP; Supplementary Fig. 1a). LiCl-based purification of mRNA was performed, mRNAs were then checked on an agarose gel and by a TapeStation RNA ScreenTape Analysis (Agilent; Supplementary Fig. 1b,c) before aliquoting at 1 μg μl$^{-1}$ and storing at −80 °C.

### DFI LNP production

We engineered a formulation of mRNA-encoded DFI within biodegradable lipopolyplexes, along with control formulations (Fig. 1e). To do this, we formulated LNPs by combining SM-102 as ionizable lipid, 1,2-distearoyl-*sn*-glycero-3-phosphocholine (DSPC), cholesterol and 1,2-dimyristoyl-rac-glycero-3-methoxypolyethylene glycol-2000 (DMG-PEG-2000) in a molar ratio of 50:10:38.5:1.5. These were formulated into LNPs along with mRNA using microfluidic mixing using a NanoAssemblr Ignite nanoparticle formulation system (Cytiva). In brief, an ethanol phase containing the above formulated lipidoid, phospholipid, cholesterol and DMG-PEG master mix was mixed with an aqueous phase (10 mM citrate buffer, pH 3) containing mRNA at a flow rate ratio of 1:3 and at a lipidoid:RNA weight ratio of 10:1. Upon formulation, mRNA–LNPs were diluted in sterile NaCl and the buffer was exchanged by concentrating with a 30-kDa spin filter (UFC9030, MilliporeSigma) to replace residual ethanol. NaCl-diluted mRNA–LNPs were stored at 4 °C until use. For all subsequent experiments, we utilized these SM-102 mRNA–LNPs encapsulating m1Ψ-5′-triphosphate-modified and m7GpppNm-capped DFI, Luc or *GFP* mRNA, respectively.

### LNP characterization

The hydrodynamic size, polydispersity index (PDI) and zeta potential (ZP) of LNPs were measured using a DynaPro NanoStar II (Wyatt). The mRNA encapsulation efficiency of LNPs were determined using a modified Quant-iT RiboGreen RNA assay (Invitrogen) and found to be more than 85% on average (Supplementary Fig. 1d). LNP endotoxin levels were consistently found to be less than 1 endotoxin unit per ml.

The average hydrodynamic diameter was approximately 75 nm with a polydispersity index of 0.02–0.03 (Supplementary Fig. 1e).

### Cell culture and transfection

Unless otherwise stated, mammalian cells were maintained in T75 flasks (156499, Thermo Fisher) at 37 °C with 5% $CO_2$ in either DMEM-GlutaMAX (10569044, Thermo Fisher) or RPMI-GlutaMAX (61870127, Thermo Fisher). All media were supplemented with 10% FBS (97068-085, VWR) and 1× penicillin–streptomycin (15140122, Thermo Fisher). For growth of primary T cells, media were also supplemented with 50 μM 2-mercaptoethanol (21985023, Thermo Fisher). Images of transfected cells were acquired on a Leica DMI8 Confocal Microscope running Leica Application Suite X (1.4.3), equipped with a Lecia Stellaris 5 camera using an HC PL APO CS2 ×20/0.75 DRY objective and a pinhole setting of 1 Airy Unit. Images were processed using Fiji (https://imagej.net/software/fiji/downloads).

### Flow cytometry

Cells were prepared and stained according to the staining protocol of each experiment outlined below, pelleted at 500*g* for 5 min and resuspended in 200 μl of flow cytometry buffer (PBS supplemented with 2% EDTA (15575020, Life Technologies) and 5% FBS (97068-085, VWR)). Samples were run on Beckman Coulter Cytoflex LX flow cytometers and analysis was performed using the FlowJo v10 software. Representative schemes for gating and threshold setting of each experiment are shown in the Extended Data and Supplementary figures.

### Antigen-specific tetramer staining

All peptide–MHC tetramers were obtained from the NIH Tetramer Core Facility. Before viability dye and surface antibody staining, cells were incubated with appropriately titrated tetramer combinations. Unless stated otherwise, tetramer staining was performed in PBS for 20 min on ice, followed by washing and addition of antibody cocktails and/or fixation.

For OVA vaccination experiments, SIINFEKL–H-2K$^b$–PE and SIINFEKL–H-2K$^b$–APC tetramers were used at 1:100 dilution and AAHAEINEA–I-A$^b$–PE and AAHAEINEA–I-A$^b$–APC tetramers were used at 1:20 dilution. For central tolerance experiments in Act-mOVA mice, the same SIINFEKL–H-2K$^b$ tetramers (1:100 dilution) were used in combination with AAHAEINEA–I-A$^b$–PE and AAHAEINEA–I-A$^b$–APC tetramers (1:20 dilution). Non-vaccinated wild-type T cells (negative) and OT-I and OT-II T cells (positive) were included as staining controls for MHC class I and class II tetramers, respectively.

For autoimmunity experiments in NOD mice, KYNKANAFL–H-2K$^b$–PE and KYNKANAFL–H-2K$^b$–APC tetramers were used at 1:50 dilution, with NY8.3 T cells serving as positive controls for staining.

For experimental autoimmune encephalitis (EAE) experiments in C57BL/6J mice, GWYRSPFSRVVH–I-A$^b$–PE and GWYRSPFSRVVH–I-A$^b$–APC tetramers were used at 1:25 dilution. Control tetramers consisted of I-A$^b$-restricted human CLIP$_{87-101}$ (PVSKMRMATPLLMQA) conjugated to PE and APC, also at 1:25 dilution.

### Animal experiments

All animal experiments were approved by the Institutional Animal Care and Use Committee of the Broad Institute (protocol ID 0017-09-14-2). Animal maintenance complied with all relevant ethical regulations and were consistent with local, state and federal regulations as applicable, including the NIH Guide for the Care and Use of Laboratory Animals. Animals were kept on a 12-h light–dark cycle between 68 °F and 79 °F and 30–70% humidity. Mice were acclimated at the animal facility for at least 7 days before performing any experiments. The sample size for vaccination experiments was decided based on a previous publication with similar experiments[48]. The sample size for tumour experiments was decided based on previous publications with similar experiments[49,50]. The sample size for NOD experiments was decided

based on a previous publication with similar experiments[52]. The sample size for EAE experiments was decided based on a previous publication with similar experiments[61]. For all other exploratory experiments, no sample size calculations were performed. For all experiments, allocation of mice into experimental groups was randomized after stratifying for age and sex. Separate investigators performed treatment and data collection. Data-collecting investigators, for example, for tumour size measurements, were blinded to the treatment groups. Data-analysing investigators were not blinded to the treatment groups, as they involved internal controls, with the exception of pathologists for toxicity studies, who were blinded for the analyses. Experimental and control animals were treated equally and, when possible, housed in mixed cages.

### In vivo delivery of mRNA
mRNA–LNPs (5 µg of each mRNA, equal molar, normalized to the control RNA) in a total volume of 100 µl sterile NaCl were injected through slow retro-orbital injection into each mouse. For Luc imaging, sterile NaCl injection was used as negative control. For all other experiments investigating DFI mRNA–LNPs, Luc mRNA–LNPs were used as negative control.

### In vivo Luc imaging
In vivo activity of Luc following delivery of Luc mRNA–LNPs was measured using a Competent IVIS-Perkin Elmer IVIS Spectrum CT System (Perkin Elmer). Mice were injected through retro-orbital injection with Luc mRNA–LNPs at a dose of 5 µg mRNA per mouse, and bioluminescence imaging was performed on an IVIS imaging system (PerkinElmer). At 6 h post-injection, mice were anaesthetized with isoflurane and intraperitoneally injected with D-luciferin potassium salt (150 mg kg$^{-1}$ (body weight)). Mice or dissected organs were imaged 10 min post-injection using auto-exposure settings. The luminescent activity was quantified using Aura 4.0 imaging software.

### Quantification of in vivo DLL1 levels in the liver
To assess DLL1 protein induction following DFI, mice were intravenously injected with either 5 µg DFI mRNA–LNPs in 100 µl NaCl. The mice were then anaesthetized with isoflurane and rapidly decapitated. Liver tissue samples were harvested from the mediolateral lobe and placed in Tissue-Tek O.C.T. Compound. Then, the liver tissue in O.C.T. was frozen in liquid nitrogen and stored at −80 °C. Liver DLL1 levels were measured using immunofluorescence. In brief, liver tissue was cryo-sectioned at 15 µm thickness, fixed with 4% paraformaldehyde for 15 min and permeabilized with 0.5% Triton X-100 and 100 mmol l$^{-1}$ glycine diluted in PBS for 10 min. Samples were blocked with blocking buffer (10% normal donkey serum (017-000-121, Jackson ImmunoResearch) in PBS–0.1% Tween-20) and stained with anti-DLL1 antibody (ab10554, Abcam; 1:800 dilution in blocking buffer) at 4 °C overnight. Samples were then washed with PBS–0.1% Triton X-100 for 3 × 10 min, stained with AF546-labelled secondary antibody (A10040, Invitrogen; 1:500 dilution in blocking buffer) at room temperature for 1 h and washed with PBS–0.1% Triton X-100 for 3 × 10 min. Samples were then stained with DAPI and AF488-phalloidin (A12379, Invitrogen) according to the manufacturer's protocol to visualize the plasma membrane of hepatocytes. Liver DLL1 levels were quantified by taking the immunofluorescence signal intensity normalized to DAPI intensity at 6, 12, 24 and 48 h after injection ($n$ = 3 mice per condition and time point). Uninjected littermates served as baseline controls (0 h). Tissue concentrations were expressed as fold change relative to baseline.

### Quantification of in vivo IL-7 and FLT3-L levels
To assess cytokine induction following DFI or recombinant protein administration, mice were intravenously injected with either 5 µg DFI mRNA–LNPs in 100 µl NaCl or 10 µg recombinant mouse IL-7 (217-17, PeproTech) or 10 µg recombinant mouse FLT3-L (250-31L, PeproTech) in NaCl containing 0.1% BSA. Blood samples were collected by terminal

cardiac puncture using EDTA-coated syringes and transferred to BD Microtainer tubes (365974, BD). Plasma was separated by centrifugation at 2,000$g$ for 10 min at room temperature and snap frozen for later analysis.

Liver tissue samples were harvested from the mediolateral lobe, weighed for normalization and snap frozen. Frozen tissue fragments were homogenized on ice in 200 µl PBS containing protease inhibitor cocktail (P8340-1ML, Sigma) using a pre-chilled glass douncer with 15 strokes. Homogenates were clarified by centrifugation (10,000 rpm for 10 min at 4 °C), and supernatants were stored at −80 °C. IL-7 and FLT3-L levels in both the serum and liver were quantified using ELISA kits (mouse IL-7, EMIL7, Invitrogen; mouse FLT3-L, EMFLT3L, Invitrogen) according to the manufacturer's protocol. Absorbance was measured at 450 nm with 570-nm background subtraction, and cytokine concentrations were calculated using logistic regression fitted to the standard curve. Samples were processed in technical triplicates, and the mean value was used for biological replicates.

Serum and liver cytokine levels were measured at 6, 12, 24 and 48 h after injection ($n$ = 3 mice per condition and time point). Uninjected littermates served as baseline controls (0 h). Tissue concentrations were normalized to organ weight and expressed as fold change relative to baseline.

To assess age-dependent changes in thymic cytokine levels, thymus lobes were isolated from untreated C57BL/6J mice across a range of ages, including 3, 6, 12, 18 and 24 months ($n$ = 3 per age group). Thymic tissue was weighed, snap frozen and processed identically to liver samples. FLT3-L concentrations were determined by ELISA, normalized to thymus weight and used to calculate total cytokine abundance across the lifespan.

### Subcutaneous immunization
Adult (6 weeks) and aged (72 weeks) mice were immunized with 1 mg ml$^{-1}$ full-length OVA protein emulsified in complete Freund's adjuvant (EK-0301, Hooke Laboratories), followed by one booster dose of 1 mg ml$^{-1}$ protein emulsified in incomplete Freund's adjuvant (IFA; EK-0311, Hooke Laboratories). Mice were injected with antigen emulsified in complete Freund's adjuvant (CFA) subcutaneously at two sites on the chest, injecting 0.1 ml at each site (total of 0.2 per mouse). The needle was kept inserted in the subcutaneous space for 10–15 s after each injection to avoid leakage of the emulsion. A booster injection of antigen emulsified in IFA was administered 14 days after immunization with antigen–CFA emulsion. The booster was given as a single subcutaneous injection with 0.1 ml of IFA emulsion, at one site on the sternum. Serum, peripheral blood or spleen samples were obtained 21 days after the initial immunization, unless otherwise specified.

### Subcutaneous tumour implantation and treatment with ICIs
OVA-expressing melanoma B16 (B16-OVA) and MC38 (MC38-OVA) cell lines were provided by M. Kilian. Mice were inoculated subcutaneously with 1 × 10$^5$ B16-OVA or 5 × 10$^5$ MC38-OVA in Matrigel Matrix (Corning). Tumour growth was monitored daily by measuring with digital calipers using the two largest perpendicular axes until the area (0.5 × larger diameter × smaller diameter$^2$). The size of the tumours was assessed in a blinded, coded manner every day following treatment start and recorded as tumour volume. Mice were euthanized when tumours reached 2,000 mm$^3$ or upon ulceration. Of anti-PDL1 antibody (10 F.9G2, BioXCell), and/or control hamster and/or control rat IgG antibody (BioXCell), 100 µg was injected intraperitoneally every 3 days, as previously described, unless otherwise specified[49].

### Isolation and staining of blood-circulating T cells
For all vaccination experiments, at the indicated time points post-vaccination, approximately 100 µl of blood was collected with EDTA-coated capillary tubes from each mouse and then transferred to an EDTA-coated tube. The collected blood samples were centrifuged at

2,000*g* for 10 min, followed by transferring the resulting plasma into another tube, and antibody staining was performed using the eBioscience one-step Fix/Lyse Solution (10X; 00-5333-54, Thermo Fisher). In brief, 50 or 100 µl of blood samples was incubated with 50 or 100 µl twofold concentrated antibody cocktails (1:100 final dilution) as well as TruStain FCX (anti-mouse CD16/32, clone 93; BioLegend; 1:50 final dilution) for 20 min at 4 °C in the dark, followed by the addition of 4 ml of one-step Fix/Lyse Solution and 15-min incubation at room temperature in the dark. Samples were then washed twice with 10 ml of PBS followed by centrifugation at 500*g* for 5 min and finally resuspended in 200 µl flow cytometry buffer. In the case of tetramer staining, samples were pre-incubated with the respective twofold concentrated tetramers (1:20 to 1:100 final dilution) for 20 min on ice in the dark, followed by incubation with twofold concentrated antibody cocktails (1:100 final dilution) as well as TruStain FCX (anti-mouse CD16/32, clone 93; BioLegend; 1:50 final dilution) for another 20 min at 4 °C in the dark.

### Quantification of cellular vaccination responses

On day 21 following the initial subcutaneous immunization with OVA–CFA, mouse spleen single-cell suspensions were prepared in RPMI 1640 medium by mashing tissue against the surface of a 70-µm cell strainer (64752-00, BD Falcon). Then, the single-cell suspension was centrifuged at 500*g* for 5 min and the supernatant was removed. Red blood cells were lysed by adding 1 ml of ACK lysis buffer (Thermo Fisher) at 4 °C for 1.5 min, followed by centrifugation and removal of the supernatant. The cells were washed once with RPMI 1640 medium and then resuspended with RPMI 1640 medium (10% FBS and 1% penicillin-streptomycin antibiotic). Of splenocytes from each mouse, $4 \times 10^6$ were cultured in RPMI medium and stimulated with SIINFEKL peptide (synthesized at 99% purity by GenScript) at a final concentration of 1 µg ml$^{-1}$ for each peptide for CD8$^+$ T cell recall or 1× Cell Stimulation Cocktail containing phorbol 12-myristate 13-acetate and ionomycin (00-4970-03, eBioscience) for CD4$^+$ T cell recall. The GolgiStop transport inhibitor cocktail (554724, BD) was added according to the manufacturer's instruction 2 h later. Then, 6 h later, the cells were collected and washed with flow cytometry buffer (PBS with 2% FBS) before Fc block with TruStain FCX (anti-mouse CD16/32, clone 93; BioLegend; 1:50 final dilution) and surface antibody staining for 20 min at 4 °C. Cells were washed with a flow cytometry buffer and then fixed and permeabilized using a BD Cytoperm fixation/permeabilization solution kit (554714, BD) according to the manufacturer's instructions. Cells were washed in perm/wash solution, followed by intracellular staining (for 45 min at 4 °C) using a cocktail of the respective cytokine or transcription factor antibodies. Finally, the cells were washed in perm/wash solution and suspended in a staining buffer. Samples were washed, resuspended in 200 µl flow cytometry buffer and acquired on a Beckman CytoFLEX LX Flow Cytometer. Analysis was performed using FlowJo v10 software.

### TCRβ repertoire sequencing

Snap-frozen spleens were submitted to Adaptive Biotechnologies for deep TCRβ repertoire profiling. Genomic DNA was extracted using the DNeasy Blood & Tissue Kit (69504, Qiagen), followed by deep sequencing of rearranged *Tcrb* gene segments using the immunoSEQ mmTCRB Deep Sequencing platform (Adaptive Biotechnologies). Library preparation, high-throughput sequencing and initial data processing, including demultiplexing, quality filtering and V(D)J gene annotation, were performed by Adaptive Biotechnologies using their proprietary pipeline. Output files included productive clonotype frequencies and diversity metrics used for downstream analysis.

### Quantification of TRECs

To quantify thymic output, signal joint T cell receptor excision circles (sjTRECs) were measured from peripheral blood. A total of 100 µl of cardiac blood was collected from mice following terminal anaesthesia and centrifuged to pellet cellular material. Genomic DNA was extracted

using the DNeasy Blood & Tissue Kit (69504, Qiagen) according to the manufacturer's instructions.

Analysis was performed using an Bio-Rad CFX Opus Real-Time PCR system, following previously published protocols for sjTREC detection in C57BL/6 mice[62,63]. The sjTREC-specific primers used were: forward, 5′-CCAAGCTGACGGCAGGTTT-3′; reverse, 5′-AGCATGGCAAGCAGCACC-3′. To control for input DNA variability, amplification of the constant region of the *Tcra* gene was used as an endogenous reference. The *Tcra* primer sequences were: forward, 5′-TGACTCCCAAATCAATGTG-3′; reverse, 5′-GCAGGTGAAGCTTGTCTG-3′. Cycle threshold (Ct) values were determined in technical duplicates for both targets. Relative sjTREC content was calculated as $\Delta Ct = Ct\_TREC − Ct\_TCRA$. For clarity, values are reported as $−\Delta Ct$, such that higher values reflect greater relative sjTREC abundance.

### Spontaneous type 1 diabetes model and monitoring

NOD/ShiLtJ (NOD) mice were treated with DFI mRNA or Luc mRNA encapsulated in SM-102 LNPs twice per week for 4 weeks followed by clinical and molecular monitoring for the development of T1D. NY8.3 mice transgenic for the autoreactive TCR recognizing NRP-V7 were included as a positive control group. Experimental end point (onset of T1D) was reached in animals with blood glucose levels of more than 200 mg dl$^{-1}$ in two independent measurements or in animals developing glucosuria[52,64].

### Experimental autoimmune encephalitis

EAE was induced as previously described[61]. All animals were of a C57BL/6J background. Mice were immunized subcutaneously with 100 µg of MOG$_{35-55}$ peptide (110582, Genemed Synthesis) emulsified in CFA, which was freshly prepared by combining 20 ml of IFA (BD263910, BD Biosciences) with 100 mg of *Mycobacterium tuberculosis* H37Ra (231141, BD Biosciences) at a 1:1 ratio (v/v; 5 mg ml$^{-1}$ final concentration). Each mouse received two subcutaneous injections of 100 µl of the MOG–CFA emulsion.

Pertussis toxin (180, List Biological Laboratories) was administered intraperitoneally at 320 ng per mouse (1.6 ng µl$^{-1}$ in 200 µl PBS) on the day of immunization and again 48 h later. Mice were monitored twice daily and scored for EAE symptoms using the following clinical scoring system: 0 for no signs; 1 for limp tail; 2 for hindlimb weakness; 3 for hindlimb paralysis; 4 for forelimb paralysis; and 5 for moribund.

To assess the effects of DFI on disease progression, mice were randomly assigned to receive DFI or Luc control mRNA–LNPs after immunization until symptom onset. mRNA–LNPs were administered every 3 days starting on the day of immunization and continued until the onset of clinical symptoms in the first animals (day +15). To avoid repeated anaesthesia, intraperitoneal injection was used instead of retro-orbital delivery of mRNA–LNPs. All clinical scoring was performed in a blinded manner.

### Quantification of MOG-specific CNS-infiltrating T cells

To isolate central nervous system (CNS)-infiltrating lymphocytes, mice were terminally anaesthetized and perfused transcardially with ice-cold PBS. Spleens were collected and weighed, and brains and spinal cords were harvested. Spinal cords were flushed by hydrostatic pressure. CNS tissues were minced and enzymatically digested in RPMI supplemented with Liberase (5401119001, Roche) for 30 min at 37 °C with gentle agitation. Following digestion, tissues were dissociated by 20 trituration cycles using a 10-ml serological pipette and filtered through a 70-µm cell strainer.

Mononuclear cells were enriched by 30% isotonic Percoll density gradient centrifugation at 800*g* for 30 min at room temperature without brake. The cell pellet was washed in PBS and resuspended in 200 µl PBS for downstream staining. For tetramer staining, 100 µl of the cell suspension was incubated for 1 h at room temperature with PE-conjugated or APC-conjugated I-A$^b$ tetramers at a 1:25 dilution. The following tetramers were used: MOG$_{38-49}$–I-A$^b$–PE and MOG$_{38-49}$–I-A$^b$

(GWYRSPFSRVVH; PE and APC; NIH Tetramer Core Facility) and control human CLIP$_{87-101}$-I-A$^b$ (PVSKMRMATPLLMQA, PE and APC; NIH Tetramer Core Facility). After tetramer incubation, cells were washed and stained for extracellular markers using standard flow cytometry antibodies. Following final washes, cells were resuspended in 200 µl flow cytometry buffer (PBS + 2% FBS + 2 mM EDTA) and immediately analysed.

## Histopathological assessment of CNS tissue from EAE mice
Mice were perfused intracardially with ice-cold 1× PBS followed by ice-cold 4% paraformaldehyde (PFA). Spinal columns were harvested and post-fixed overnight in 4% PFA at 4 °C. For histopathological analysis of both meningeal and parenchymal compartments, spinal columns were subsequently transferred to 0.5 M EDTA (pH 7.4; S28291GAL, Thermo Fisher Scientific) and decalcified at 4 °C with continuous inversion for 7 days.

Following decalcification, tissues were processed in toto, paraffin-embedded, sectioned and mounted on glass slides. Serial sections were stained with haematoxylin and eosin and Luxol Fast Blue to assess inflammation and demyelination, respectively. All histological analyses were performed by a board-certified anatomical pathologist (IDEXX BioAnalytics).

Microscopic lesions were evaluated using established semiquantitative grading systems. Inflammatory changes were scored on a four-point scale as previously described[65,66], ranging from no detectable inflammatory cells (score 0), to scattered infiltrates (score 1), perivascular clustering (score 2), and extensive perivascular cuffing with parenchymal extension or diffuse infiltration (score 3). Demyelination was graded based on Luxol Fast Blue staining from minimal subpial demyelination (score 1), through marked subpial and perivascular involvement (score 2), confluent subpial or perivascular demyelination (score 3), to extensive demyelination affecting one-half (score 4) or the entirety (score 5) of the spinal cord section, often accompanied by parenchymal immune infiltration.

All samples were well preserved with minimal or no autolytic or decalcification artefacts. Numerical scores were used to quantify total lesion burden and to compare the severity and prevalence of pathological changes across treatment groups.

## STARmap protocol
STARmap padlock and primer probes were designed as previously described[67]. For the 64-gene STARmap data collection, 2–6 pairs of primer and padlock probes (Supplementary Table 4) were designed for each gene.

The mice used in this study were anaesthetized with isoflurane and rapidly decapitated. The mouse spleen tissue was collected and placed in Tissue-Tek O.C.T. Compound. Two biological replicates were collected for each condition (Luc, DFI and wild type). For the post-DFI-treatment analysis, three biological replicates were collected for each condition (NaCl, Luc, DFI and 4-weeks post-DFI). Then, the spleen tissue in O.C.T. was frozen in liquid nitrogen and stored at −80 °C. For mouse spleen tissue sectioning, the spleen tissue was transferred to cryostat (CM1950, Leica) and cut into 20-µm cross-ections at −20 °C. The slices were transferred and attached to glass-bottom 24-well plates pretreated with 3-(trimethoxysilyl)propyl methacrylate and poly-D-lysine.

The STARmap procedure was conducted as previously described[67]. In brief, the spleen slices were fixed with 400 µl 4% PFA in 1× PBS at room temperature for 15 min, then permeabilized with 600 µl pre-chilled methanol at −20 °C for 1 h. The samples were then taken from −20 °C freezer to room temperature for 5 min, then quenched with 400 µl quenching solution (0.1 mg ml$^{-1}$ yeast tRNA, 0.1 U µl$^{-1}$ SUPERase·In RNase inhibitor, 100 mM glycine and 0.1% Tween-20 in PBS) at room temperature for 10 min. After quenching, the samples were rinsed with 600 µl PBSTR (1× PBS supplemented with 0.1% Tween-20 and 0.02 U µl$^{-1}$ SUPERase·In RNase Inhibitor) twice. Then, the samples were incubated with 200 µl of 1× hybridization buffer (2× SSC, 10% formamide, 20 mM

ribonucleoside vanadyl complex, 0.1 mg ml$^{-1}$ yeast tRNA, 0.1 U µl$^{-1}$ SUPERase·In, 1% Triton X-100, pooled padlock and primer probes at a concentration of 10 nM per oligo) at 40 °C in a humidified oven with parafilm wrapping and shaking for 20 h. The samples were washed with 300 µl PBSTR twice and 300 µl high-salt washing buffer (4× SSC dissolved in PBSTR) once, for 20 min at 37 °C for each wash. Finally, the samples were rinsed once with 300 µl PBSTR at room temperature.

The samples were then incubated with a 200 µl ligation mixture (0.1 Weiss U µl$^{-1}$ T4 DNA ligase, 0.5 mg ml$^{-1}$ BSA and 0.2 U µl$^{-1}$ of SUPERase·In RNase inhibitor in 1× T4 DNA ligase buffer) at room temperature for 3 h with gentle shaking. After the ligation reaction, the samples were washed twice with 300 µl PBSTR and then incubated with 200 µl rolling circle amplification mixture (0.2 Weiss U µl$^{-1}$ Phi29 DNA polymerase, 250 µM dNTP, 20 µM 5-(3-aminoallyl)-dUTP, 0.5 mg ml$^{-1}$ BSA and 0.2 U µl$^{-1}$ of SUPERase·In RNase inhibitor in 1× Phi29 buffer) at 4 °C for 30 min then 30 °C for 3 h with gentle shaking. The samples were washed twice with 0.1% Tween-20 in PBS (PBST) before being temporarily stored at 4 °C overnight.

The next day, the samples were treated with 300 µl freshly prepared modification mixture (20 mM methacrylic acid NHS ester in 100 mM sodium bicarbonate buffer) at room temperature for 1 h and then washed once by PBST for 5 min. The samples were rinsed once and then incubated with 200 µl monomer buffer (4% acrylamide, 0.2% bis-acrylamide and 0.2% tetramethylethylenediamine in 2× SSC) at 4 °C for 15 min. Then, the buffer was aspirated, and 50 µl polymerization mixture (0.2% ammonium persulfate dissolved in pre-cooled monomer buffer) was added to the centre of the sample and immediately covered by Gel Slick-coated glass coverslip (72226-01, Electron Microscopy Sciences). The polymerization reaction was performed for 1 h at room temperature in an N2 box, then washed by PBST twice for 5 min each. The samples embedded in hydrogel were digested with 300 µl proteinase K mixture (0.5 mg ml$^{-1}$ proteinase K and 1% SDS in 2× SSC) at 37 °C for 3 h, then washed by PBST three times for 5 min each. Subsequently, the samples were treated with 200 µl dephosphorylation mixture (0.25 U µl$^{-1}$ Antarctic phosphatase, 0.5 mg ml$^{-1}$ BSA in 1× Antarctic phosphatase buffer) at 37 °C overnight and washed by PBST three times for 5 min each.

For SEDAL sequencing, each sequencing cycle began with treating the sample with 800 µl stripping buffer (60% formamide and 0.1% Triton X-100 in H$_2$O) at room temperature twice for 10 min each, followed by washing with 1 ml PBST three times for 5 min each. Then, the samples were incubated with a 250 µl sequencing mixture (0.1875 U µl$^{-1}$ T4 DNA ligase, 0.5 mg ml$^{-1}$ BSA, 10 µM reading probe and 5 µM decoding probes in 1× T4 DNA ligase buffer) at room temperature for at least 5 h. The samples were washed with 900 µl washing and imaging buffer (10% formamide in 2× SSC buffer) three times for 10 min each, then immersed in washing and imaging buffer for imaging. Images were acquired using Leica TCS SP8 confocal microscopy with ×40 oil immersion objective (NA 1.3) and a voxel size of 194 nm × 194 nm × 350 nm ($x × y × z$). DAPI staining was performed before the first cycle using 5× DAPI in PBST for 3 h at room temperature. The DAPI signal was collected at the first cycle of imaging. Four cycles of imaging were performed to decode 64 genes.

After SEDAL sequencing, the samples were treated with 800 µl of stripping buffer three times at room temperature for 10 min each. The samples were then washed with 1 ml PBST three times for 5 min each. To visualize the plasma membrane and aid cell segmentation, the samples were treated with 300 µl Flamingo staining mixture (1× Flamingo fluorescent gel stain and 5× DAPI in PBS) at room temperature overnight, then washed with 300 µl PBST three times for 5 min each. The Flamingo signal was collected and the DAPI signal was reimaged while the samples were immersed in PBST with 0.1× Flamingo fluorescent gel stain.

## STARmap data analysis
STARmap data analysis was performed as previously described[43,67]. In brief, image deconvolution was achieved with Huygens Essential

(v21.04; Scientific Volume Imaging (http://svi.nl)), using the CMLE algorithm, with SNR:10 and 10 iterations. Image registration, spot calling and barcode filtering were performed as previously described.

For 3D cell segmentation, a synthetic image with improved contrast between cell nuclei was generated by multiplying the inverted Flamingo staining image and DAPI staining image after enhancing contrast using Fiji for each field of view (FOV). A StarDist 3D segmentation model was then trained using a manually labelled training dataset created from the synthetic data[68]. Subsequently, the model was applied to predict segmentation for each FOV.

For tissue region identification, to isolate low-frequency (that is, large length scale) transcriptional patterns over the tissue, Laplacian smoothing over a spatial Delaunay triangulation (that is, a nearest neighbour mesh of cells) was performed. The specific low-pass filter used was a heat kernel with time $t = 10$. Principal component analysis was then performed on the resulting low-pass-filtered cell-by-gene matrix to identify region features, followed by clustering in the principal component space to produce categorical region labels. $K$-means was used for clustering to avoid smoothing-induced spatial autocorrelation artefacts[69].

For quality control and cell-type classification, cells with less than two reads and expressed fewer than two genes were excluded. Gene expression profiles were normalized and scaled using standard Scanpy procedures[70]. A hierarchical clustering approach was then utilized to create a three-level cell-type annotation (Supplementary Fig. 2a). Initially, 24 clusters were identified through $k$-means clustering of the preprocessed gene expression profile containing 19 genes, which were further categorized into four level 1 cell types (T cells, B cells, macrophages and dendritic cells; Supplementary Fig. 2b). Cells lacking expression of any of the selected 19 gene markers were removed from subsequent analysis. Each level 1 cell-type underwent additional $k$-means clustering to establish level 2 annotations (Supplementary Fig. 2c). Level 2 CD4$^+$ or CD8$^+$ T cells then underwent a level 3 $k$-means clustering using a panel of 15 markers for further refinement in classification (Supplementary Fig. 2d).

## RIBOmap procedure

Sixteen genes were profiled to validate the expression pattern of the three therapeutic mRNAs on liver and spleen tissues. The three therapeutic mRNAs, *Dll1*, *Il7* and *Flt3l*, were profiled using RIBOmap, whereas the 13 cell-type marker genes were profiled using STARmap. RIBOmap enables the quantification of translation levels of *Dll1*, *Il7* and *Flt3l*, which is a better reflection of protein production than mRNA levels. RIBOmap also excludes endosomal and extracellular mRNA signals, which are commonly found in mRNA-injected tissue samples. For both RIBOmap and STARmap, 5–6 pairs of primer and padlock probes were designed as described in Zeng et al.[30] (Supplementary Tables 1 and 4). The mice used in this experiment were anaesthetized with isoflurane and rapidly decapitated. The mouse liver and spleen tissues were collected and placed in Tissue-Tek O.C.T. Compound. Three biological replicates with therapeutic mRNA injections were collected, and one control sample with Luc mRNA injection was collected. Then, the liver and spleen tissues in O.C.T. were frozen in liquid nitrogen and stored at −80 °C.

The RIBOmap procedure was conducted similarly to the STARmap procedure described above[30]. The only difference is that RIBOmap splint probes targeting the 18S rRNA were included in the hybridization mixture. Data analysis was performed as described in the STARmap data analysis section. The translation levels of *Dll1*, *Il7* and *Flt3l* were quantified at a single-cell level.

## Serum toxicity analyses

Heart blood was aspirated by cardiac puncture using EDTA-precoated syringes and collected in BD Microtainer Tubes with potassium EDTA additive (365974, BD). Serum was isolated by centrifugation at 2,000$g$

for 10 min at room temperature and snap frozen on dry ice. All samples were received by IDEXX BioAnalytics and stored securely at −80 °C before analysis. Serum AST, ALT, CK, albumin, triglycerides and GGT were measured by an Olympus AU5400 (IDEXX BioAnalytics). Samples demonstrating haemolysis were excluded from the analysis.

## Serum cytokine analyses

Heart blood was aspirated by cardiac puncture using EDTA-precoated syringes and collected in BD Microtainer Tubes with potassium EDTA additive (365974, BD). Serum was isolated by centrifugation at 2,000$g$ for 10 min at room temperature and snap frozen on dry ice. All samples were received by IDEXX BioAnalytics and stored securely at −80 °C before analysis. Samples were tested on the Milliplex MAP mouse cytokine/chemokine magnetic bead panel (MCYTOMAG-70K-PMX, Millipore) according to the kit protocol as qualified. Data were collected by xPONENT 4.3 (Luminex) and data analysis was completed using BELYSA 1.1.0 software. The data collected by the instrument software are expressed as median fluorescence intensity (MFI). MFI values for each analyte were collected per each individual sample well. Analyte standards, quality controls and sample MFI values were adjusted for background. Calibrator data were fit to either a five-parameter logistic or four parameter logistic model depending on best fit to produce accurate standard curves for each analyte. Quality control and sample data were interpolated from the standard curves and then adjusted according to the dilution factor to provide calculated final concentrations of each analyte present in the sample. Samples demonstrating haemolysis were excluded from the analysis.

## Histopathological assessment of liver tissue

To evaluate the hepatic safety of LNP administration, liver tissue was harvested from female C57BL/6J mice. Twelve mice ($n = 4$ per group) comprising adult untreated controls, aged mice treated for 28 days with Luc–LNPs and aged mice treated for 28 days with DFI–LNPs were intracardially perfused under terminal anaesthesia with ice-cold 1× PBS, followed by 4% PFA. Livers were dissected, post-fixed in 4% PFA overnight at 4 °C and submitted to IDEXX BioAnalytics for blinded histopathological assessment.

Upon receipt, liver tissues were trimmed, paraffin embedded, sectioned and mounted onto glass slides. Sections were routinely stained with haematoxylin and eosin to assess general histology and hepatocellular integrity. Serial sections were stained with Masson's Trichrome to visualize connective tissue and assess the presence of fibrosis. Microscopic evaluation was performed by board-certified veterinary pathologists blinded to treatment groups. Histopathological changes were graded for severity using a standardized semi-quantitative scale: 0 for no significant findings, 1 for minimal, 2 for mild, 3 for moderate, and 4 for severe. Evaluation criteria adhered to the International Harmonization of Nomenclature and Diagnostic Criteria standards (https://www.toxpath.org/inhand.asp). Numerical lesion scores were used to quantify both prevalence and severity of histological changes within and across experimental groups.

## Dissociated scRNA-seq and data preprocessing

Heart blood was obtained through cardiac puncture in mice following cervical dislocation and stored in 0.5% EDTA for subsequent processing. Red blood cells were lysed using 1 ml ACK lysis buffer (Thermo). Freshly isolated immune cells post-lysis of red blood cells were blocked with rat anti-mouse CD16/32 (0.5 μg per well, eBioscience). Subsequently, respective antibodies in PBS were added in a total volume of 50 μl and stained for 30 min. eFluor 780 fixable viability dye (eBioscience) was used per the manufacturer's protocol to exclude dead cells.

For scRNA-seq, cells were divided into eight aliquots per animal sample and pre-incubated for 10 min with titrated amounts of TotalSeq hashtag antibodies (C0301-C0308, BioLegend). Cells were sorted on a BD Aria Fusion cell sorter using a 100-μM nozzle and four-way purity

mode. From peripheral blood, viable T cells (live, CD45[+] and CD3[+]) were sorted in 20 μl 0.04% BSA in PBS and kept on ice until processing.

Single-cell capture, reverse transcription and library preparation were conducted on the Chromium platform (10X Genomics) with the single-cell 5′ reagent v2 kit (10X Genomics) following the manufacturer's protocol, using 40,000 cells as input per channel. Each pool of cells underwent library quality testing, and library concentration was assessed. The final library for each pool was subjected to paired-end sequencing (26 bp and 92 bp) on one Illumina NovaSeq 6000 S2 lane. Raw sequencing data were processed and aligned to the mouse genome (GRCm39 − mm39) using the CellRanger pipeline (10X Genomics, v7.1.0).

## scRNA-seq analysis of circulating T cells

Seurat datasets were generated for blood T cells at each time point. Singlets were identified per the published Seurat vignette (https://satijalab.org/seurat/articles/hashing_vignette.html) and used for downstream analyses. In addition, only cells with more than 500 and less than 4,000 unique features were detected, and less than 5% of mitochondrial counts were considered for further analysis. Blood datasets for each age group underwent merging and integration using the harmony package, following the guidelines in the published vignette (https://portals.broadinstitute.org/harmony/SeuratV3.html). The integration process involved the application of the following arguments: NormalizeData(), FindVariableFeatures(selection.method = "vst", nfeatures = 2000),ScaleData(), RunPCA(), RunHarmony ("orig.ident", plot_convergence = TRUE, dims.use = 1:20), RunUMAP (reduction = "harmony", dims = 1:20), FindNeighbors(reduction = "harmony", dims = 1:20), FindClusters(resolution = 0.5). After removing cells failing quality control, integration was reiterated with the same settings, and final transcriptional clusters for downstream analyses were identified using the FindClusters function (resolution = 0.7).

Annotation of mouse peripheral blood datasets was carried out manually, based on the final transcriptional clustering post-quality control and integration, using canonical marker genes and differential gene expression analyses through the FindAllMarkers function in Seurat. For subsequent visualization subsets for CD8 T cells (clusters: CD8_T activated, CD8_T cytotox, CD8_T effector-like, CD8_T exhausted, CD8_T IFN-responsive, CD8_T memory-like and CD8_T naive-like; 56,967 cells) and CD4 T cells (clusters: CD4_T activated, CD4_T IFN-responsive, CD4_T memory-like/naive, CD4_T naive and Treg; 39,716 cells) were built and a new UMAP embedding based on the harmony components was calculated.

The statistical significance of changes in cell-type abundance over time in the single-cell data was assessed using Pearson correlation through the cor.test(…, method= "pearson") function. For visualization, $\log_2$ fold changes of cell-type proportions at each time point relative to the 4-week time point were computed and graphically represented. Cell types exhibiting a significant correlation with increasing age ($P < 0.05$) were colour annotated. Density UMAP visualization was performed using the ggplot2 stat_density_2d() function.

## scRNA-seq and V(D)J-seq analyses of B16-OVA TILs

B16-OVA were explanted at day +12 post-injection so that tumour sizes remained comparable between treatment groups. Tumour tissues were minced and enzymatically digested in RPMI supplemented with Liberase (5401119001, Roche) for 30 min at 37 °C with gentle agitation. Following digestion, tissues were filtered through a 70-μm cell strainer and centrifuged. The resulting cell pellet was resuspended in a 40% Percoll–PBS solution and layered on top of a 80% Percoll–PBS solution in 50 ml followed by centrifugation at 1,260*g* for 20 min at room temperature with the acceleration at the lowest setting and no break. The middle interface layer containing TILs was isolated and stained with a fixable viability dye (eFluor780) respective fluorophore-labelled antibodies, as well as an oligo-conjugated and PE-labelled H-2Kb SIINFEKL

dextramer (JD02163DXG PE 25, Immudex). Cells were sorted on a Sony MA900 cell sorter using a 100-μM nozzle. Viable T cells (live, CD45[+] and CD3[+]) were sorted in 20 μl 0.04% BSA in PBS and kept on ice until processing. Single-cell capture, reverse transcription and library preparation were conducted on the Chromium platform (10X Genomics) with the single-cell 5′ reagent v2 kit (10X Genomics) following the manufacturer's protocol, using 20,000 cells as input per channel. Each pool of cells underwent library quality testing, and library concentration was assessed. The final library for each pool was subjected to paired-end sequencing (10 bp, 10 bp and 90 bp) on one Illumina NovaSeq X 25B lane. Raw sequencing data were processed and aligned to the mouse genome (GRCm39 − mm39) using the CellRanger pipeline (10X Genomics, v7.1.0).

Single-cell transcriptomic and V(D)J data were generated from four parallel runs, each consisting of sorted TILs isolated and pooled from two animals. Raw gene expression and antibody-derived tag (ADT) matrices using the conjugated oligo sequence (CCCATATAAGAAA) of the H-2Kb SIINFEKL dextramer were processed in R (v4.3.2) using the Seurat package (v5.3.0)[71]. Cells were retained if they expressed at least 200 but no more than 7,000 genes and exhibited less than 5% mitochondrial gene content. Genes detected in fewer than three cells (CreateSeuratObject(…,min.cells = 3)) were excluded. ADT assays were normalized using centred log-ratio transformation. TCR contig annotations were added to the scRNA dataset per sample using scRepertoire (v2.0.7)[72]. Clones were called based on the CDR3 amino acid sequence throughout the study. Clonal proportion was calculated per run. For downstream analyses, only cells with annotated TCR were used. T cells were projected onto the default TIL reference atlas using ProjecTILs (v3.5.2) to enable canonical cell-type annotation[73]. For visualization, the reference atlas UMAP embedding was used. H-2K[b]-SIINFEKL dextramer staining was analysed from ADT assays. Cells with a centred log-ratio-normalized tetramer signal of more than 1.7 were classified as SIINFEKL specific (Tet[+]). Thresholding was guided by ridge plot distribution and cell-type distribution of the signal. Clonal diversity was calculated using scRepertoire function clonalDiversity(…,cloneCall = "aa"). Treemap plots of the tetramer-positive TCR repertoire were generated using the treemap package (v2.4-4). To this end, cells per treatment condition were downsampled to the same number of cells and used for visualization.

## Quantifying spatial organization via interaction count statistic

Upstream data generation and analysis have been published elsewhere[16]. For each pair of cell types at each time point, a *Z*-score was computed by comparing the observed frequency of cell interactions for the pair within a 50-μm radius to the null hypothesis of the frequency of cell interactions for the same pair within a 150-μm radius. Subsequently, these *Z*-scores were plotted over time, and Pearson's *r* was used to assess the strength of the correlation between changing *Z*-scores and age.

## Receptor–ligand analysis

Upstream data generation and analysis have been published elsewhere[16]. We used the Squidpy[74] integration of CellPhoneDB[75] and Omnipath[76] to identify shifts in receptor–ligand interactions at each time point. Subsequently, the mean values of each receptor–ligand cell–cell pair across time were used to calculate Spearman's *R* correlation and a *P* value. After correcting the *P* values through Bonferroni correction, unique interactions were categorized into those decreasing and increasing with age for visualization.

## Chemicals and enzymes

Chemicals and enzymes are listed as name (catalog number, vendor): Tissue-Tek O.C.T. Compound (4583, SAKURA); glass bottom 24-well plates (P24-1.5H-N, Cellvis); 3-(trimethoxysilyl) propyl methacrylate (M6514, Sigma-Aldrich); poly-D-lysine (A-003-M, Sigma-Aldrich);

16% PFA (15710-S, Electron Microscopy Sciences); UltraPure DNase/RNase-free distilled water (10977023, Invitrogen); methanol (34860-1L-R, Sigma-Aldrich); PBS (10010-023, Gibco); Tween-20, 10% solution (655206, Calbiochem); yeast tRNA (AM7119, Thermo Fisher Scientific); SUPERase·In RNase inhibitor (AM2696, Thermo Fisher Scientific); UltraPure SSC, 20× (15557044, Invitrogen); formamide (655206, Calbiochem); ribonucleoside vanadyl complex (S1402S, New England Biolabs); T4 DNA ligase, 5 Weiss U µl$^{-1}$ (EL0012, Thermo Fisher Scientific); Phi29 DNA polymerase (EP0094, Thermo Fisher Scientific); deoxynucleotide (dNTP) solution mix (N0447L, New England Biolabs); UltraPure BSA (AM2618, Thermo Fisher Scientific); 5-(3-aminoallyl)-dUTP (AM8439, Thermo Fisher Scientific); methacrylic acid NHS ester, 98% (730300, Sigma-Aldrich); DMSO, anhydrous (D12345, Invitrogen); acrylamide solution, 40% (161-0140, Bio-Rad); Bis solution, 2% (161-0142, Bio-Rad); ammonium persulfate (A3678, Sigma-Aldrich); N,N,N′,N′-tetramethylethylenediamine (T9281, Sigma-Aldrich); Gel Slick Solution (50640, Lonza Bioscience); OminiPur SDS, 20% (7991, Calbiochem); proteinase K solution, RNA grade (25530049, Thermo Fisher Scientific); Antarctic phosphatase (M0289L, New England Biolabs); DAPI (D1306, Molecular Probes); 10% Triton X-100 (93443, Sigma-Aldrich); and Flamingo fluorescent protein gel stain (1610490, Bio-Rad).

## Antibodies

For immunohistochemistry, immunofluorescence and STARmap analyses of DLL1: anti-DLL1 antibody 1:800 (ab10554, Abcam) and AF546-labelled secondary donkey anti-rabbit IgG antibody 1:500 (A10040, Invitrogen) were used.

For mouse T cell analyses, anti-mouse IL-2 PE 1:100 (clone JES6-5H4, lots B351622 and B377599; 503808, BioLegend), anti-mouse CD45 BV510 1:100 (clone 30-F11, lots B386738, B360620 and B384034; 103138, BioLegend), anti-mouse IFNγ APC 1:100 (clone XMG1.2; lots B354911, B370994 and B396246; 505810, BioLegend), anti-mouse/human CD44 PE 1:100 (clone IM7, lot B343363; 103024, BioLegend), anti-Mo CD8a eBioscience eFluor 450 1:100 (clone 53-6.7, lot 2527379; 48-0081-82, Invitrogen), rat anti-mouse CD4 PerCP 1:100 (clone RM4-5, lots 2279727, 3201956 and 1334056; 553052, BD Bioscience), anti-mouse CD62L APC 1:100 (clone MEL-14, lot B371017; 104412, BioLegend), anti-mouse CD3 FITC 1:100 (clone 17A2; lots B388315, B388061 and B406287; 100204, BioLegend), hamster anti-mouse TCRβ APC 1:100 (clone H57-597; lot 3030398 and 2076848; 553174, BD Bioscience), TruStain FcX anti-mouse CD16/32 1:50 (clone 93; lots B398113, B380119, B368516, B372578 and B419152; 101320, BioLegend), eBioscience Fixable Viability Dye eFluor 780 1:1,000 (lot 2752774; 65-0865-14, Invitrogen), anti-mouse CD4 PE-Cy5.5 1:500 (clone RM4-5, lot B398342; 100514, BioLegend), anti-mouse CD8a BV510 1:100 (clone 53-6.7, lot B427044; 100752, BioLegend), anti-mouse/human CD44 PE 1:200 (clone IM7, lot B343363; 103024, BioLegend), anti-mouse CD62L APC 1:100 (clone MEL-14, lot B371017; 104412, BioLegend), anti-mouse CD3 Pacific Blue 1:100 (clone 17A2, lot B427533; 100214, BioLegend) and anti-mouse CD279 (PD1) APC 1:100 (clone 29F.1A12, lot B376789; 135210, BioLegend).

For mouse thymus analyses, anti-mouse CD4 PE-Cy5.5 1:500 (clone RM4-5, lot B398342; 100514, BioLegend), anti-mouse CD8a BV510 1:100 (clone 53-6.7, lot B427044; 100752, BioLegend), anti-mouse CD25 PerCP 1:200 (clone PC61, lot B378943; 102028, BioLegend), anti-mouse/human CD44 PE 1:200 (clone IM7, lot B417002; 103007, BioLegend), rat anti-mouse CD117 APC 1:100 (clone 2B8, lot 3199305; 553356, BD Bioscience), anti-mouse CD28 FITC 1:100 (clone E18, lot B373960; 122008, BioLegend), anti-mouse CD24 Pacific Blue 1:500 (clone M1/69, lot B385383; 101820, BioLegend), anti-mouse TCRβ chain PE/cyanine7 1:400 (clone H57-597, lot B394527; 109222, BioLegend), rat anti-mouse CD45R/B220 BUV661 1:100 (clone RA3-6B2, lot 3292032; 612972, BD Bioscience), TruStain FcX anti-mouse CD16/32 1:50 (clone 93, lots B398113, B380119, B368516, B372578 and B419152; 101320,

BioLegend) and eBioscience Fixable Viability Dye eFluor 780 1:1,000 (lot 2752774; 65-0865-14, Invitrogen).

## Reporting summary

Further information on research design is available in the Nature Portfolio Reporting Summary linked to this article.

## Data availability

All single-cell profiling data of circulating T cells and B16-OVA TILs in ageing mice are available on Zenodo[77] (accession number 17385897). All spatial profiling (RIBOmap and STARmap PLUS) data of liver and spleen samples are available on Zenodo[78] (accession number 13858686). All spatial profiling (Slide-TCR-seq) data of thymuses across age are available on Zenodo[79] (accession number 17525729). The mouse genome assembly (GRCm39) was used for data alignment (NCBI RefSeq assembly; GCF_000001635.27).

## Code availability

All codes necessary to reproduce all spatial thymus RNA-seq analyses (https://github.com/immunoliugy/aging_mouse_thymus) and all RIBOmap and STARmap PLUS analyses (https://github.com/wanglab-broad/hepatic_recon) are available on GitHub.

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

**Acknowledgements** We thank all members of the Zhang laboratory and M. Kilian for their support and useful discussions; and the Broad Comparative Medicine and the Broad Flow Cytometry Core staff for their technical support and guidance. F.Z. is supported by the Howard Hughes Medical Institute; the K. Lisa Yang Brain-Body Center at MIT; the Broad Institute Programmable Therapeutics Gift Donors; the Pershing Square Foundation, William Ackman and Neri Oxman; and by the Phillips family and J. and P. Poitras. M.J.F. was funded by the Deutsche Forschungsgemeinschaft (German Research Foundation; 512475649), supported by an EMBO Postdoctoral Fellowship (ALTF 689-2023) and acknowledges the support from

the Deutsche José Carreras Leukämie-Stiftung (DJCLS 01ZI/2022), the Dr. Rolf M. Schwiete Stiftung (Project ID 2025-018) and the Else Kröner-Fresenius-Stiftung (2025_EKMS.52). X.W. acknowledges support from the Merkin Institute Fellowship, the Packard Fellowship, the Sloan Research Fellowship and the NIH DP2 New Innovator Award. S.L. acknowledges support from Impetus Grants, the Mark & Lisa Schwartz AI Initiative, the Margaret Q. Landenberger Research Foundation and BroadIgnite.

**Author contributions** M.J.F. and F.Z. conceived the project. M.J.F. designed all the experiments with input from F.Z., R.K.M. and other authors. M.J.F. and J.P. performed and analysed all the experiments relating to identification and in vivo reconstitution of immune trophic factors. J.T., H.C. and J.H. performed the STARmap and RIBOmap experiments and analyses. M.J.F., J.P. and S.L. performed the longitudinal scRNA–TCR-seq and Slide-TCR-seq experiments. N.K. and S.L. performed the longitudinal scRNA–TCR-seq and Slide-TCR-seq analyses. B.L. provided assistance and guidance in cloning and production of mRNA–LNPs. F.C. and X.W. provided critical mentorship and guidance in technical procedures. F.Z. supervised this research and experimental design with support from R.K.M. M.J.F. and R.K.M. wrote the manuscript with input from all authors.

**Competing interests** M.J.F. and F.Z. are co-inventors on US provisional patent application no. 63/530,465 filed by the Broad Institute and MIT relating to this work. F.Z. is a scientific advisor and cofounder of Beam Therapeutics, Pairwise Plants, Arbor Biotechnologies, Aera Therapeutics and Moonwalk Biosciences; and is a scientific advisor for Octant. M.J.F. reports speaker honoraria and consulting fees from Pfizer, Roche, Kerna Ventures and Moonwalk Biosciences. X.W. is a scientific cofounder and consultant of Stellaromics and Convergence Bio. All other authors declare no competing interests.

## Additional information

**Correspondence and requests for materials** should be addressed to Feng Zhang.

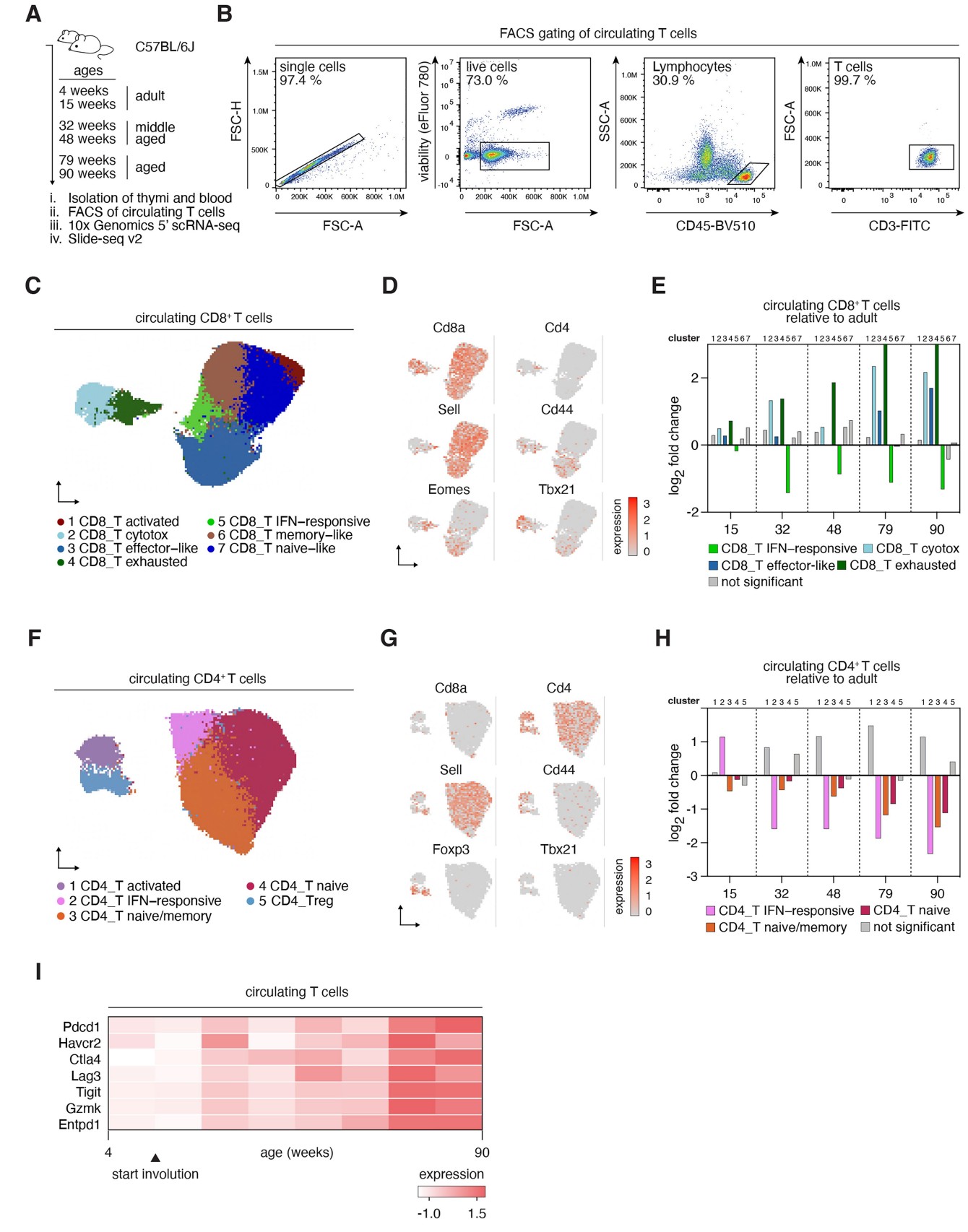

**Extended Data Fig. 1** | See next page for caption.

**Extended Data Fig. 1 | Single-cell analysis reveals age-related shifts in peripheral and thymic T cell composition.** (A) Experimental overview for longitudinal profiling of peripheral and thymic T cells. Circulating T cells were isolated from peripheral blood of adult (6 weeks), middle-aged (36 weeks), and aged (72 weeks) mice and subjected to 10x Genomics 5′ single-cell RNA-sequencing. Thymuses were explanted for spatial transcriptomic profiling using Slide-seq v2. (B) Representative flow cytometry gating strategy for sorting peripheral blood CD3+ T cells for single-cell RNA-sequencing. (C) UMAP of CD8+ T cells (CD45+ CD3+ CD8+) across all age groups with cell-type annotations. Total of n = 56,967 CD8+ T cells. (D) Expression feature plots of canonical gene markers overlaid on UMAP from (C). (E) Proportional changes in CD8+ T cell subsets with age. Data shown as fold change relative to 4-week-old mice. Statistically significant changes highlighted in color (methods). (F) UMAP of CD4+ T cells (CD45+ CD3+ CD4+) across all age groups with cell-type annotations. Total of n = 39,716 CD4+ T cells. (G) Expression feature plots of canonical gene markers overlaid on UMAP from (F). (H) Proportional changes in CD4+ T cell subsets with age. Data shown as fold change relative to 4-week-old mice. Statistically significant changes highlighted in color (methods). (I) Heatmap showing expression of exhaustion-associated genes across circulating T cells stratified by age. Arrowhead denotes the average time point of overt thymic involution in C57BL/6 J mice.

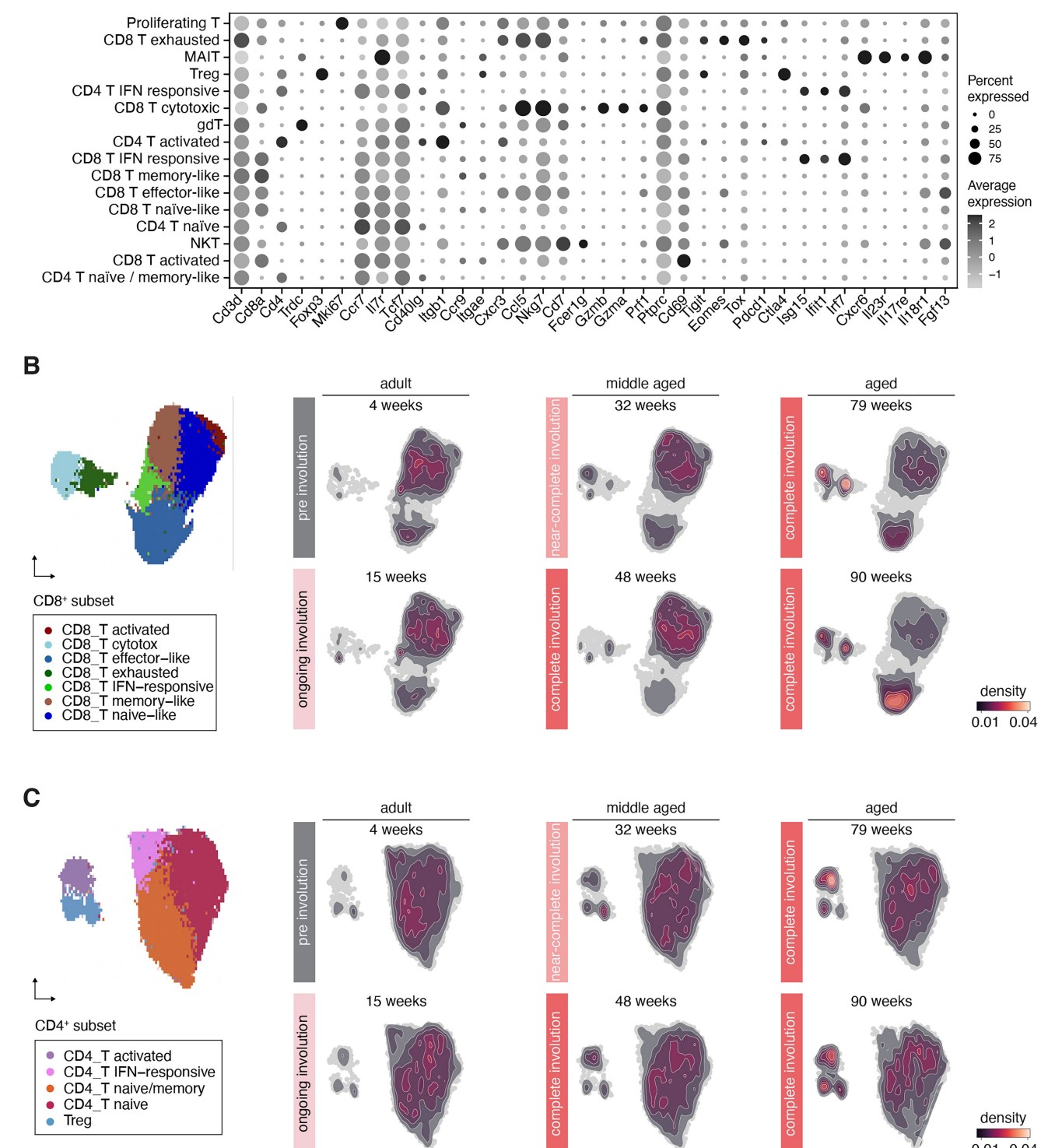

**Extended Data Fig. 2 | Age-associated transcriptional reprogramming in peripheral CD4⁺ and CD8⁺ T cells.** **(A)** Dot plot of gene expression across annotated T cell subsets identified in single-cell RNA-seq data from peripheral CD45⁺ CD3⁺ T cells. **(B)** Left: UMAP of CD8⁺ T cells across all age groups annotated by subset. Right: same UMAP colored by mouse age. Total of n = 56,967 CD8⁺ T cells. **(C)** Left: UMAP of CD4⁺ T cells across all age groups annotated by subset. Right: same UMAP colored by mouse age. Total of n = 39,716 CD4⁺ T cells.

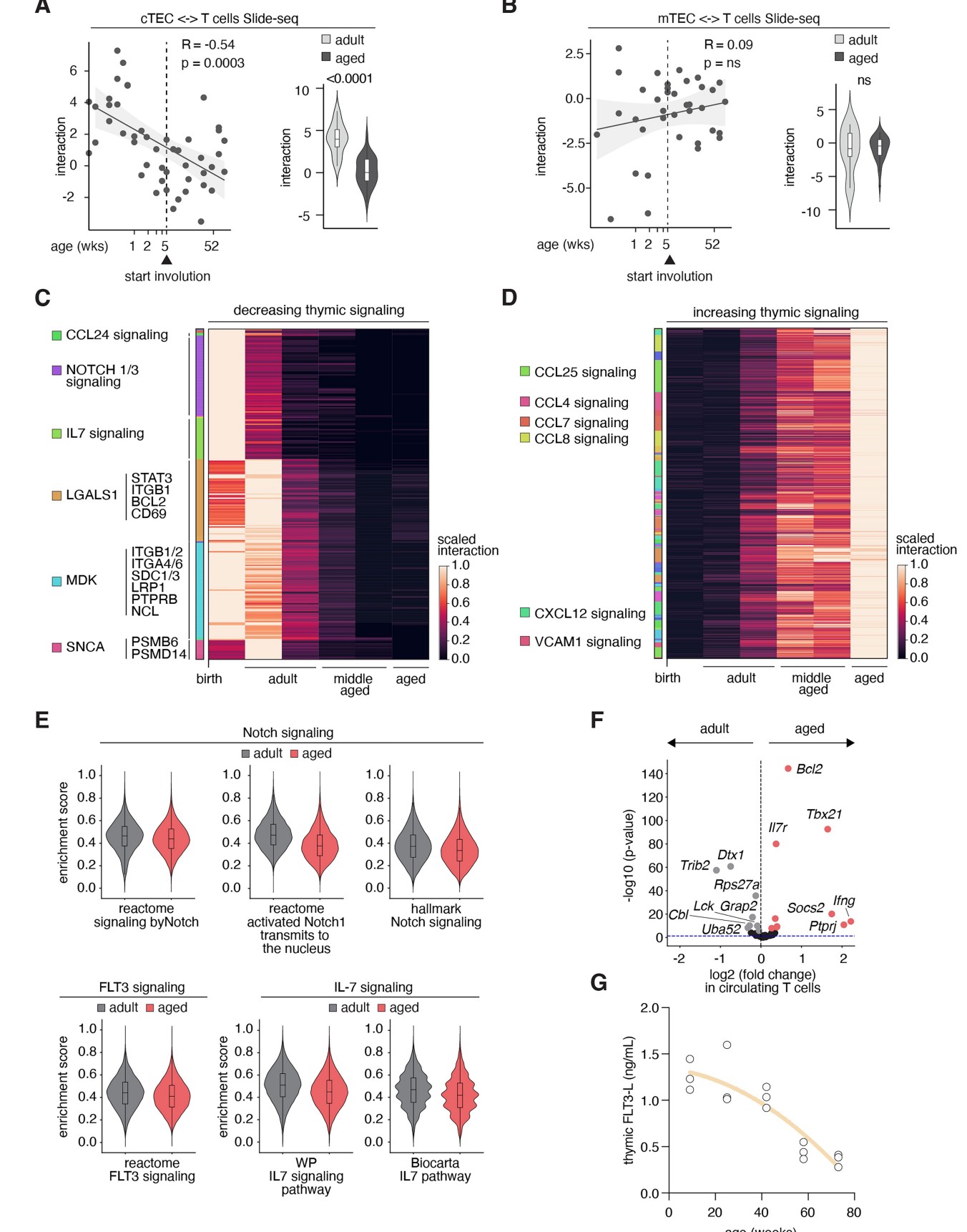

**Extended Data Fig. 3** | See next page for caption.

**Extended Data Fig. 3 | Age-driven decline in Notch, FLT3, and IL-7 signaling in intrathymic and peripheral T cells. (A-B)** Age-associated decline in spatial interaction strength between thymic cortical epithelial cells (cTEC) and thymocytes (A), and medullary epithelial cells (mTEC) and thymocytes (B). Interaction scores derived from ligand-receptor inference models. Pearson's correlation (R) and P values shown. Arrowheads indicate time of thymic involution. Violin plots display mean interaction scores before and after involution; significance assessed by two-tailed unpaired t-test. **(C-D)** Receptor-ligand interaction heatmaps showing cell-cell communication pathways that decrease (C) or increase (D) with age across thymic cell populations. Interactions grouped by signaling axis and stratified by age category: neonatal (n = 7 spatial arrays from 3 mice), adult (n = 15 from 7 mice), middle-aged (n = 13 from 6 mice), and aged (n = 10 from 4 mice). **(E)** Pathway-level enrichment of Notch, FLT3, and IL-7 signaling in peripheral T cells. Violin plots of average enrichment scores per mouse. Data from n = 96,683 peripheral blood T cells. **(F)** Differential expression of Notch, FLT3, and IL-7 target genes in circulating T cells of adult vs. aged mice. Shown as $\log_2$ fold change. Data from n = 96,683 peripheral blood T cells. **(G)** Absolute FLT3-L levels in explanted thymuses from mice across age (n = 3 animals per age group).

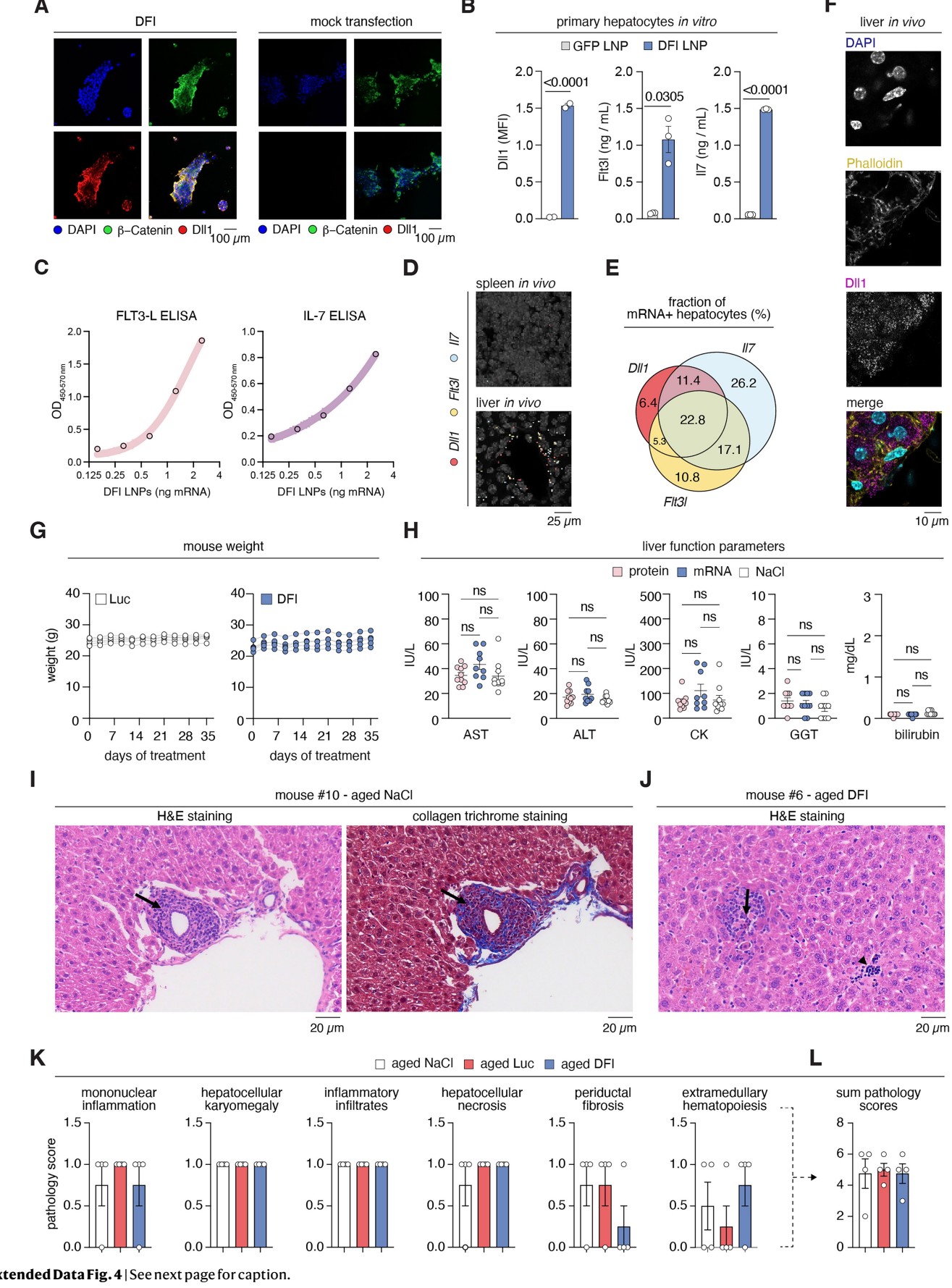

**Extended Data Fig. 4** | See next page for caption.

**Extended Data Fig. 4 | Translation and toxicity of DLL1-FLT3L-IL7 (DFI) mRNA–LNPs. (A-B)** Verification of DFI factor production by primary hepatocytes. (**A**) Confocal immunofluorescence of primary murine hepatocytes stained for DLL1 surface expression and β-catenin (membrane marker). Left panel: representative image following DFI transfection; right panel: mock-transfected control. Scale bar, 100 μm. (**B**) Extracellular concentrations of biologically active IL-7 and FLT3-L secreted by mRNA-transfected immortalized hepatocytes, as measured by IL-7 and FLT3-L ELISA. Data are mean ± s.e.m. n = 3 independent transfection experiments. Statistical significance was tested using two-tailed unpaired t-tests. (**C**) Supernatant cytokine levels of FLT3L and IL-7 from primary hepatocytes following DFI mRNA-LNP treatment, quantified by ELISA. (**D**) Representative RIBOmap deconvolved images of ribosome-bound transcripts in liver and spleen 6 h post-DFI injection. Note liver image (bottom) is repeated from Fig. 1f for ease of comparison. Representative image from three imaged DFI-treated animals. (**E**) Venn diagram showing overlap of hepatocytes expressing ribosome-bound transcripts for each of the three DFI mRNAs. n = 3 mice. (**F**) Representative immunofluorescence staining of liver tissue for DLL1 protein (yellow), F-actin (Phalloidin, magenta), and nuclei (DAPI, cyan) following DFI administration. See Fig. 1g for quantification. (**G**) Body weights of 72-week-old C57BL/6 J mice over 28 days of treatment with either Luc control or DFI mRNA–LNPs. n = 5 mice per group. (**H**) Serum liver function parameters including ALT, AST, ALP, and bilirubin measured in aged animals treated with vehicle (NaCl), DFI mRNA–LNPs, or recombinant IL-7/FLT3L protein. n = 10 (NaCl), n = 9 (DFI), n = 10 (recombinant cytokines). Data are presented as mean ± s.e.m. Statistical analysis was performed using one-way ANOVA followed by Tukey's multiple comparison test. (**I-J**) Representative hematoxylin and eosin (H&E) and Masson's trichrome stainings of liver sections from aged mice treated with NaCl (C) or DFI mRNA–LNPs (D). Representative images from one out of 12 imaged animals. (**K**) Histopathological scoring of liver lesions based on inflammation, hepatocellular degeneration, and fibrosis. Criteria followed INHAND nomenclature guidelines. n = 4 mice per group. (**L**) Cumulative pathology scores across all assessed features in aged NaCl-, Luc-, and DFI-treated mice. n = 4 mice per group.

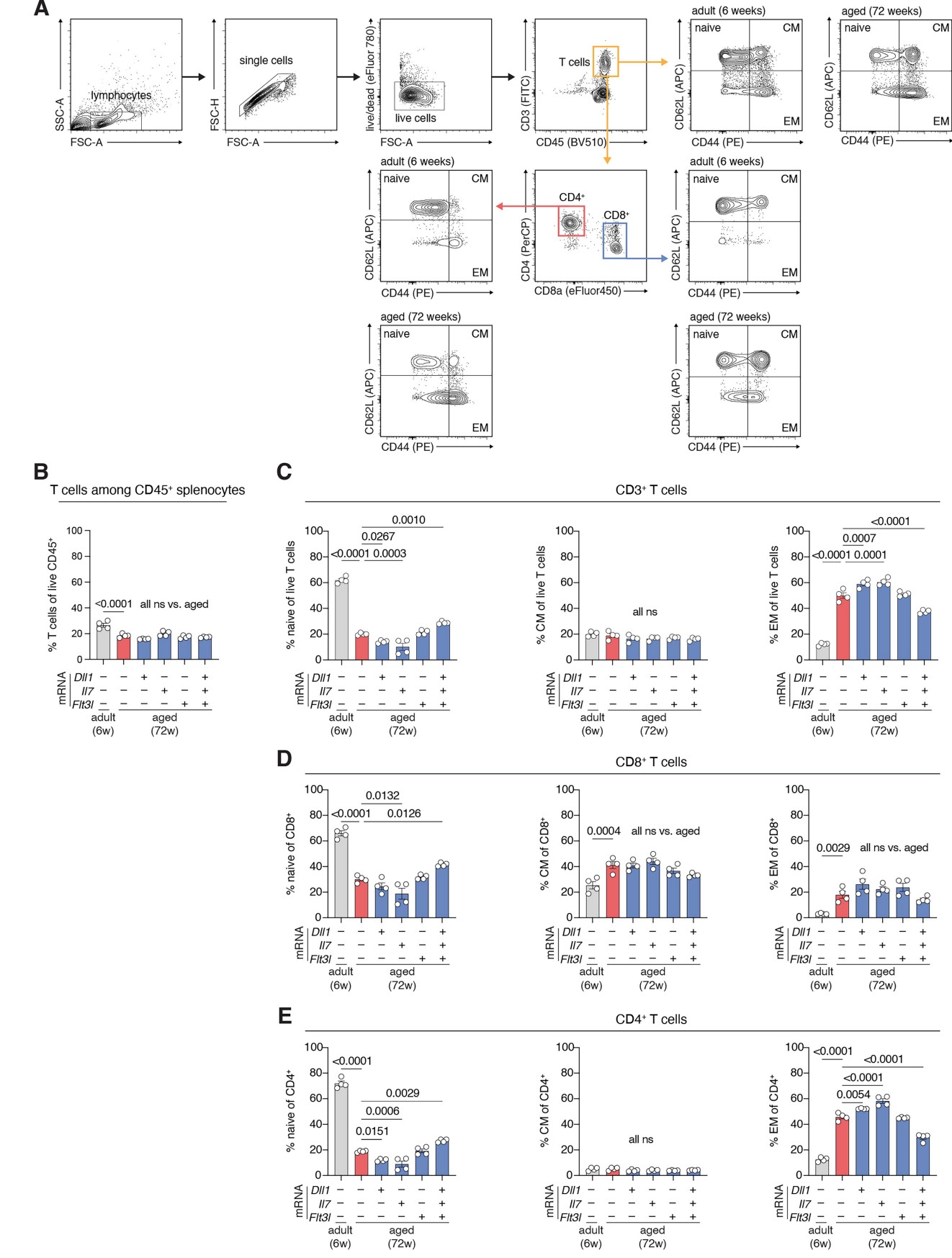

**Extended Data Fig. 5** | See next page for caption.

**Extended Data Fig. 5 | DFI treatment enhances peripheral naïve T cell populations in aged mice.** (**A**) Representative flow cytometry gating strategy for the identification and quantification of CD45$^+$ CD3$^+$ T cells in spleens from adult (6 weeks) and aged (72 weeks) mice. (**B**) Quantification of total CD45$^+$ CD3$^+$ T cell frequency following 28 days of biweekly DFI or Luc mRNA-LNP treatment. n = 4 mice per group. Data are shown as mean ± s.e.m. Statistical significance was assessed using one-way ANOVA followed by Tukey's post hoc test. (**C-E**) Frequencies of naïve (CD44$^-$ CD62L$^+$), central memory (CM; CD44$^+$ CD62L$^+$), and effector memory (EM; CD44$^+$ CD62L$^-$) T cells (**C**), CD8$^+$ T cells (**D**), and CD4$^+$ T cells (**E**). n = 4 mice per group. Data represent mean ± s.e.m. Statistical comparisons were performed by one-way ANOVA with Tukey's post hoc test.

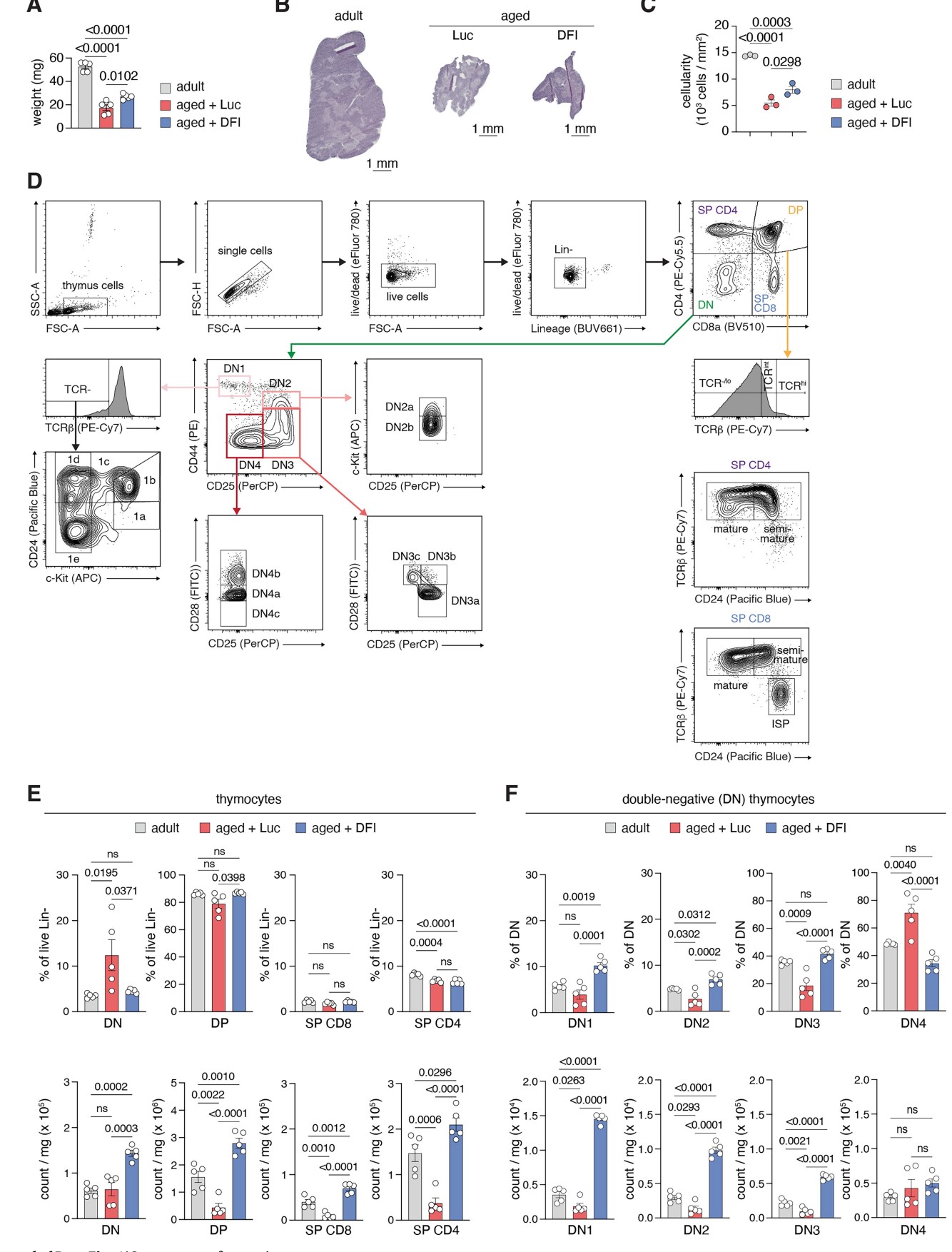

**Extended Data Fig. 6** | See next page for caption.

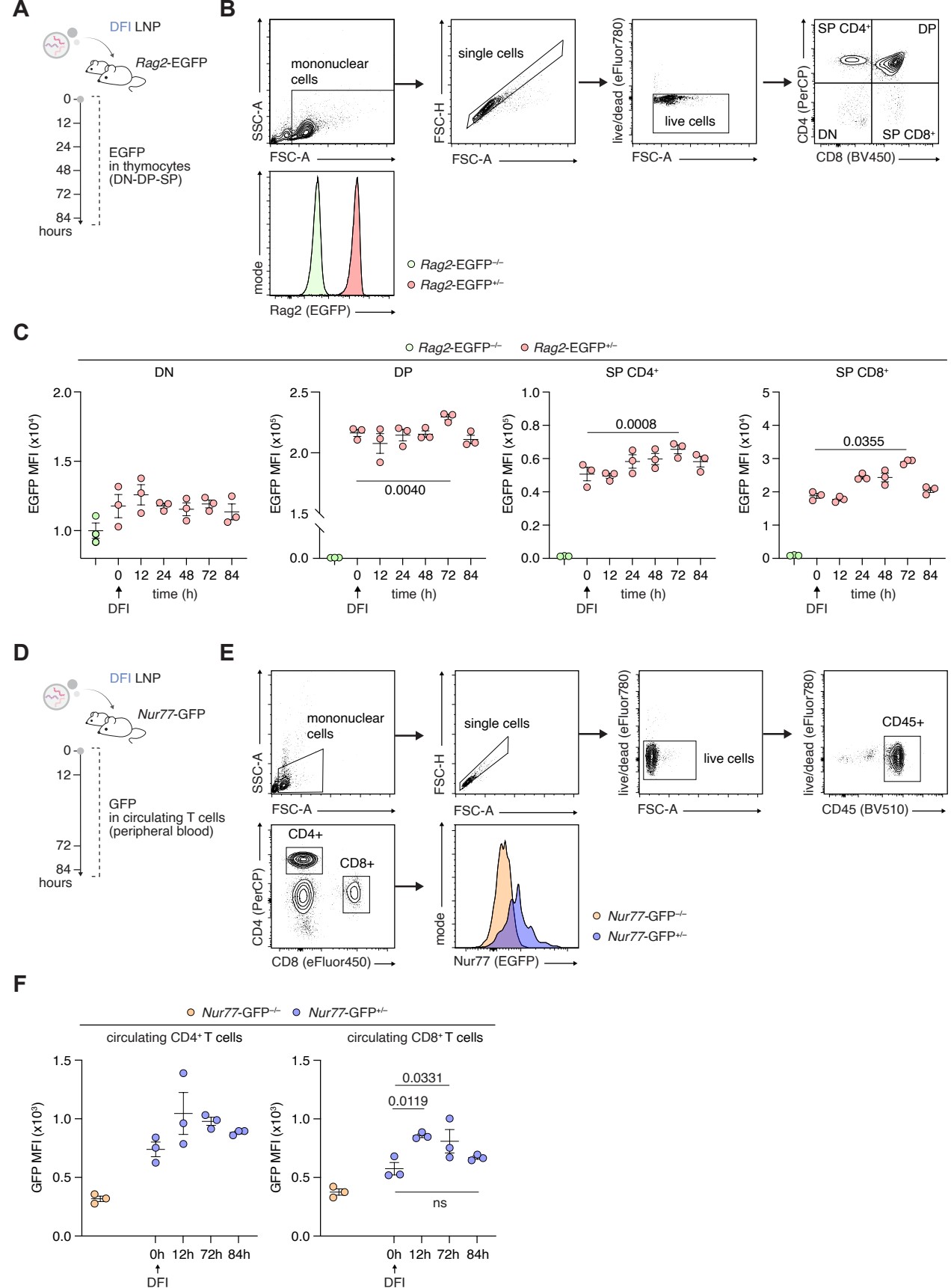

**Extended Data Fig. 6 | DFI promotes thymopoiesis and restores thymocyte developmental stages in aged mice.** (**A**) Weights of explanted thymuses from adult (6 weeks), aged + Luc mRNA, and aged + DFI mRNA-treated (72 weeks) mice. n = 5 mice per group. Statistical significance was determined by one-way ANOVA followed by Tukey's multiple-comparison post hoc test. (**B**) Representative H&E-stained thymic tissue sections from each group, highlighting cortical and medullary organization. (**C**) Quantification of thymic cellularity ($\times 10^3$ cells/mm$^2$) from explanted thymuses. n = 3 mice per group. One-way ANOVA with Tukey's post hoc test was used to assess statistical significance. (**D**) Representative flow cytometry gating strategy for quantification of thymocyte subsets from explanted thymuses. (**E**) Quantification of major thymocyte subsets: double-negative (DN), double-positive (DP), and single-positive (SP) CD4$^+$ and CD8$^+$ thymocytes. Absolute counts were normalized to thymus weight (mg). n = 5 mice per group. Data are shown as mean ± s.e.m., one-way ANOVA with Tukey's post hoc test. (**F**) Absolute counts of DN1-DN4 subsets. Normalized to thymus weight (mg). n = 5 mice per group. Data are shown as mean ± s.e.m., one-way ANOVA with Tukey's post hoc test.

**Extended Data Fig. 7** | See next page for caption.

**Extended Data Fig. 7 | Analysis of Rag2- and Nur77 expression in thymocytes and T cells.** (**A**) Overview of the *Rag2*-EGFP experiment. DFI was injected at 0 h. (**B**) Representative flow cytometry gating strategy to quantify *Rag2* expression levels in double- negative (DN), double-positive (DP), single-positive CD4$^+$ (SP CD4$^+$) and single-positive CD8$^+$ (SP CD8$^+$) thymocytes. (**C**) Mean fluorescence intensity (MFI) of *Rag2*-EGFP at indicated timepoints. n = 3 mice per timepoint. n = 3 Rag2-EGFP$^{-/-}$ mice were used as negative control. Data are shown as mean ± s.e.m., repeated measures two-way ANOVA with Dunnett's post hoc test.

(**D**) Overview of the *Nur77*-GFP experiment. DFI was injected at 0 h and 72 h. (**E**) Representative flow cytometry gating strategy to quantify *Nur77* expression levels in CD4$^+$ and CD8$^+$ T cells in peripheral blood. (**F**) Mean fluorescence intensity (MFI) of *Nur77*-GFP at indicated timepoints in CD4$^+$ (left) and CD8$^+$ (right) T cells in peripheral blood. n = 3 mice per timepoint. n = 3 *Nur77*-GFP$^{-/-}$ mice were used as negative control. Data are shown as mean ± s.e.m., one-way ANOVA with Dunnett's post hoc test.

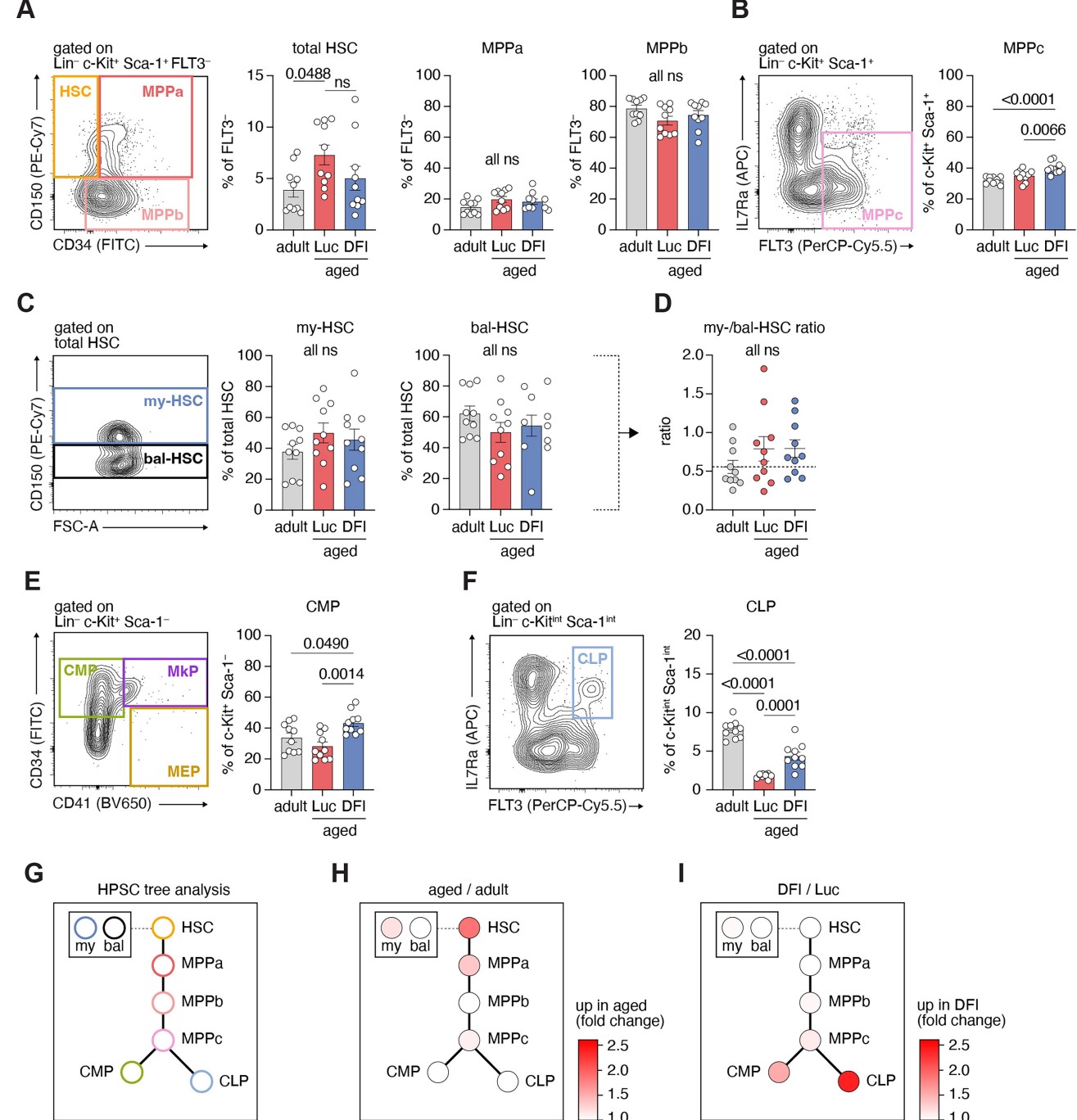

**Extended Data Fig. 8 | Modulation of committed progenitors in aged bone marrow by DFI.** (**A**) Representative flow cytometry gating (left) and quantification (right) of total hematopoietic stem cells (HSCs; Lin⁻ c-Kit⁺ Sca-1⁺ FLT3⁻ CD34⁻ CD150⁺), multipotent progenitor a (MPPa; Lin⁻ c-Kit⁺ Sca-1⁺ FLT3⁻ CD34⁺ CD150⁺), and multipotent progenitor b (MPPb; Lin⁻ c-Kit⁺ Sca-1⁺ FLT3⁻ CD34⁺ CD150⁻), as described previously[17]. (**B**) Gating (left) and quantification (right) of multipotent progenitor c (MPPc; Lin⁻ c-Kit⁺ Sca-1⁺ FLT3⁺ CD34⁺ CD150⁻), as described previously[17]. (**C**) Gating (left) and quantification (right) of myeloid-biased HSCs (my-HSCs; Lin⁻ c-Kit⁺ Sca-1⁺ FLT3⁻ CD34⁻ CD150ʰⁱᵍʰ) and balanced HSCs (bal-HSCs; Lin⁻ c-Kit⁺ Sca-1⁺ FLT3⁻ CD34⁻ CD150ˡᵒʷ). Gate to define my-HSC vs. bal-HSC was set as described previously[17]. (**D**) Ratio of my-HSCs to bal-HSCs across experimental groups. (**E**) Gating (left) and quantification (right) of

common myeloid progenitors and granulocyte–macrophage progenitors (CMP/GMP; Lin⁻ c-Kit⁺ Sca-1⁻ CD34⁺ CD41⁻) (**F**) Gating (left) and quantification (right) of common lymphoid progenitors (CLPs; Lin⁻ c-Kitⁱⁿᵗ Sca-1ⁱⁿᵗ IL-7Rα⁺ FLT3⁺). (**G**) Schematic illustration of the hierarchical hematopoietic stem and progenitor cell (HSPC) lineage tree used for comparative analysis. (**H**) Fold changes in average frequencies of each HSPC subset in aged relative to adult bone marrow. (**I**) Fold changes in average frequencies of each HSPC subset in aged DFI-treated relative to aged Luc-treated bone marrow. For (**A-F**), quantifications were performed on n = 30 mice total (n = 10 adult, n = 10 aged + Luc, n = 10 aged + DFI). Data represent mean ± s.e.m. Statistical significance was assessed by one-way ANOVA followed by Tukey's multiple-comparison post hoc test.

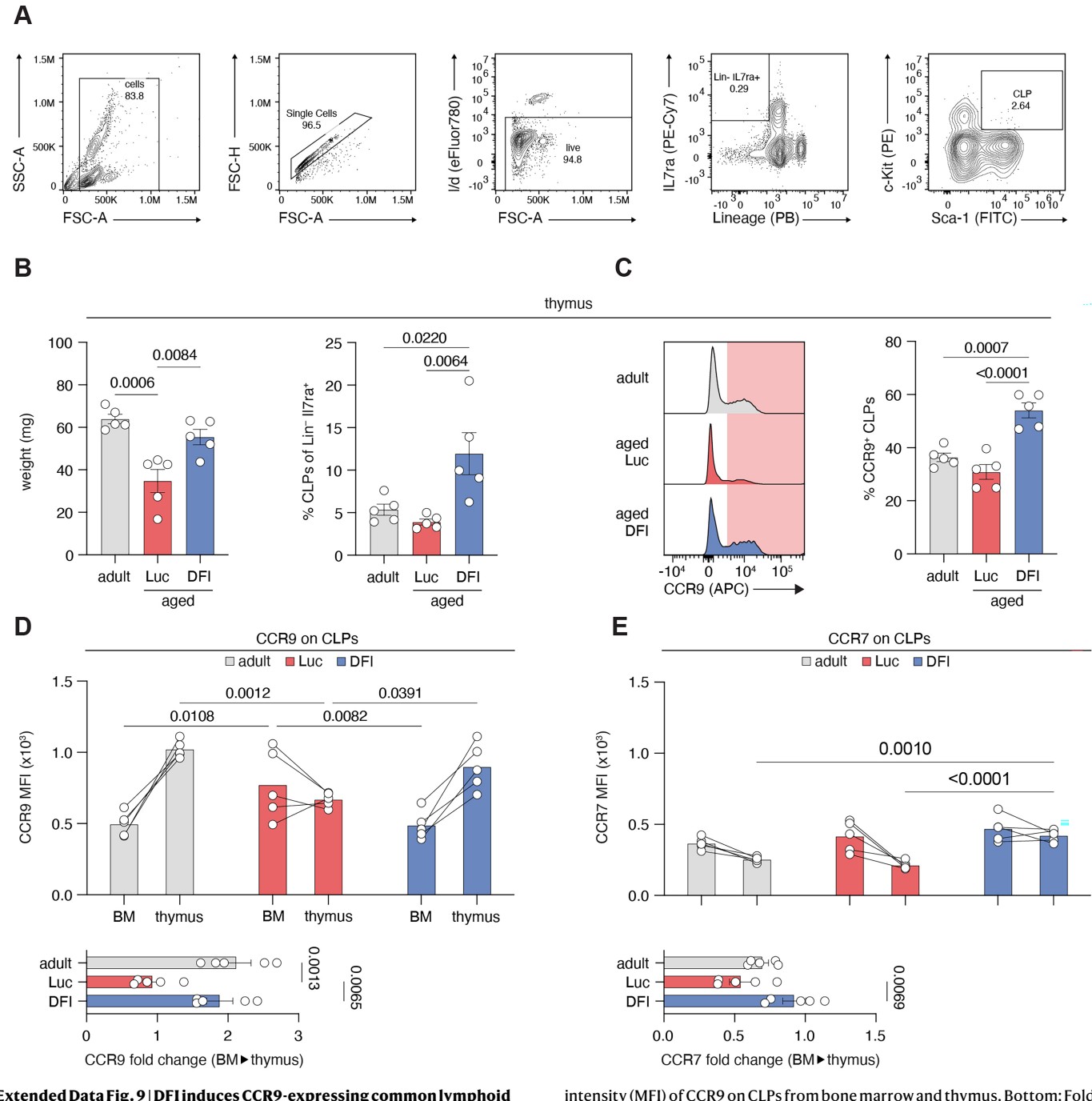

**Extended Data Fig. 9 | DFI induces CCR9-expressing common lymphoid progenitors (CLPs) in aged mice.** (**A**) Representative flow cytometry gating strategy to identify circulating CLPs (Lin⁻ IL-7Rα⁺ Sca-1⁺ c-Kit⁺) in peripheral blood. (**B**) Left: Weights of explanted thymuses from each group. Right: Frequencies of intrathymic CLPs. n = 5 mice per group. (**C**) Left: Representative histograms showing CCR9 surface expression on intrathymic CLPs. Right: Frequencies of CCR9⁺ CLPs in thymus. n = 5 per group. (**D**) Top: Mean fluorescence intensity (MFI) of CCR9 on CLPs from bone marrow and thymus. Bottom: Fold change in CCR9 expression (thymus:bone marrow). n = 5 mice per group. Data analyzed by repeated measures two-way ANOVA and Tukey's post hoc test. (**E**) Top: MFI of CCR7 on CLPs from bone marrow and thymus. Bottom: Fold change in CCR7 expression (thymus:bone marrow). n = 5 mice per group. Statistical analyses as in (**D**).

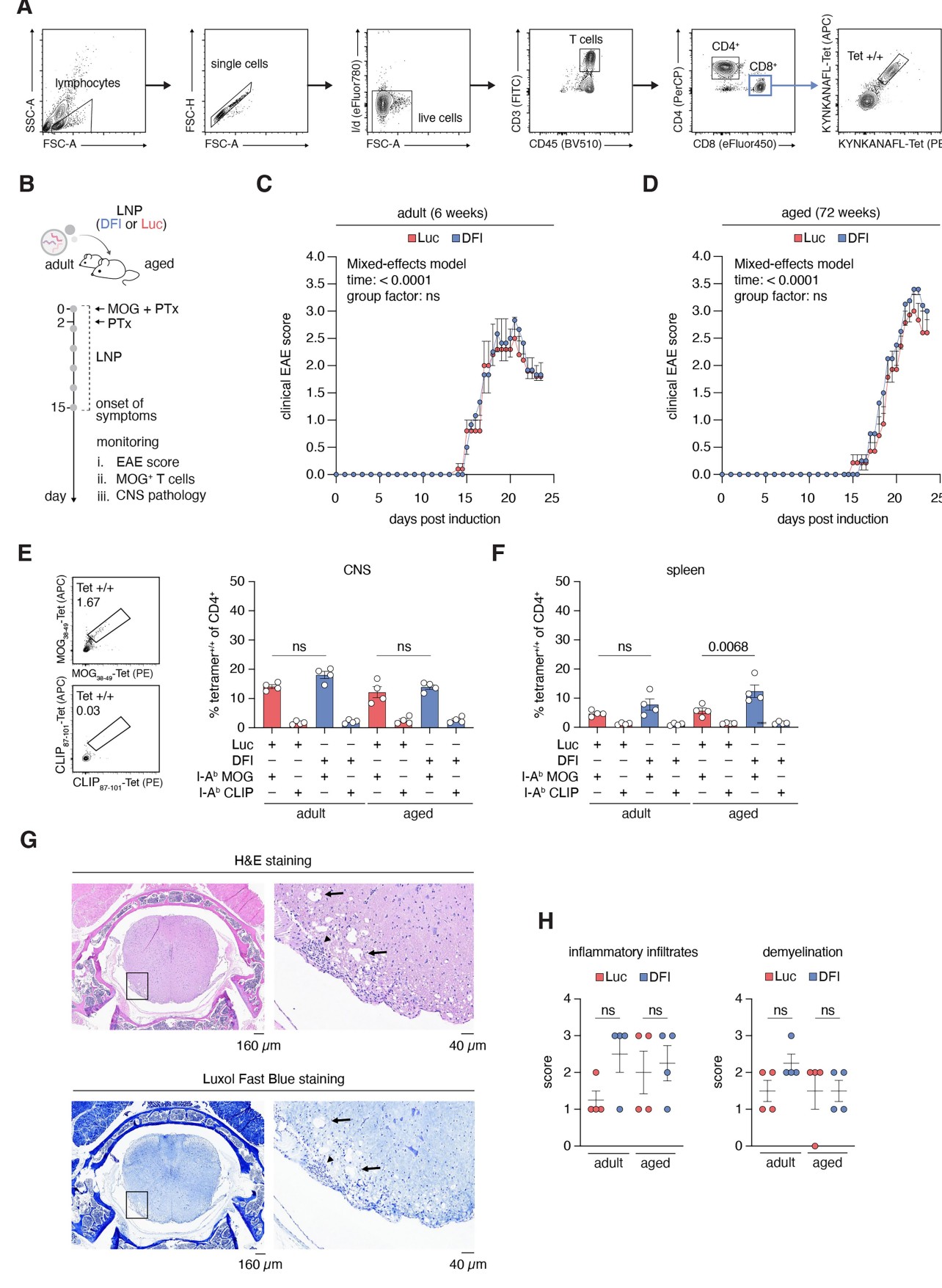

**Extended Data Fig. 10** | See next page for caption.

**Extended Data Fig. 10 | DFI treatment does not exacerbate autoimmune responses in T1D and EAE models.** (**A**) Gating strategy for tetramer staining of KYNKANAFL-H-2$^d$-specific CD8$^+$ T cells in the peripheral blood of NOD mice. (**B**) Schematic of the experimental design for experimental autoimmune encephalomyelitis (EAE) induction in adult and aged mice treated with DFI or Luc mRNA. Mice were immunized with MOG$_{35-55}$ peptide and pertussis toxin, treated with mRNA–LNPs every 3 days until symptom onset, and monitored over 28 days. (**C-D**) Longitudinal EAE clinical scores in (C) adult and (D) aged mice. Mice were scored twice daily using a 0–5 scale: 0, no signs; 1, limp tail; 2, hindlimb weakness; 3, hindlimb paralysis; 4, forelimb paralysis; 5, moribund. (**E-F**) Frequencies of MOG$_{38-49}$-I-A$^b$ tetramer-binding CD4$^+$ T cells in spleens. Control tetramer: CLIP$_{87-101}$-I-A$^b$. n = 4 per group. (**G**) Representative spinal cord sections stained with H&E (top) and Luxol Fast Blue (bottom) to evaluate inflammatory infiltrates and demyelination. Arrows indicate areas of demyelination; arrowheads mark inflammatory foci. (**H**) Quantification of demyelination severity in spinal cords. n = 4 per group. P values calculated using one-way ANOVA with Tukey's multiple-comparison post hoc test.

# Reporting Summary

## Statistics

For all statistical analyses, confirm that the following items are present in the figure legend, table legend, main text, or Methods section.

| n/a | Confirmed | |
|-----|-----------|---|
| ☐ | ☒ | The exact sample size (*n*) for each experimental group/condition, given as a discrete number and unit of measurement |
| ☐ | ☒ | A statement on whether measurements were taken from distinct samples or whether the same sample was measured repeatedly |
| ☐ | ☒ | The statistical test(s) used AND whether they are one- or two-sided <br> *Only common tests should be described solely by name; describe more complex techniques in the Methods section.* |
| ☐ | ☒ | A description of all covariates tested |
| ☐ | ☒ | A description of any assumptions or corrections, such as tests of normality and adjustment for multiple comparisons |
| ☐ | ☒ | A full description of the statistical parameters including central tendency (e.g. means) or other basic estimates (e.g. regression coefficient) AND variation (e.g. standard deviation) or associated estimates of uncertainty (e.g. confidence intervals) |
| ☐ | ☒ | For null hypothesis testing, the test statistic (e.g. *F*, *t*, *r*) with confidence intervals, effect sizes, degrees of freedom and *P* value noted <br> *Give P values as exact values whenever suitable.* |
| ☒ | ☐ | For Bayesian analysis, information on the choice of priors and Markov chain Monte Carlo settings |
| ☒ | ☐ | For hierarchical and complex designs, identification of the appropriate level for tests and full reporting of outcomes |
| ☐ | ☒ | Estimates of effect sizes (e.g. Cohen's *d*, Pearson's *r*), indicating how they were calculated |

*Our web collection on statistics for biologists contains articles on many of the points above.*

## Software and code

Policy information about availability of computer code

| Data collection | No custom software was used for data collection. |
|-----------------|--------------------------------------------------|

| Data analysis | Raw sequencing data were processed and aligned to the mause genome (GRCm39 - mm39) using the CellRanger pipeline (10x Genomics, version 7.1.0). |
|---|---|
| | For IVIS imaging, the Aura imaging software (v4.0) was used. |
| | The mouse genome assembly (GRCm39) was used for data alignment (NCBI RefSeq assembly; GCF_000001635.27). |
| | For spatial sequencing data, we used the Squidpy (v1.4.0) integration of CellPhoneDB and Omnipath to identify shifts in receptor-ligand interactions at each time point, which are each publicly available. |
| | For single-cell sequencing data, we used Seurat v5.3.0 (for single-cell sequencing analysis), Harmony v1.0 (for single-cell data integration), Scanpy vl.8.2 and Matplotlib v3.8, which are each publicly available. |
| | STARmap data analysis was performed similarly to Wang et al and Shi et al. Image deconvolution was achieved with Huygens Essential version 21.04 (Scientific Volume Imaging, The Netherlands, http://svi.nl), using the CMLE algorithm, with SNR:10 and 10 iterations. Image registration, spot calling, and barcode filtering were performed as previously described by the above-mentioned publications. |
| | All code necessary to reproduce all spatial thymus RNA-seq analyses is available at: https://github.com/immunoliugy/aging_mouse_thymus. All code necessary to reproduce all RIBOmap / STARmap PLUS analyses is available at: https://github.com/wanglab-broad/hepatic_recon. |

For manuscripts utilizing custom algorithms or software that are central to the research but not yet described in published literature, software must be made available to editors and reviewers. We strongly encourage code deposition in a community repository (e.g. GitHub). See the Nature Portfolio guidelines for submitting code & software for further information.

## Data

Policy information about availability of data

All manuscripts must include a data availability statement. This statement should provide the following information, where applicable:
- Accession codes, unique identifiers, or web links for publicly available datasets
- A description of any restrictions on data availability
- For clinical datasets or third party data, please ensure that the statement adheres to our policy

All single-cell profiling data of circulating T cells and B16-OVA TILs in aging mice are available at Zenodo (accession number 17385897). All spatial profiling (RIBOmap / STARmap PLUS) data of liver and spleen samples are available at Zenodo (accession number 13868686). All spatial profiling (Slide-TCR-seq) data of thymuses across age are available at Zenodo (accession number 17525728). The processed spatial thymus dataset across age is publicly available at the Broad Single Cell Portal (accession number SCP2424). The mouse genome assembly (GRCm39) was used for data alignment (NCBI RefSeq assembly; GCF_000001635.27).

## Research involving human participants, their data, or biological material

Policy information about studies with human participants or human data. See also policy information about sex, gender (identity/presentation), and sexual orientation and race, ethnicity and racism.

| Reporting on sex and gender | No human participants, their materials, or their data were included in this study. |
|---|---|
| Reporting on race, ethnicity, or other socially relevant groupings | No human participants, their materials, or their data were included in this study. |
| Population characteristics | No human participants, their materials, or their data were included in this study. |
| Recruitment | No human participants, their materials, or their data were included in this study. |
| Ethics oversight | No human participants, their materials, or their data were included in this study. |

Note that full information on the approval of the study protocol must also be provided in the manuscript.

# Field-specific reporting

Please select the one below that is the best fit for your research. If you are not sure, read the appropriate sections before making your selection.

☒ Life sciences ☐ Behavioural & social sciences ☐ Ecological, evolutionary & environmental sciences

For a reference copy of the document with all sections, see nature.com/documents/nr-reporting-summary-flat.pdf

# Life sciences study design

All studies must disclose on these points even when the disclosure is negative.

| Sample size | The sample size for vaccination experiments was decided based on prior publications with similar experiments [Zhivaki, D. et al. Correction of age-associated defects in dendritic cells enables CD4+ T cells to eradicate tumors. Cell 187, 3888–3903.e18 (2024). |

The sample size for tumor experiments was decided based on prior publications with similar experiments [Tsukamoto, H. et al. Aging-associated and CD4 T-cell-dependent ectopic CXCL13 activation predisposes to anti-PD-1 therapy-induced adverse events. Proc. Natl. Acad. Sci. U. S. A. 119, e2205378119 (2022) and Georgiev, P. et al. Age-associated contraction of tumor-specific T cells impairs antitumor immunity. Cancer Immunol. Res. 12, 1525–1541 (2024)].

The sample size for NOD experiments was decided based on prior publications with similar experiments [Trudeau, J. D. et al. Prediction of spontaneous autoimmune diabetes in NOD mice by quantification of autoreactive T cells in peripheral blood. J. Clin. Invest. 111, 217–223 (2003)].

The sample size for EAE experiments was decided based on prior publications with similar experiments [Sanmarco, L. M. et al. Lactate limits CNS autoimmunity by stabilizing HIF-1α in dendritic cells. Nature 620, 881–889 (2023)].

For all other exploratory experiments, no sample size calculations were performed.

| | |
|---|---|
| Data exclusions | For animal experiments other than survival studies, mice diseased during the experiment were excluded from the analyses. From longitudinal single-cell RNA/V(D)J sequencing experiments, one sampling timepoint was excluded. |
| Replication | All attempts at replication were successful. Replicated experiments were identically or similarly designed. In vivo experiments were replicated using the same mouse strains and facilities at the Broad Institute. NOD and EAE experiments were not repeated. |
| Randomization | For all experiments, allocation of mice into experimental groups was randomized after stratifying for age and sex. |
| Blinding | Separate investigators performed treatment and data collection. Data collecting investigators, e.g., for tumor size measurements, were blinded to the treatment groups. Data analyzing investigators were not blinded to the treatment groups, as they involved internal controls, with the exception of pathologists for toxicity studies, who were blinded for the analyses. Experimental and control animals were treated equally and, when possible, housed in mixed cages. |

# Reporting for specific materials, systems and methods

We require information from authors about some types of materials, experimental systems and methods used in many studies. Here, indicate whether each material, system or method listed is relevant to your study. If you are not sure if a list item applies to your research, read the appropriate section before selecting a response.

## Materials & experimental systems

| n/a | Involved in the study |
|---|---|
| ☐ | ☒ Antibodies |
| ☐ | ☒ Eukaryotic cell lines |
| ☒ | ☐ Palaeontology and archaeology |
| ☐ | ☒ Animals and other organisms |
| ☒ | ☐ Clinical data |
| ☒ | ☐ Dual use research of concern |
| ☒ | ☐ Plants |

## Methods

| n/a | Involved in the study |
|---|---|
| ☒ | ☐ ChIP-seq |
| ☐ | ☒ Flow cytometry |
| ☒ | ☐ MRI-based neuroimaging |

## Antibodies

| | |
|---|---|
| Antibodies used | Immunohistochemistry, immunofluorescence and STARmap analyses of DLL1: anti-DLL1 antibody 1:800 (Abcam ab10554) and AF546-labeled secondary donkey anti-rabbit IgG antibody 1:500 (Invitrogen A10040).<br><br>Mouse T cell analyses: anti-mouse IL-2 PE 1:100 (Biolegend; 503808; clone JES6-5H4; lot B351622, B377599), anti-mouse CD45 BV510 1:100 (Biolegend; 103138; clone 30-F11; lot B386738, B360620, B384034), anti-mouse IFN-γ APC 1:100 (Biolegend; 505810; clone XMG1.2; lot B354911, B370994, B396246), anti-mouse/human CD44 PE 1:100 (Biolegend; 103024; clone IM7; lot B343363), Anti-Mo CD8a eBioscience eFluor 450 1:100 (Invitrogen; 48-0081-82; clone 53-6.7; lot 2527379), Rat Anti-Mouse CD4 PerCP 1:100 (BD Bioscience; 553052; clone RM4-5; lot 2279727, 3201956, 1334056), anti-mouse CD62L APC 1:100 (Biolegend; 104412; clone MEL-14; lot B371017), anti-mouse CD3 FITC 1:100 (Biolegend; 100204; clone 17A2; lot B388315, B388061, B406287), Hamster Anti-Mouse TCRβ APC 1:100 (BD bioscience; 553174; clone H57-597; lot 3030398, 2076848), TruStain FcX anti-mouse CD16/32 1:50 (Biolegend; 101320; clone 93; lot B398113, B380119, B368516, B372578, B419152), eBioscience Fixable Viability Dye eFluor 780 1:1000 (Invitrogen; 65-0865-14; lot 2752774), anti-mouse CD4 PE-Cy5.5 1:500 (Biolegend; 100514; clone RM4-5; lot B398342), anti-mouse CD8a BV510 1:100 (Biolegend; 100752; clone 53-6.7; lot B427044), anti-mouse/human CD44 PE 1:200 (Biolegend; 103024; clone IM7; lot B343363), anti-mouse CD62L APC 1:100 (Biolegend; 104412; clone MEL-14; lot B371017), anti-mouse CD3 Pacific Blue 1:100 (Biolegend; 100214; clone 17A2; lot B427533), anti-mouse CD279 (PD-1) APC 1:100 (Biolegend; 135210; clone 29F.1A12; lot B376789).<br><br>Mouse thymus analyses: anti-mouse CD4 PE-Cy5.5 1:500 (Biolegend; 100514; clone RM4-5; lot B398342), anti-mouse CD8a BV510 1:100 (Biolegend; 100752; clone 53-6.7; lot B427044), anti-mouse CD25 PerCP 1:200 (Biolegend; 102028; clone PC61; lot B378943), anti-mouse/human CD44 PE 1:200 (Biolegend; 103007; clone IM7; lot B417002), Rat Anti-Mouse CD117 APC 1:100 (BD bioscience; 553356; clone 2B8; lot 3199305), anti-mouse CD28 FITC 1:100 (Biolegend; 122008; clone E18; lot B373960), anti-mouse CD24 Pacific Blue 1:500 (Biolegend; 101820; clone M1/69; lot B385383), anti-mouse TCR β chain PE/Cyanine7 1:400 (Biolegend; 109222; clone |

H57-597; lot B394527), Rat Anti-Mouse CD45R/B220 BUV661 1:100 (Bd bioscience; 612972; clone RA3-6B2; lot 3292032), TruStain FcX anti-mouse CD16/32 1:50 (Biolegend; 101320; clone 93; lot B398113, B380119, B368516, B372578, B419152), eBioscience Fixable Viability Dye eFluor 780 1:1000 (Invitrogen; 65-0865-14; lot 2752774).

Tetramer stainings:
For OVA vaccination experiments, SIINFEKL-H-2K$^b$-PE and SIINFEKL-H-2K$^b$-APC tetramers were used at 1:100 dilution and AAHAEINEA-I-A$^b$-PE and AAHAEINEA-I-A$^b$-APC tetramers were used at 1:20 dilution. For central tolerance experiments in Act-mOVA mice, the same SIINFEKL-H-2K$^b$ tetramers (1:100 dilution) were used in combination with AAHAEINEA-I-A$^b$-PE and AAHAEINEA-I-A$^b$-APC tetramers (1:20 dilution). Non vaccinated wild-type T cells (negative) and OT-I and OT-II T cells (positive) were included as staining controls for MHC class I and II tetramers, respectively.

For autoimmunity experiments in NOD mice, KYNKANAFL-H-2K$^d$-PE and KYNKANAFL-H-2K$^d$-APC tetramers were used at 1:50 dilution, with NY8.3 T cells serving as positive controls for staining.

For EAE experiments in C57BL/6J mice, GWYRSPFSRVVH-I-A$^b$-PE and GWYRSPFSRVVH-I-A$^b$-APC tetramers were used at 1:25 dilution. Control tetramers consisted of I-A$^b$-restricted human CLIP87-101 (PVSKMRMATPLLMQA) conjugated to PE and APC, also at 1:25 dilution.

Validation

All antibodies and tetramers used in this study have been previously validated by commercial manufactures, previous publications, and/or this study.

Immunohistochemistry, immunofluorescence and STARmap analyses of DLL1:
anti-DLL1 antibody (Abcam ab10554), reported to recognize mouse, human and rat DLL1 (https://www.abcam.com/en-us/products/primary-antibodies/dll1-antibody-ab10554)
AF546-labeled secondary donkey anti-rabbit IgG antibody, reported to recognized rabbit IgG (manufacturer's website)

Flow cytometry and fluorescence-activated cell sorting:
PE anti-mouse IL-2 Antibody (Biolegend; 503808), reported to recognize mouse IL-2 (https://www.biolegend.com/en-us/products/pe-anti-mouse-il-2-antibody-954)
Violet 510™ anti-mouse CD45 Antibody BV510 (Biolegend; 103138), reported to recognize mouse CD45 (https://www.biolegend.com/en-us/products/brilliant-violet-510-anti-mouse-cd45-antibody-7995)
APC anti-mouse IFN-γ Antibody (Biolegend; 505810), reported to recognize mouse IFN- γ (https://www.biolegend.com/en-us/products/apc-anti-mouse-ifn-gamma-antibody-993)
PE anti-mouse/human CD44 Antibody (Biolegend; 103024), reported to recognize mouse/human CD44 (https://www.biolegend.com/en-us/products/pe-anti-mouse-human-cd44-antibody-2206)
CD8a Monoclonal Antibody (53-6.7), eFluor 450 (Invitrogen; 48-0081-82), reported to recognize mouse CD8a (https://www.thermofisher.com/antibody/product/CD8a-Antibody-clone-53-6-7-Monoclonal/48-0081-82)
Phamingen PerCP Rat Anti-Mouse CD4 (BD Bioscience; 553052), reported to recognize mouse (https://www.bdbiosciences.com/en-de/products/reagents/flow-cytometry-reagents/research-reagents/single-color-antibodies-ruo/percp-rat-anti-mouse-cd4.553052?tab=product_details)
FITC anti-mouse CD3 Antibody (Biolegend; 100204), reported to recognize mouse CD3 (https://www.biolegend.com/nl-be/products/fitc-anti-mouse-cd3-antibody-45)
BD Pharmingen APC Hamster Anti-Mouse TCR ß Chain (BD bioscience; 553174), reported to recognize mouse TCRβ (https://www.bdbiosciences.com/en-de/products/reagents/flow-cytometry-reagents/research-reagents/single-color-antibodies-ruo/apc-hamster-anti-mouse-tcr-chain.553174?tab=product_details)
APC anti-mouse CD62L Antibody (Biolegend; 104412), reported to recognize mouse CD62L (https://www.biolegend.com/en-us/products/apc-anti-mouse-cd62l-antibody-381)
Pacific Blue anti-mouse CD3 (Biolegend; 100214), reported to recognize mouse CD3 (https://www.biolegend.com/en-us/products/pacific-blue-anti-mouse-cd3-antibody-3317)
APC anti-mouse CD279 /PD-1) Antibody (PD-1) APC (Biolegend; 135210), reported to recognize mouse CD279 (https://www.biolegend.com/en-us/products/apc-anti-mouse-cd279-pd-1-antibody-6497)
PE/Cyanine5 anti-mouse CD4 Antibody (Biolegend; 100514), reported to recognize mouse CD4 (https://www.biolegend.com/en-us/products/pe-cyanine5-anti-mouse-cd4-antibody-483)
Violet 510 anti-mouse CD8a Antibody (Biolegend; 100752), reported to recognize mouse CD8a (https://www.biolegend.com/en-us/products/brilliant-violet-510-anti-mouse-cd8a-antibody-7992)
PerCP anti-mouse CD25 Antibody (Biolegend; 102028), reported to recognize mouse CD25 (https://www.biolegend.com/en-us/products/percp-anti-mouse-cd25-antibody-4263)
PE anti-mouse/human CD44 Antibody (Biolegend; 103007), reported to recognize mouse/human CD44 (https://www.biolegend.com/en-us/products/pe-anti-mouse-human-cd44-antibody-2206)
Rat Anti-Mouse CD117 APC (BD bioscience; 553356), reported to recognize mouse CD117
FITC anti-mouse CD28 Antibody (Biolegend; 122008), reported to recognize mouse CD28 (https://www.biolegend.com/en-us/products/fitc-anti-mouse-cd28-antibody-3777)
Pacific Blue anti-mouse CD24 Antibody (Biolegend; 101820), reported to recognize mouse CD24 (https://www.biolegend.com/en-us/products/pacific-blue-anti-mouse-cd24-antibody-3584)
PE/Cyanine7 anti-mouse TCR ß chain Antibody (Biolegend; 109222), reported to recognize mouse TCR β chain (https://www.biolegend.com/en-us/products/pe-cyanine7-anti-mouse-tcr-beta-chain-antibody-4144)
Horizon BUV661 Rat Anti-Mouse CD45/RB220 (BD Bioscience; 612972), reported to recognize mouse CD45R/B220 BUV661 (https://www.bdbiosciences.com/en-de/products/reagents/flow-cytometry-reagents/research-reagents/single-color-antibodies-ruo/buv661-rat-anti-mouse-cd45r-b220.612972?tab=product_details)
 (c-Kit) Monoclonal Antibody (2B8), APC-eFluor 780 (ThermoFischer; 47-1171-82), reported to recognize mouse and pig cKit (https://www.thermofisher.com/antibody/product/CD117-c-Kit-Antibody-clone-2B8-Monoclonal/47-1171-82)
BD Horizon BUV395 Rat Anti-Mouse Ly-6A/E (BD; 744328), reported to recognize mouse SCA1 (https://www.bdbiosciences.com/en-de/products/reagents/flow-cytometry-reagents/research-reagents/single-color-antibodies-ruo/buv395-rat-anti-mouse-ly-6a-e.563990?tab=product_details)
BD Horizon BUV737 Mouse Anti-Human CD45 (367-0451-82), reported to recognize mouse CD45 (https://www.bdbiosciences.com/

en-de/products/reagents/flow-cytometry-reagents/research-reagents/single-color-antibodies-ruo/buv737-mouse-anti-human-cd45.568524?tab=product_details)
BD OptiBuild BV650 Rat Anti-Mouse CD41 (BD; 740504), reported to recognize mouse CD41 (https://www.bdbiosciences.com/en-eu/products/reagents/flow-cytometry-reagents/research-reagents/single-color-antibodies-ruo/BV650-Rat-Anti-Mouse-CD41.740504?tab=product_details)
APC anti-mouse CD127 (IL-7Ra) Antibody (BioLegend; 135012), reported to recognize mouse IL7Ra (https://www.biolegend.com/en-us/products/apc-anti-mouse-cd127-il-7ralpha-antibody-6191)
anti-mouse FLT3 PerCP-eFluor710 (eBioscience; 46-1351-82), reported to recognize mouse FLT3 (https://www.thermofisher.com/antibody/product/CD135-Flt3-Antibody-clone-A2F10-Monoclonal/46-1351-82)
PE/Cyanine7 anti-mouse CD150 (SLAM) Antibody BioLegend; 115914), reported to recognize mouse CD150 (https://www.biolegend.com/en-us/products/pe-cyanine7-anti-mouse-cd150-slam-antibody-3056)
TruStain FcX (anti-mouse CD16/32) Antibody (Biolegend; 101320), reported to recognize mouse CD16/32 (https://www.biolegend.com/en-us/products/trustain-fcx-anti-mouse-cd16-32-antibody-5683)
Invitrogen eBioscience Fixable Viability Dye eFluor 780 (Invitrogen; 65-0865-14), reported to recognize live cells from dead cells (https://www.fishersci.de/shop/products/fixable-viability-dye-efluor-780-1/13539140?srsltid=AfmBOopvFHFehhCHI9VFYBg8JDn94rU93NlIL6IKEPFjlz-I1kTm4P18)

SIINFEKL-H-2 Kb – PE, validated internally by isolating T cells from OT-I mouse as staining control
SIINFEKL-H-2 Kb – APC, validated internally by isolating T cells from OT-I mouse as staining control
AAHAEINEA-I-A b – PE, validated internally by isolating T cells from OT-II mouse as staining control
AAHAEINEA-I-A b – PE, validated internally by isolating T cells from OT-II mouse as staining control
KYNKANAFL-H-2 Kd – PE, validated internally by isolating T cells from NY8.3 mouse as staining control
KYNKANAFL-H-2 Kd – APC, validated internally by isolating T cells from NY8.3 mouse as staining control

# Eukaryotic cell lines

Policy information about cell lines and Sex and Gender in Research

| | |
|---|---|
| Cell line source(s) | OVA-expressing melanoma B16 (B16-OVA) and MC38 (MC38-OVA) cell lines were kindly provided by Michael Kilian (Harvard Medical School, Boston, USA). |
| Authentication | None of the cell lines used were authenticated by our laboratory. |
| Mycoplasma contamination | The cell lines were not tested for mycoplasma contamination. |
| Commonly misidentified lines (See ICLAC register) | Not applicable. |

# Animals and other research organisms

Policy information about studies involving animals; ARRIVE guidelines recommended for reporting animal research, and Sex and Gender in Research

Laboratory animals

Wild-type C57BL/6J mice
For experiments in Figures 1-5 and associated extended data and supplementary figures: female and male C57BL/6J mice (Jackson Labs, strain #000664).
For experiments in Figures 1B-D and associated extended data and supplementary figures: age between 6 and 100 weeks (longitudinal experiment).
For experiments in Figure 2A and associated extended data and supplementary figures: age between 6 and 90 weeks (longitudinal experiment).
For experiments in Figures 2B,C,D,E,G,H,I,J,K,L and associated extended data and supplementary figures: age 6 weeks (adult) or 72 weeks (aged).
For experiments in Figures 3B,C,D,E;F,G,H and associated extended data and supplementary figures: age 6 weeks (adult) or 72 weeks (aged).
For experiments in Figure 3I and associated extended data and supplementary figures: age between 6 and 90 weeks (longitudinal experiment).
For experiments in Figures 4B,C,D,F,G,H,J,K,L,M,N,O,P,Q,R and associated extended data and supplementary figures: age 72 weeks (aged).
For all other experiments: age 6-12 weeks.

Transgenic mice
For experiments in Figure 2 and associated extended data figures: female and male C57BL/6-Tg(Nr4a1-EGFP/cre)820Khog/J (Nur77-GFP) mice (Jackson Labs, strain #016617; 6-12 weeks of age), FVB-Tg(Rag2-EGFP)1Mnz/J (Rag2-EGFP) mice (Jackson Labs, strain #005688; 6-12 weeks of age).
For experiments in Figure 5 and associated extended data figures: female NOD mice (Jackson Labs, strain #000664; 6 weeks of age at experiment start); female and male C57BL/6-Tg(CAG-OVAL)916Jen/J (Act-mOVA) mice (Jackson Labs, strain #005145; 6-12 weeks of age), female and male C57BL/6-Tg(TcraTcrb)1100Mjb/J (OT-I) mice (Jackson Labs, strain #003831; 6-12 weeks of age) female and male B6.Cg-Tg(TcraTcrb)425Cbn/J (OT-II) mice (Jackson Labs, strain #004194; 6-12 weeks of age).

Animals were kept on a 12-h light/dark cycle between 68°F and 79°F and 30–70% humidity. Mice were acclimated at the animal

facility for at least 7 days before performing any experiments.

| | |
|---|---|
| Wild animals | No wild animals were used in this study. |
| Reporting on sex | Except where noted and experimentally warranted (NOD and EAE experiments), all experiments were conducted with sex-matched animals, without bias to either sex. Sex-based analysis was not performed. |
| Field-collected samples | No field-collected samples were used in this study. |
| Ethics oversight | All animal experiments were approved by the Institutional Animal Care and Use Committee (IACUC) of the Broad Institute (Protocol ID 0017-09-14-2). Animal maintenance complied with all relevant ethical regulations and were consistent with local, state and federal regulations as applicable, including the National Institutes of Health Guide for the Care and Use of Laboratory Animals. |

Note that full information on the approval of the study protocol must also be provided in the manuscript.

## Plants

| | |
|---|---|
| Seed stocks | No plants were used in this study. |
| Novel plant genotypes | No plants were used in this study. |
| Authentication | No plants were used in this study. |

## Flow Cytometry

### Plots

Confirm that:

☒ The axis labels state the marker and fluorochrome used (e.g. CD4-FITC).

☒ The axis scales are clearly visible. Include numbers along axes only for bottom left plot of group (a 'group' is an analysis of identical markers).

☒ All plots are contour plots with outliers or pseudocolor plots.

☒ A numerical value for number of cells or percentage (with statistics) is provided.

### Methodology

| | |
|---|---|
| Sample preparation | Cells were prepared and stained according to the staining protocol of each experiment outlined in the methods section, pelleted at 500 g for 5min, and resuspended in 200 μL of flow cytometry buffer (PBS supplemented with 2% EDTA (Life Technologies 15575020) and 5% FBS (VWR 97068-085)). All antibodies and tetramers used can be found in the Reporting Summary. |
| Instrument | Samples were run for flow cytometry analysis on a Beckman Coulter Cytoflex LX flow cytometer. For FACS, cells were run either on a |
| Software | Analysis was performed using the FlowJo v10 software. |
| Cell population abundance | Cell population abundances are provided in the figures depicting gating strategies. For single-cell RNA-seq of purified T cells, post-sort purity was approximately 90%. |
| Gating strategy | FACS gating strategies for all analyses are supplied in Extended Data Figures and Supplementary Information. Population definitions are stated in the manuscript main text or methods. For analysis of non-lineage antibodies, the gating was determined by florescence-minus-one (FMO). For antigen-specific cell analysis, cells from naive mice, unvaccinated mice, Act-mOVA mice, OT-I or OT-II mice, were used as negative and positive controls for gating strategy, respectively. |

☒ Tick this box to confirm that a figure exemplifying the gating strategy is provided in the Supplementary Information.

