## [Peer Review File · Nature]

Transient hepatic reconstitution of trophic factors enhances aged immunity

Corresponding Author: Professor Feng Zhang

Version 1:

Reviewer comments:

Referee #1

(Remarks to the Author)

Transient hepatic reconstitution of DLL1, FLT3-L, and IL-7 enhances immune function in aged mice", by Professor Zhang and colleagues (manuscript number 2024-10-20824A),

1. Key results. Please summarize what you consider to be the outstanding features of the work.

Friedrich et al have presented a compelling body of work with potential implications for enhancing vaccine efficacy and advancing cancer immunotherapy in aging populations. Although the biological effects are currently demonstrated only in mouse models, the use of lipid nanoparticle (LNP) technology for mRNA delivery to produce proteins in the liver is a well-established platform that could accelerate the clinical translation of their findings.

The authors present an innovative approach that combines scRNASeq and spatial transcriptomics to identify key ligand-receptor interactions in the thymus of young and aged mice and correlate these findings with genetic pathways in circulating T cells that change with aging. They focused on three signaling pathways deficient in aged mice (Notch1, Flt3, and IL7R) by using lipid nanoparticle (LNP) mRNA to deliver exogenous ligands for production in liver that activate these receptors. The authors then assessed whether this approach could rejuvenate T cell responses in aged mice.

The three-ligand LNP mRNA treatment (designated DFI), successfully reduced aging- and exhaustion-associated T cell phenotypes, shifting the T cell profile in aged mice to resemble that of younger mice, with an increase in naïve T cells. The treatment also promoted the expansion of splenic mature B cells, reduced age-associated B cells, and increased the number of splenic conventional dendritic cells type 1 (cDC1) without affecting cDC2.

Furthermore, DFI LNP treatment enhanced the response to vaccination with a foreign antigen (OVA), specifically boosting CD8+ T cell responses in aged mice to levels closer to those in younger adults. In a B16-OVA tumor model, the authors demonstrated enhanced tumor antigen immunity when DFI LNP was co-administered with immune checkpoint inhibitors (ICIs). While DFI LNP showed some activity as a prophylactic treatment followed by ICI, its impact was stronger when delivered alongside ICIs.

Importantly, DFI LNP treatment did not promote autoimmunity in a non-obese diabetic (NOD) model, nor did it break tolerance in a germline OVA-expressing mouse model. Notably, while IL-7 mRNA LNP delivery alone could expand T cells, the authors found that the DFI combination was superior for boosting naïve T cell populations and CD8+ T cell recall responses to OVA in aged mice.

2. Validity. Does the manuscript have flaws that should prohibit its publication? If so, please provide details.

The study is rigorous and comprehensive, with no critical flaws that would hinder publication. However, I have included questions for the authors to address and recommend providing additional information that could further enhance the

implications of their findings.

3. Originality and significance. If the conclusions are not original, please provide relevant references. On a more subjective note, do you feel that the results presented are of immediate interest to many people in your own discipline, or to people from several disciplines?

Conclusion, approach and findings are original and would have broad interest in understanding of vaccine response in aged and cancer immunotherapy.

4. Data & methodology. Is the approach valid? Are the data and presentation of good quality? Please note that we expect our reviewers to review all data, including the Supplementary Information.

The data and presentation are high quality. The presentation of main figures supports the major findings, and the extended data provides additional context and details that further support the main data presented.

5. Appropriate use of statistics and treatment of uncertainties (if applicable). All error bars should be defined in the corresponding figure legends. Please include in your report a specific comment on the appropriateness of any statistical tests, and the accuracy of the description of any error bars and probability values.

Statistical analysis was applied appropriately throughout the manuscript.

6. Conclusions. Are the conclusions and data interpretation robust, valid, appropriate and reliable?

The authors conclusions are well supported by robust and appropriately interpreted data.

7. Suggested improvements. Please list additional experiments or data that could help strengthen the work in a revision.

A. Major Points:

The evidence that liver produced (and liver bound) DLL1 is contributing to the observed DFI LNP treatments is one of the weak points of otherwise well supported conclusions in the manuscript. The contribution of liver-bound DLL1 in the DFI treatment is important for understanding the impact of liver-restricted expression of surface bound ligands to (partially) rejuvenate the thymic niche.

1. The authors claim that the three-ligand combination of DLL1, Flt3L and IL7 (DFI) are necessary to induce the enhanced immune response (primarily CD8+ T cell) they observe in treated aged mice. The evidence they provide is based on single-factor response experiments compared to negative (Luc LNP) and presumptive-positive (DFI LNP) controls (Fig 2 B-D; Fig 3 C-D; Fig ED5 B-E & Fig ED8G). However, the authors do not provide evidence that any two factors are not sufficient, particularly IL7 + Flt3L. While the mechanism of the impact of systemic distribution of Flt3L and IL7 is more easily explained, the contribution of DLL1 restricted to cell surface of hepatocytes and interaction with developing T cells is less clear (although I appreciate the explanation of the liver ECM-associated IL7 as a possible reasoning Lines 129-131). Since the authors see increased thymocytes in thymi of aged mice following DFI LNP, particularly the absolute number of DN1-3 (Fig 6ED C-D), is it possible that endogenous Notch1 ligands in the cTEC in DFI-treated aged mice are providing sufficient Notch1 signalling for developing T cell progenitors (TSP/ETP and especially DN) that are expanded by IL7/Flt3L (albeit more limited in involuted thymi of aged mice)? Along these lines, the expansion (and activation) of cDC1 by Flt3L would be expected to contribute to the CD8+ T cell recall. Taken together, is it possible that Flt3L and IL7 together are sufficient for most of the observed effect size? If not, can the authors demonstrate the impact of Dll1 in the DFI LNP treatment by other means? For example, would IL7 and Flt3L mRNA LNP treatment of aged mice increase the Notch1 signalling in the thymus (as in Fig 1B)?

2. The authors show the kinetics of Flt3L and IL7 in vivo translated protein expression after DFI LNP treatment (Fig 1 H-I). But they do not show how long the DLL1 protein surface expression in the liver is maintained in vivo post DFI LNP delivery. Evidence for translation and expression appears to be limited to in vitro primary hepatocytes and in vivo RIBOmap. This evidence confirms that mRNA for Dll1 is likely being translated in the liver in vivo but does not confirm in vivo surface expression of the ligand. I realize that there is no easy recombinant cytokine comparator (like for FLT3L and IL7) but demonstration of in vivo DLL1 protein expression in the liver is lacking throughout. This is important since the authors show that DLL1 mRNA alone does not appear to have any (or much) biological impact. Encouraging exceptions are in Fig 2C&D; Fig ED5 C & E that show small (but statistically significant) impact of Dll1 alone.

B. Minor Points:

1. It is interesting that some of the thymic chemokine signaling appears to increase with age (Fig 3ED D) – particularly CCL25. Do they authors see changes in the number or frequency of lymphoid progenitor (LP) populations in mouse bone

marrow (BM) with DFI LNP treatment (especially CCR9+ LPs)? In other words, does the DFI treatment increase the lymphoid progenitor pool in the BM – especially those progenitors that retain thymocyte potential? In general, an analysis of how the DFI treatment impacts BM and blood hematopoietic populations may prove to be informative.

2. The analysis of the 'aged' improved vaccine response in Fig 3 would benefit from a recall response from draining lymph nodes and/or spleen to determine CD4 Th1/Th2 proclivity and an analysis of OVA-specific immunoglobulin isotypes in circulation.

3. Can the authors provide a rationale for the choice of DLL1 (vs DLL4) for the reconstitution of Notch signalling? Was the decision informed by the Slide-Seq cTEC spatial interactions or an empirical choice?

4. Have the authors determined the longevity of the DFI benefit post treatment? After the DFI LNP treatment regimen – do the mice maintain a rejuvenated naïve T cell phenotype, splenic cDC1/B cell phenotype? CD8+ response to OVA immunization?

5. FIG1ED E & H are a bit hard to interpret. I understand that the grey bars correspond to populations in the UMAP that do not change significantly compared to 4-week-old mice, but it is not immediately obvious which grey bars correspond to which populations. I surmise they are in the same order from L to R /top to bottom as the UMAP populations in Fig1ED C & F, respectively. Perhaps the authors could label the populations 1-7 for 1C and 1-5 for 1F. Alternatively, perhaps exclude the not significant grey bars in Figure 1ED E & H and focus on the significant changes only?

6. LINE 44-45 – define DLL1, FLT3-L, and IL-7 in first appearance

7. Fig 3ED F – typo - 'sample' is misspelled in figure title

8. Fig 3ED L – The authors speculated that it is the LNP responsibly for slightly elevated circulating IL-6 in vivo (Lines 134-146). A better control that may answer this definitively would be Luc mRNA (or other RNA) LNP rather than NaCl/saline – as they have done for other experiments.

9. Fig 2G – CD83? – CD80 is described in the text (Line 185). I'm not sure if the Figure or text is correct?

10. Fig 7ED C–CD83 in figure – it is not clear which one the authors meant given previous point.

11. Line 227 – call out should be Fig 3H (not 3F)

12. Line 231 – call out should be Fig 3I (not 3G)

8. References. Does the manuscript reference previous literature appropriately?

The manuscript is well-cited and provide appropriate background context to position their work within current literature.

9. Clarity and context. Is the abstract clear and accessible? Are the abstract and introduction appropriate?

The abstract and introduction are well crafted to introduce the paper.

10. Please indicate any particular aspect of the manuscript, data or analyses that you feel is outside the scope of your expertise, or that you were unable to assess fully.

I did not re-analyse the raw data files provided.

Referee #2

(Remarks to the Author)

In this manuscript, Friedrich et al. applying several cutting-edge omics to thymus samples from aged mice identify that IL7 and Notch signaling are linked to immune system aging. By hydrodynamic delivery of lipid nanoparticles containing the mRNA for three selected factors related to these pathways and FLT-3L (based on previous reports) they achieve a transitory production of these three factors in the liver. Notably, this is sufficient to rejuvenate immune function in aged mice, leading to improved responses to vaccination and cancer immunotherapy without any observed side effect. Although the factors that the authors use to reconstitute immune function in aged mice were already established for thymus repair and regeneration, the novelty of this paper resides in the technology based on lipid nanoparticles containing mRNA as a delivering method to the liver is a potential strategy to improve immune function deterioration with aging. This paper supposes a breakthrough in the immune aging field and opens exciting opportunities to overcome immune cell dysfunction during aging.

Major comments:

- With age, the TCR repertoire diminishes, which compromises the ability of T cells to mount an effective immune response.

To really understand the immune reconstitution potential of this strategy, it is essential to know if TCR repertoire diversity is restored upon the treatment or still remains reduced. Moreover, the authors claim that the treatment is enhancing T cell development in the thymus. To really demonstrate this, it will be required to analyze TRECs, as a readout of T cell development and TCR recombination.

- Is the rejuvenation of the immune system attributed to a re-organization/formation of the thymus? Are LNPs able to rejuvenate T cells in the absence of the thymus? Considering FLT-3 is a well-known regulator of hematopoiesis, I am not entirely convinced that they are not improving immune system function just by rebalancing the typical myeloid-biased of hematopoiesis that occurs during aging. In other words, how are the levels of hematopoietic precursors (HSCs, GMPs, CMPs and CLPs) in aged mice treated with LNPs? How are the levels of monocytes and neutrophils in the blood and bone marrow of these mice?

-How does old treated mice response to a viral infection challenge?

- Have the authors tested how long these effects last after stopping the administration of LNPs? How long these factors keep upregulated in the blood after stopping LNP delivery? Related to this, have they tested whether extending this treatment (maybe with some intermediate resting periods) could exert more potent responses or, opposite, would lead to a loosening of the observed effects?

-To firmly demonstrate that self-tolerance is not compromised, it will be ideal to add an additional model of autoimmunity avoiding the use of TCR transgenic mouse models (like NOD), such as EAE.

- Most cancer models used, while valid for the tested hypothesis, are somehow not very representative of the real scenario and incorporate the co-administration of immunotherapy. I would be curious of whether LNPs could also affect the endogenous response and the cancer outcome in any model of typically immunogenic cancer without the co-administration of ICI.

Minor comments

- The abstract states that "Aging leads to a progressive decline in immune function, marked by a reduced t cell diversity...". While TCR diversity is clearly reduced during aging, I do not think stating that T cells are less diverse in aged mice is correct. Indeed, recent findings suggest that aging is accompanied by the appearance of several new T cell subsets (PMID: 31457092, PMID: 33271118). Therefore, I would recommend editing this particular statement.

- There are a couple of mistakes in the text regarding some figures: line 115 should be Fig 1F instead of 1E, line 117 should be Fig 1F-G instead of Fig 1E-F, line 227 should be Fig. 3H instead of 3F and line 231 should be Fig. 3I instead of 3G

Referee #3

(Remarks to the Author)

Recommendations, major revisions.

Review:

T cells are known to be essential to maintain adaptive immunity. Treatments for lymphopenia are lacking for aged individuals or patients undergoing cytoreductive therapies. In this study the authors propose a treatment with mRNAs with known lymphopoietic factors to improve T cell production.

Major comments:

- 1) The authors claim the delivery of LNPs of well-known thymopoietic factors affects immune function partially through rejuvenation of T cell production and claim this is by boosting thymic function.
- 2) Although the manuscript has multiple orthogonal ways to show immune function the manuscript does describe well known thymopoietic factors. The manuscript would benefit for a better mechanistic understanding of the effect on thymic biology, transient vs long-lasting, bone marrow phenotype, CLP, ETPs etc. vs stromal cells.
- 3) Is the effect dependent on functional thymic tissue or is there extrathymic T cell production? Is the same phenomenon observed in FOXP1- mice?
- 4) Intrathymic function. The authors could do this through a) flow cytometry of thymic epithelial cells, 2) other supporting stroma, 3) FOXP1 expression in cTECs and/or mTECs, what is the effect on $\alpha\beta$ vs. $\gamma\delta$ T cells? Intrathymic DCs?
- 5) Thymic output: The increase in CD44⁺ CD62L⁺ naïve T cells in the spleen, could be due to peripheral expansion and/or thymic production. Are TRECs increased? What happens in RAG2GFP mice? Nur77GFP or other models to assess recent thymic emigrants? What about TCR repertoire? Is IL7 leading to peripheral expansion or de novo seeding of T cells? Is the exhaustion profile altered in the LNP treated group in B16 melanoma?
- 6) If the treatment leads to more Thymic function, is it a lasting effect? Is the thymus still bigger after discontinuation? What happens after stopping treatment? Is there a sustained benefit?
- 7) Is the tumor survival due to a larger T cell repertoire? Or higher number of T cells? Hard to assess since the number of T cells in the model are equal. T cell diversity assessment would be helpful.

8) In figure 5, OT mice are used where OVA is expressed as a self-antigen, it would be unexpected for these mice to have a break in tolerance unless the treatment disrupted central or peripheral tolerance mechanisms. Partial AIRE deficiency models or other models of tolerance and medullary thymic function should be used instead or some references to the use of breaking of tolerance in OT mice provided.

Minor comments:

- 1) Is there off-target delivery? Is the treatment specific to the liver? The authors might have a sentence about that, the overemphasis on the liver is nice but isn't the delivery mainly going to the liver because it's the organ which is first and foremost targeted by the LNPs.
- 2) NASH in the liver is associated with Notch H&E staining's at least of normal liver tissue after LNP is preferable.
- 3) Why is there an increase in IL6 specifically? Is that due to Liver injury with the LNP?
- 4) Figure 5B: There is no increase in circulating autoreactive T cell, what is the effect of the treatment of LNPs. Is the medullary Thymic epithelial cell function also ameliorated by the treatment?
- 5) There is an extensive amount of computational data being presented in supplementary figures. This may be nice but the factors used have been widely studied, the authors should emphasize these previous publications, e.g. of IL7 in both mouse and men.

Version 2:

Reviewer comments:

Referee #1

(Remarks to the Author)

The authors have undertaken extensive additional analyses employing complementary models and orthogonal approaches that not only robustly validate their original conclusions but also meaningfully extend the scope of the initial findings. Collectively, the original and newly generated data provide valuable mechanistic insights that are likely to be of strong interest to Nature's readership and have important implications for the field of immune aging, including potential clinical applications. I was enthusiastic about the initial submission, and I believe the authors' comprehensive and well-reasoned rebuttal has further strengthened the clarity and impact of the work.

The statistical analyses appear to be consistent and appropriate throughout the manuscript and in line with the journal's requirements.

Referee #2

(Remarks to the Author)

The authors have made a great effort to clarify the mechanisms by which DFI works and have added significant new experiments supporting the beneficial effect of the treatment in different contexts while discarding potential undesirable harmful effects of the treatment. The only concern I have is that in the title and throughout the text these three factors DFI are referred to as thymic factors. While it is true that they are expressed there, some of their effects are also extrathymic, specially for FLT3L which may lead to confusion. I would therefore suggest going back to the original version of the title and softening the statement in order to avoid any potential misunderstanding.

Referee #3

(Remarks to the Author)

This is a strong, well-executed study with clear novelty and solid methodological rigor. The authors' revisions and responses were adequate with a considerable amount of additional data. I have no outstanding substantive concerns. I recommend acceptance of this manuscript.

Response to Reviews for 2024-10-20824A

Transient hepatic reconstitution of DLL1, FLT3-L, and IL-7 enhances immune function in aged mice

Friedrich *et al.*

We thank the Reviewers for their time and constructive feedback. Through the Reviewers' guidance, we believe we have substantially strengthened the mechanistic foundation of our findings and addressed key concerns. We expanded our analyses of thymic biology, bone marrow progenitor dynamics, and peripheral immune remodeling by integrating additional functional assays and multi-omics datasets. The new data show that DFI treatment amplifies and conditions common lymphoid progenitors through systemic FLT3-L and hepatocyte-bound DLL1 and IL-7, inducing a transient but potent burst of thymopoiesis without altering thymic stroma or compromising tolerance. We further delineated the temporal dynamics of these effects, demonstrating durable remodeling of splenic T, B, and dendritic cell compartments, but a strictly time-limited improvement of thymic output and vaccine responsiveness, consistent with the short-lived expression of the delivered mRNAs.

We also incorporated additional autoimmunity and tolerance testing (EAE), extensive safety profiling, and multiple repertoire analyses (TRECs, Rag2/Nur77 reporters, single-cell TCR profiling) to further substantiate that DFI enhances immune competence without inducing autoimmunity, hepatotoxicity, or clonal skewing. Finally, we added data demonstrating that DFI improves humoral immunity upon vaccination and enhances endogenous anti-tumor immunity in a slow-growing tumor model without immune checkpoint inhibition as a standalone preventive regimen.

Collectively, these revisions clarify DFI's mechanism as a controllable and temporally tunable strategy to rejuvenate adaptive immunity in aged hosts, with translational potential for vaccination and cancer immunotherapy. Below we expand on these changes point-by-point.

Referee #1 (Remarks to the Author):

Transient hepatic reconstitution of DLL1, FLT3-L, and IL-7 enhances immune function in aged mice", by Professor Zhang and colleagues (manuscript number 2024-10-20824A),

1. Key results. Please summarize what you consider to be the outstanding features of the work.

Friedrich et al have presented a compelling body of work with potential implications for enhancing vaccine efficacy and advancing cancer immunotherapy in aging populations. Although the biological effects are currently demonstrated only in mouse models, the use of lipid nanoparticle (LNP) technology for mRNA delivery to produce proteins in the liver is a well-established platform that could accelerate the clinical translation of their findings.

The authors present an innovative approach that combines scRNASeq and spatial transcriptomics to identify key ligand-receptor interactions in the thymus of young and aged mice and correlate these

findings with genetic pathways in circulating T cells that change with aging. They focused on three signaling pathways deficient in aged mice (Notch1, Flt3, and IL7R) by using lipid nanoparticle (LNP) mRNA to deliver exogenous ligands for production in liver that activate these receptors. The authors then assessed whether this approach could rejuvenate T cell responses in aged mice.

The three-ligand LNP mRNA treatment (designated DFI), successfully reduced aging- and exhaustion-associated T cell phenotypes, shifting the T cell profile in aged mice to resemble that of younger mice, with an increase in naïve T cells. The treatment also promoted the expansion of splenic mature B cells, reduced age-associated B cells, and increased the number of splenic conventional dendritic cells type 1 (cDC1) without affecting cDC2.

Furthermore, DFI LNP treatment enhanced the response to vaccination with a foreign antigen (OVA), specifically boosting CD8⁺ T cell responses in aged mice to levels closer to those in younger adults. In a B16-OVA tumor model, the authors demonstrated enhanced tumor antigen immunity when DFI LNP was co-administered with immune checkpoint inhibitors (ICIs). While DFI LNP showed some activity as a prophylactic treatment followed by ICI, its impact was stronger when delivered alongside ICIs.

Importantly, DFI LNP treatment did not promote autoimmunity in a non-obese diabetic (NOD) model, nor did it break tolerance in a germline OVA-expressing mouse model. Notably, while IL-7 mRNA LNP delivery alone could expand T cells, the authors found that the DFI combination was superior for boosting naïve T cell populations and CD8⁺ T cell recall responses to OVA in aged mice.

2. Validity. Does the manuscript have flaws that should prohibit its publication? If so, please provide details.

The study is rigorous and comprehensive, with no critical flaws that would hinder publication. However, I have included questions for the authors to address and recommend providing additional information that could further enhance the implications of their findings.

3. Originality and significance. If the conclusions are not original, please provide relevant references. On a more subjective note, do you feel that the results presented are of immediate interest to many people in your own discipline, or to people from several disciplines?

Conclusion, approach and findings are original and would have broad interest in understanding of vaccine response in aged and cancer immunotherapy.

4. Data & methodology. Is the approach valid? Are the data and presentation of good quality? Please note that we expect our reviewers to review all data, including the Supplementary Information.

The data and presentation are high quality. The presentation of main figures supports the major findings, and the extended data provides additional context and details that further support the main data presented.

5. Appropriate use of statistics and treatment of uncertainties (if applicable). All error bars should be defined in the corresponding figure legends. Please include in your report a specific comment on the

appropriateness of any statistical tests, and the accuracy of the description of any error bars and probability values.

Statistical analysis was applied appropriately throughout the manuscript.

6. Conclusions. Are the conclusions and data interpretation robust, valid, appropriate and reliable?

The authors conclusions are well supported by robust and appropriately interpreted data.

7. Suggested improvements. Please list additional experiments or data that could help strengthen the work in a revision.

A. Major Points:

The evidence that liver produced (and liver bound) DLL1 is contributing to the observed DFI LNP treatments is one of the weak points of otherwise well supported conclusions in the manuscript. The contribution of liver-bound DLL1 in the DFI treatment is important for understanding the impact of liver-restricted expression of surface bound ligands to (partially) rejuvenate the thymic niche.

1. The authors claim that the three-ligand combination of DLL1, Flt3L and IL7 (DFI) are necessary to induce the enhanced immune response (primarily CD8+ T cell) they observe in treated aged mice. The evidence they provide is based on single-factor response experiments compared to negative (Luc LNP) and presumptive-positive (DFI LNP) controls (Fig 2 B-D; Fig 3 C-D; Fig ED5 B-E & Fig ED8G). However, the authors do not provide evidence that any two factors are not sufficient, particularly IL7 + Flt3L. While the mechanism of the impact of systemic distribution of Flt3L and IL7 is more easily explained, the contribution of DLL1 restricted to cell surface of hepatocytes and interaction with developing T cells is less clear (although I appreciate the explanation of the liver ECM-associated IL7 as a possible reasoning Lines 129-131). Since the authors see increased thymocytes in thymi of aged mice following DFI LNP, particularly the absolute number of DN1-3 (Fig 6ED C-D), is it possible that endogenous Notch1 ligands in the cTEC in DFI-treated aged mice are providing sufficient Notch1 signalling for developing T cell progenitors (TSP/ETP and especially DN) that are expanded by IL7/Flt3L (albeit more limited in involuted thymi of aged mice)? Along these lines, the expansion (and activation) of cDC1 by Flt3L would be expected to contribute to the CD8+ T cell recall. Taken together, is it possible that Flt3L and IL7 together are sufficient for most of the observed effect size? If not, can the authors demonstrate the impact of Dll1 in the DFI LNP treatment by other means? For example, would IL7 and Flt3L mRNA LNP treatment of aged mice increase the Notch1 signalling in the thymus (as in Fig 1B)?

We thank the reviewer for raising this critical point regarding the contribution of DLL1 to the effects of DFI. To clarify the necessity of DLL1, we performed a set of additional experiments addressing its expression, function, and non-redundant role relative to IL-7 and FLT3-L.

First, to confirm *in vivo* production of functional DLL1, we combined immunofluorescence staining with STARmap spatial transcriptomics to track its localization in the liver sinusoids over time (0-48 hours post-delivery). We observed robust, time-dependent surface expression of DLL1 on hepatocytes, peaking at 6-12 hours and persisting for up to 48 hours (**Fig. 1F-H, Extended Data Fig.**

4F). These findings establish that DLL1 is translated, trafficked, and presented on hepatocytes at sufficient density to engage Notch receptors on sinusoid-trafficking immune progenitors.

[Redacted text]

In vivo, DFI treatment also increased CCR9 expression on CLPs as they transitioned from bone marrow to thymus (**Extended Data Fig. 7F**). This pattern, typical of young but not untreated aged mice, suggests that hepatocyte-bound DLL1 delivers a peripheral Notch signal that may enhance thymus homing or T lineage bias of circulating progenitors. While indirect, these findings align with DLL1's established role in conditioning progenitors prior to intrathymic differentiation.

Third, we directly compared DFI to a control combination in which DLL1 was replaced by luciferase (LFI) in parallel vaccination and recall studies (**Point-by-point Fig. 2A**). Only DFI-treated aged mice showed consistent restoration specifically of thymic mass (**Point-by-point Fig. 2B-C**), expansion of antigen-specific CD8⁺ T cells (**Point-by-point Fig. 2D**), robust IFN- γ and TNF production upon recall (**Point-by-point Fig. 2E**), and enhanced OVA-specific IgG responses (**Point-by-point Fig. 2F**). LFI failed to reproduce these benefits despite comparable expression of IL-7 and FLT3-L.

Finally, as suggested by the Reviewer, we evaluated whether FI indirectly augments Notch activity in the thymus by stimulating endogenous ligand expression. Slide-seq analysis of thymic tissue after DFI treatment revealed increased cellularity and thymocyte content, consistent with improved progenitor recruitment or survival, but no significant increase in endogenous Notch ligands in thymic epithelial cells (**Point-by-point Fig. 3F-H**). Thus, IL-7 and FLT3-L do not restore Notch signaling via endogenous mechanisms, although residual DLL1/DLL4 in aged cTECs appears sufficient to support differentiation once CLP influx is enhanced by DFI (as shown in **Extended Data Fig. 7D; Fig. 2F-G**).

In sum, our data indicate that DLL1 provides a non-redundant Notch signal within the DFI regimen. The combined activity of IL-7 and FLT3-L may support CLP expansion and survival but is insufficient to recapitulate the full spectrum of immunological benefits spanning thymocyte development, peripheral T cell output, and vaccine responses without DLL1. We therefore believe that hepatocyte-displayed DLL1 represents a means of delivering spatially restricted Notch activation that cannot be achieved through soluble cytokines alone.

2. The authors show the kinetics of Flt3L and IL7 *in vivo* translated protein expression after DFI LNP treatment (Fig 1 H-I). But they do not show how long the DLL1 protein surface expression in the liver is maintained *in vivo* post DFI LNP delivery. Evidence for translation and expression appears to be limited to *in vitro* primary hepatocytes and *in vivo* RIBOmap. This evidence confirms that mRNA for Dll1 is

likely being translated in the liver *in vivo* but does not confirm *in vivo* surface expression of the ligand. I realize that there is no easy recombinant cytokine comparator (like for FLT3L and IL7) but demonstration of *in vivo* DLL1 protein expression in the liver is lacking throughout. This is important since the authors show that DLL1 mRNA alone does not appear to have any (or much) biological impact. Encouraging exceptions are in Fig 2C&D; Fig ED5 C & E that show small (but statistically significant) impact of Dll1 alone.

We appreciate the reviewer's thoughtful comment highlighting the need for direct *in vivo* confirmation of DLL1 surface expression following LNP-mRNA delivery. As noted above, we have now performed a series of experiments combining STARmap spatial transcriptomics with immunofluorescence (IF) staining of DLL1 and F-actin staining via Phalloidin of liver sections at multiple time points post-DFI LNP administration (0h, 6h, 12h, 24h, and 48h), now shown in **Fig. 1F-H and Extended Data Fig. 4D-F**. These data provide time-resolved evidence that Dll1 mRNA is robustly induced in hepatocytes after DFI-LNP treatment, peaking at 6-12 hours and remaining elevated up to 48 hours and, importantly, DLL1 protein is detectable on the surface of hepatocytes *in vivo*, with a distribution and temporal pattern that closely mirrors the mRNA kinetics. Interestingly, we also observed baseline DLL1 expression in hepatocytes of untreated mice, consistent with prior findings suggesting that the liver may possess intrinsic Notch-modulating capacity under homeostatic conditions (Matsumoto et al., *Hepatology* 2001; Hu et al., *JCI Insight* 2018). While the presence of endogenous DLL1 may contribute modestly to basal Notch signaling, our quantitative image analysis shows that DFI treatment significantly increases DLL1 surface density. The transient but substantial overexpression achieved by mRNA delivery likely creates a non-physiological density of DLL1 ligands sufficient to elicit Notch activation in circulating or sinusoid-trafficking immune cells, an effect that cannot be achieved with protein-based delivery due to DLL1's membrane-bound nature. Moreover, we note that despite baseline DLL1 presence, FI alone (IL-7 + FLT3L) fails to fully recapitulate the immunological improvements observed with the full DFI combination, as shown in our comparative recall and vaccination studies above. This underscores that endogenous DLL1 on hepatocytes is insufficient to compensate for the absence of exogenous DLL1 when delivering only FI.

B. Minor Points:

1. It is interesting that some of the thymic chemokine signaling appears to increase with age (Fig 3ED D) – particularly CCL25. Do they authors see changes in the number or frequency of lymphoid progenitor (LP) populations in mouse bone marrow (BM) with DFI LNP treatment (especially CCR9+ LPs)? In other words, does the DFI treatment increase the lymphoid progenitor pool in the BM – especially those progenitors that retain thymocyte potential? In general, an analysis of how the DFI treatment impacts BM and blood hematopoietic populations may prove to be informative.

We thank the reviewer for this insightful suggestion, which prompted us to expand our analysis of hematopoietic populations in bone marrow and blood. These additional experiments have proven pivotal for clarifying how DFI enhances thymic and peripheral immune compartments.

We find that DFI's primary effect is at the level of committed lymphoid progenitors rather than hematopoietic stem cells (HSCs). Consistent with prior reports, total HSCs ($\text{Lin}^- \text{c-Kit}^+ \text{Sca-1}^+ \text{FLT3}^- \text{CD34}^- \text{CD150}^+$) increase with age as part of a compensatory but functionally impaired expansion,

and aged bone marrow shifts toward myeloid-biased HSCs at the expense of balanced HSCs (**Extended Data Fig. 8A-D**). DFI did not reverse this HSC compositional bias or significantly alter multipotent progenitor frequencies, apart from a modest increase in the lymphoid-primed MPPc subset (**Extended Data Fig. 8B**). Thus, unlike strategies such as antibody-mediated my-HSC depletion (Ross et al., *Nature* 2024), DFI does not act by remodeling the stem cell tier.

Instead, DFI exerts its dominant effect downstream, markedly expanding common lymphoid progenitors (CLPs) in the bone marrow and peripheral blood (**Extended Data Fig. 7C and 8F**). These CLPs, which typically decline sharply with age (Miller and Allman, *Nat Rev Immunol* 2003; Rossi et al., *Cell Stem Cell* 2005), approached adult levels in DFI-treated mice. Indeed, as hypothesized by the Reviewer, DFI selectively increased the fraction of CCR9⁺ CLPs, which are biased toward T lineage development and possess enhanced thymus-homing potential (**Extended Data Fig. 7D-G**). We observed that the expression of CCR9, but not CCR7 (for which CCL25 is not a ligand) on CLPs was further upregulated as they trafficked from bone marrow to thymus, recapitulating the pattern seen in young animals but absent in aged controls (**Extended Data Fig. 7F-G**). These findings suggest that hepatocyte-bound DLL1 may deliver a peripheral Notch signal to circulating progenitors in the liver sinusoids, priming them for efficient thymic entry and T lineage bias, while systemic IL-7 and FLT3-L support their expansion and survival.

Notably, this expansion of CCR9⁺ CLPs aligns with the observed restoration of DN1-DN3 thymocytes, increased thymic size and cellularity, and the subsequent rise in naïve T cells and recent thymic emigrants (**Fig. 2B-G, Extended Data Fig. 6F and 9**).

Together, we believe these data unify the effects of DFI into a mechanistic framework: by amplifying and priming committed lymphoid progenitors, DFI bypasses stem cell-intrinsic aging constraints and boosts thymopoiesis, ultimately supporting peripheral naïve T cell pools in aged hosts. We are grateful to the reviewer for prompting this analysis, as these results provide key evidence for the mechanistic basis of DFI's action and solidify our conclusion that its immune rejuvenating effects stem from targeted amplification and conditioning of lymphoid progenitors rather than wholesale remodeling of the HSC compartment.

2. The analysis of the 'aged' improved vaccine response in Fig 3 would benefit from a recall response from draining lymph nodes and/or spleen to determine CD4 Th1/Th2 proclivity and an analysis of OVA-specific immunoglobulin isotypes in circulation.

We thank the reviewer for this suggestion, which allowed us to further characterize the quality of the immune response elicited by DFI treatment. We have now performed peptide vaccination followed by *ex vivo* recall assays using splenocytes from aged mice treated with DFI, analyzing both CD8⁺ and CD4⁺ compartments. These studies confirmed our initial data in that the recall response was predominantly CD8⁺ driven, with robust IFN- γ and TNF production (**Fig. 3H; Point-by-point Fig. 2E**), while OVA-specific CD4⁺ T cells showed no consistent increase in frequency or enhancement of Th1 or Th2 markers, including IFN- γ , T-bet, or IL-17 (**Supplementary Fig. 4A-E**,

Point-by-point Fig. 2E). This suggests that DFI preferentially supports cytotoxic T cell responses rather than broadly augmenting helper T cell polarization.

We also extended our analysis to the humoral arm of the response. Consistent with the observed remodeling of the splenic B cell compartment by DFI (**Fig. 2K-L**), we found a robust increase in total antigen-specific IgG levels in the serum of aged mice, nearly approaching levels in young controls (**Fig. 3I; Supplementary Fig. 4F-G**). In parallel, IgM levels, which were elevated in aged controls, were significantly reduced after DFI treatment, indicating a shift toward class-switched, mature humoral immunity (**Supplementary Fig. 4F**). Subclass-specific ELISAs, however, revealed no detectable IgG1 or IgG2c (**Supplementary Fig. 4H-J**), suggesting a bias toward non-canonical subclasses such as IgG2b or IgG3. Such skewing is consistent with prior observations in aged C57BL/6J mice, where impairments in T helper cell function, germinal center architecture, and Th1/Th2 polarization can lead to atypical isotype profiles despite enhanced class switching and might therefore also be reflective of the limited effect of DFI on the T helper cell compartment (Frasca et al., *Front Immunol.* 2017; Tongren et al., *Infection and Immunity* 2006).

Together, these results indicate that DFI primarily boosts cytotoxic CD8⁺ recall responses while restoring aspects of humoral immunity through enhanced IgG production and class switching, despite the persistent helper T cell limitations characteristic of aged hosts.

3. Can the authors provide a rationale for the choice of DLL1 (vs DLL4) for the reconstitution of Notch signalling? Was the decision informed by the Slide-Seq cTEC spatial interactions or an empirical choice?

We thank the reviewer for highlighting this important point. The selection of DLL1 over DLL4 was deliberate and based on both biological safety considerations and functional differences between these Notch ligands:

DLL4 is a potent inducer of angiogenesis through its high-affinity activation of Notch1 and Notch4 in endothelial cells. Sustained or ectopic DLL4 expression in the liver has been associated with vascular remodeling and hepatic sinusoidal dysfunction, especially in aged or diseased tissue (Hellström et al., *Nature* 2007, Shen et al., *Am J. Pathol* 2016). Given the advanced age of our study animals and our aim to avoid potential complications in liver vasculature, DLL1 was the safer choice.

While both DLL1 and DLL4 can activate Notch1 signaling, DLL4 induces stronger and more sustained signaling and has been shown to restrict B cell development more severely than DLL1, which could be detrimental in the context of immune aging where B cell function is already compromised (Mohtashami et al., *J Immunol.* 2010; Feyerabend et al., *Immunity* 2009). By contrast, DLL1 provides sufficient Notch activation to support T lineage development and immune cell homeostasis without severely suppressing B lineage output, a balance particularly important in aged hosts.

Furthermore, it is worth noting that DLL1 is not typically associated with the pro-fibrotic, cholangiopathic, or carcinogenic outcomes reported for DLL4 or Jagged family ligands in the liver (Shen et al., *Am J. Pathol* 2016; Nakano et al., *Commun Biol.* 2022), which we believe reduces the likelihood of potential hepatotoxicity by DFI.

While our Slide-seq data showed reduced Notch ligand expression in aged cTECs, both DLL1 and DLL4 transcripts declined with age. However, based on the above functional considerations and prior literature, we prioritized DLL1 as the more appropriate ligand for ectopic hepatic expression. Future studies may explore whether DLL4 delivery is feasible in specific contexts such as cancer immunotherapy, but we opted for DLL1 to preserve a broader spectrum of immune function in aging.

We have updated the main text to include these rationales (Lines 94-99).

4. Have the authors determined the longevity of the DFI benefit post treatment? After the DFI LNP treatment regimen – do the mice maintain a rejuvenated naïve T cell phenotype, splenic cDC1/B cell phenotype? CD8+ response to OVA immunization?

We thank the Reviewer for this important question, which we also received from the other Referees. To address the longevity and stability of DFI's effects, we performed an extended analysis of immune output, immune composition and functional vaccine responses after cessation of treatment, now presented in **Supplementary Fig. 6**. These results distinguish the transient and persistent components of the DFI response.

Thymic output:

DFI treatment significantly increased intrathymic T cell receptor recombination (**Fig. 2F**) and T cell excision circles (TRECs) in the blood, confirming enhanced thymic emigrant production during the dosing period (**Fig. 2G**). However, TRECs returned to baseline within four weeks after the final dose (**Supplementary Fig. 6C**), reflecting the short half-life of the delivered mRNAs and the decline of Rag2 expression in thymocytes within 72 hours after one dose (**Fig. 2F; Extended Data Fig. 9C-D**).

Peripheral immune compartments:

Despite the loss of ongoing thymic export, DFI-induced increases in total splenic T cell and dendritic cell numbers persisted for at least four weeks post-treatment (**Supplementary Fig. 6G**). STARmap spatial transcriptomics of spleens collected immediately and four weeks after treatment confirmed stable enrichment of mature B cells and DCs within periarteriolar lymphoid sheaths. These findings indicate that, while thymic output is transient, the expanded peripheral compartments generated during treatment remain partially stabilized even in the absence of continued dosing. However, immunological challenges might deplete these compartments again, as outlined below.

Functional vaccine responses:

To determine whether these persistent cellular changes translate into equally persistent functional benefits, we delayed vaccination until four weeks after the last DFI dose (**Supplementary Fig. 6H**). At this point, we no longer observed enhanced antigen-specific CD8⁺ T cell expansion (**Supplementary Fig. 6I**), attenuated exhaustion (**Supplementary Fig. 6J**), or favorable shifts in naïve, effector, and memory T cell pools (**Supplementary Fig. 6K**). This suggests that the window for optimized vaccine or immunotherapy responses coincides with the active treatment period, when both thymic export and peripheral remodeling are maximized.

In summary, DFI induces a reversible thymic activation that drives *de novo* T cell output only during dosing, while concurrently reshaping splenic T, B, and dendritic cell compartments in a manner that persists for weeks. However, to achieve maximal functional benefits such as improved vaccine responses, DFI administration should be synchronized with antigen exposure, as the full breadth of immune enhancement does not persist beyond the treatment window. This transient and controllable profile may however represent a clinical advantage, as it allows for timed immunological boosting without prolonged exposure, thereby minimizing risks of chronic stimulation or autoimmunity.

5. FIG1ED E & H are a bit hard to interpret. I understand that the grey bars correspond to populations in the UMAP that do not change significantly compared to 4-week-old mice, but it is not immediately obvious which grey bars correspond to which populations. I surmise they are in the same order from L to R /top to bottom as the UMAP populations in Fig1ED C & F, respectively. Perhaps the authors could label the populations 1-7 for 1C and 1-5 for 1F. Alternatively, perhaps exclude the not significant grey bars in Figure 1ED E & H and focus on the significant changes only?

We thank the reviewer for pointing out this issue. To improve clarity, we have now labeled the UMAP populations in Fig. 1ED C and F as populations 1-7 and 1-5, respectively, and used the same labels in Fig. 1ED E and H, as suggested. This should allow for a direct and intuitive correspondence between the UMAP plots and the bar graphs.

6. LINE 44-45 – define DLL1, FLT3-L, and IL-7 in first appearance

Thank you for pointing this out, we have defined the three factors at this point.

“To counteract age-associated thymic involution, we delivered mRNAs encoding Delta-like ligand 1 (DLL1), Fms-like tyrosine kinase 3 ligand (FLT3-L), and interleukin-7 (IL-7) to the liver using lipid nanoparticles (LNPs), thereby providing key thymic signals ectopically.”

7. Fig 3ED F – typo - ‘sample’ is misspelled in figure title

We thank the reviewer for catching this and have corrected it.

8. Fig 3ED L – The authors speculated that it is the LNP responsibly for slightly elevated circulating IL-6 in vivo (Lines 134-146). A better control that may answer this definitively would be Luc mRNA (or other RNA) LNP rather than NaCl/saline – as they have done for other experiments.

We thank the reviewer for this suggestion. We performed an extended comparison using Luc mRNA-LNP as a control, as suggested, alongside each DFI component and the full combination, after four weeks of repeated dosing in aged mice (**Supplementary Fig. 1I**). IL-6 levels were not significantly elevated in any group, including Luc mRNA-LNP, indicating that neither the LNP formulation nor the DFI components induce sustained systemic IL-6 or generalized inflammation under our treatment conditions.

9. Fig 2G – CD83? – CD80 is described in the text (Line 185). I’m not sure if the Figure or text is correct?

| We thank the reviewer for catching this and have corrected it (It is CD83).

10. Fig 7ED C–CD83 in figure – it is not clear which one the authors meant given previous point.

| We thank the reviewer for catching this and have corrected it (It is CD83).

11. Line 227 – call out should be Fig 3H (not 3F)

| We thank the reviewer for catching this and have corrected it.

12. Line 231 – call out should be Fig 3I (not 3G)

| We thank the reviewer for catching this and have corrected it.

8. References. Does the manuscript reference previous literature appropriately?

The manuscript is well-cited and provide appropriate background context to position their work within current literature.

9. Clarity and context. Is the abstract clear and accessible? Are the abstract and introduction appropriate?

The abstract and introduction are well crafted to introduce the paper.

10. Please indicate any particular aspect of the manuscript, data or analyses that you feel is outside the scope of your expertise, or that you were unable to assess fully.

I did not re-analyse the raw data files provided.

Referee #2 (Remarks to the Author):

In this manuscript, Friedrich et al. applying several cutting-edge omics to thymus samples from aged mice identify that IL7 and Notch signaling are linked to immune system aging. By hydrodynamic delivery of lipid nanoparticles containing the mRNA for three selected factors related to these pathways and FLT-3L (based on previous reports) they achieve a transitory production of these three factors in the liver. Notably, this is sufficient to rejuvenate immune function in aged mice, leading to improved responses to vaccination and cancer immunotherapy without any observed side effect. Although the factors that the authors use to reconstitute immune function in aged mice were already established for thymus repair and regeneration, the novelty of this paper resides in the technology based on lipid nanoparticles containing mRNA as a delivering method to the liver is a potential strategy to improve immune function deterioration with aging. This paper supposes a breakthrough in the immune aging field and opens exciting opportunities to overcome immune cell dysfunction during aging.

Major comments:

- With age, the TCR repertoire diminishes, which compromises the ability of T cells to mount an effective immune response. To really understand the immune reconstitution potential of this strategy, it is essential to know if TCR repertoire diversity is restored upon the treatment or still remains reduced. Moreover, the authors claim that the treatment is enhancing T cell development in the thymus. To really demonstrate this, it will be required to analyze TRECs, as a readout of T cell development and TCR recombination.

We thank the reviewer for this helpful suggestion, which we have acted upon. We found that DFI treatment significantly increased TRECs in peripheral blood, indicating enhanced recent thymic emigrant output (**Fig. 2G**). This is further supported by several other lines of evidence:

To further substantiate active T cell development, we made use of *Rag2*-EGFP reporter mice, which allow real-time tracking of TCR recombination in early thymocytes. Here, we observed the induction of *Rag2* expression in thymocytes within 72 of DFI administration (**Fig. 2F**; **Extended Data Fig. 9A-C**).

To further confirm that these new T cells originate from thymic selection rather than peripheral expansion, we utilized *Nur77*-GFP reporter mice, which mark cells undergoing TCR signaling, particularly during positive selection. DFI-treated aged mice displayed increased *Nur77*-GFP expression in thymocytes, consistent with heightened self-pMHC-driven selection, and a higher frequency of *Nur77*-GFP⁺ naïve T cells in peripheral blood (**Extended Data Fig. 9E-G**). As residual GFP signal in peripheral naïve cells reflects their recent thymic origin, these results provide an additional line of evidence that DFI drives *bona fide* thymic output rather than expansion of pre-existing clones.

Functionally, we assessed how these newly generated T cells impact repertoire quality. Bulk TCR β sequencing of splenic T cells showed no significant change in global diversity metrics (**Fig. 2E**), which likely reflects the small numerical contribution of recent thymic emigrants relative to the large, established peripheral pool. However, single-cell RNA + TCR sequencing of tumor-infiltrating lymphocytes (TILs) in B16-OVA-bearing mice revealed that DFI preconditioning (i) increased TCR clonotype diversity within tumors, (ii) expanded tumor-specific naïve-like CD8⁺ T cells, and (iii)

reduced the proportion of terminally exhausted clones (**Fig. 4T-V**). These data suggest that while bulk splenic repertoire diversity appears unchanged, the functional repertoire at the tumor site is rejuvenated, contributing to the enhanced endogenous anti-tumor immunity and therapeutic synergy with anti-PD-1 observed in our model.

Taken together, increased TRECs, induction of Rag2 and Nur77 reporters, and the observed shift in TIL clonality demonstrate that DFI treatment restores thymic output and contributes to the functional diversification of the T cell compartment. The ability to transiently reseed the T cell pool with newly generated, diverse clones suggests that DFI could potentially be clinically feasible not only for immune aging but also for patients recovering from lymphodepleting therapies, where rapid restoration of a functional repertoire is critical.

- Is the rejuvenation of the immune system attributed to a re-organization/formation of the thymus? Are LNPs able to rejuvenate T cells in the absence of the thymus? Considering FLT-3 is a well-known regulator of hematopoiesis, I am not entirely convinced that they are not improving immune system function just by rebalancing the typical myeloid-biased of hematopoiesis that occurs during aging. In other words, how are the levels of hematopoietic precursors (HSCs, GMPs, CMPs and CLPs) in aged mice treated with LNPs? How are the levels of monocytes in the blood and bone marrow of these mice?

Re-organization/formation of the thymus:

In light of the new data added in the revised manuscript, which now provide us with a more holistic view of DFI's effects across bone marrow, blood, liver, and thymus, we conclude that immune rejuvenation is not attributable to a direct re-organization or re-formation of the thymus.

We performed in-depth analyses of explanted thymuses from aged mice after four weeks of DFI or Luc treatment, using histology, Slide-seq for transcriptional profiling of thymic epithelial cells (TECs), as TECs are highly sensitive to dissociation and therefore challenging to assess using conventional single-cell RNA-seq (**Point-by-point Fig. 3**), as well as flow cytometry of intrathymic DCs and mTECs (**Point-by-point Fig. 4**). While thymus weight consistently increased in DFI-treated mice, this reflected a rise in overall cellularity (**Point-by-point Fig. 3C**), likely driven by enhanced homing of CCR9⁺ CLPs (**Extended Data Fig. 7D-E**) and expansion of thymocytes (**Extended Data Fig. 6E**), rather than structural remodeling.

Specifically, we detected no numerical increase in early TECs, cortical TECs (cTECs), or medullary TECs (mTECs) (**Point-by-point Fig. 3E**). Slide-seq revealed no transcriptional alterations in key transcription factors, Notch ligands, or functional markers in any TEC subset (**Point-by-point Fig. 3F-H**), arguing against compositional or functional reprogramming of the thymic stroma. Although DFI slightly increased the relative frequency of intrathymic DCs compared to Luc controls, their absolute numbers remained unchanged (**Point-by-point Fig. 4A**). Additionally, surface expression of CD80 and MHC-II on intrathymic DCs and RANK, CD80 and AIRE on mTECs was stable across age and treatment (**Point-by-point Fig. 4A-B**), indicating that mechanisms central to negative selection remain intact and are not perturbed by DFI.

These findings support our mechanistic model that DFI exerts its rejuvenating effects primarily through FLT3-L mediated induction in the bone marrow and liver-localized DLL1/IL-7 signaling,

which might expand and condition CLPs before thymic entry, rather than by reorganizing the thymic architecture. The thymus acts as a receptive site for these progenitors, but its stromal composition and negative selection machinery remain stable, underscoring our animal model data in that the increased thymopoiesis induced by DFI does not come at the expense of central tolerance.

[Redacted text]

HSC myeloid bias and profiling of hematopoietic precursors:

We thank the reviewer for raising this point. To determine whether DFI's effects are mediated by rebalancing the age-associated myeloid bias in hematopoiesis, we comprehensively profiled hematopoietic stem and progenitor populations in adult, aged, and DFI-treated aged mice (**Extended Data Fig. 8A-I**).

Consistent with prior reports (Morrison et al., 1996; Saçma et al., 2019; Pang et al., 2011), total HSCs (Lin⁻ c-Kit⁺ Sca-1⁺ FLT3⁻ CD34⁻ CD150⁺) were increased in frequency with age, and we confirmed the expected compositional shift toward CD150^{high} myeloid-biased HSCs at the expense of balanced HSCs (bal-HSCs) (**Extended Data Fig. 8A, C**). Importantly, DFI treatment did not significantly alter

this bias or rebalance the HSC compartment (**Extended Data Fig. 8C-D**), unlike strategies such as antibody-mediated my-HSC depletion (Ross et al., *Nature* 2024). Multipotent progenitor (MPP) frequencies were similarly unchanged, apart from a modest increase in the lymphoid-primed MPPc subset (**Extended Data Fig. 8B**).

The dominant effect of DFI was observed downstream, at the level of committed progenitors. Common lymphoid progenitors (CLPs) decline markedly with age (Miller and Allman, *Nat Immunol* 2003; Rossi et al., *Nature* 2005), however, DFI robustly expanded CLPs in the bone marrow (**Extended Data Fig. 8E-F**). We attribute this effect, at least in part, to the stably increased systemic levels of FLT3-L following DFI administration, which likely drive CLP expansion and survival within the marrow. In parallel, hepatocyte-bound DLL1 and liver-derived IL-7 likely act on circulating progenitors to promote the observed CCR9 upregulation (**Extended Data Fig. 7D-G**), which modulates their thymus-homing capacity and T lineage bias. Together, these coordinated signals more likely explain the observed rise in DN1-DN3 thymocytes and the subsequent replenishment of naïve T cells in peripheral compartments we observed (**Fig. 2B-G**).

Thus, we believe that DFI does not rejuvenate immunity by reorganizing the HSC pool or correcting the age-associated myeloid bias, but by leveraging FLT3-L-driven progenitor expansion in the marrow and DLL1/IL-7-mediated priming in the periphery, thereby bypassing stem cell-intrinsic aging constraints to restore thymopoiesis and T cell production in aged hosts.

Monocytes in the bone marrow and blood:

We thank the reviewer for this question. We analyzed circulating monocyte frequencies in adult and aged mice treated with Luc, each individual DFI component, or the full DFI combination. Overall frequencies of total monocytes in blood were not significantly altered by age or by any treatment (**Point-by-point Fig. 6A-B**).

When examining monocyte subsets, however, we found patterns consistent with published reports of age-associated myeloid remodeling. Classical, Ly6C^{high} CCR2^{high} CX3CR1^{low} monocytes decline with age (as described in humans by Seidler et al., *BMC Immunol* 2010), while non-classical, Ly6C^{low} CCR2^{low} CX3CR1^{high} patrolling monocytes are relatively increased in aged circulation. DFI treatment partially restored classical monocyte frequencies to levels comparable to those in young adult controls, while reducing the excess of non-classical monocytes (**Point-by-point Fig. 6C-D**).

This shift aligns with our observation that DFI-treated aged mice exhibit increased levels of common myeloid progenitors (CMPs) in bone marrow (**Point-by-point Fig. 6A**). Classical monocytes predominantly arise from CMP-derived monocyte–dendritic progenitors (MDPs), whereas non-classical monocytes derive in part from alternative, less proliferative pathways. The re-expansion of CMPs with DFI, likely driven by FLT3-L-mediated support, may thus preferentially bolster classical monocyte output, shifting the circulating monocyte compartment toward a distribution seen in adult animals.

Although the functional consequences of this rebalancing remain to be fully elucidated, classical monocytes play a key role in antigen presentation and recruitment during infection and vaccination,

whereas excessive non-classical monocytes in aging have been linked to low-grade inflammation, senescence and impaired pathogen clearance (Ong et al, *Cell Death & Disease* 2018).

-How does old treated mice response to a viral infection challenge?

The reviewer asks a great question, and although a viral infection model would be highly relevant, due to institutional biosafety restrictions in our animal facility, we are currently unable to perform live viral infections in aged mice. We deeply regret this limitation and appreciate the reviewer's understanding. To partially address this important point, we have added two new studies using well-established surrogate models of immune responsiveness:

1. Tumor challenge models reveal enhanced T cell diversity and functionality at the effector site (**Fig. 4**), as well as increased spontaneous rejection of immunogenic tumors, providing evidence for improved antigen-specific immunity.
2. Increased thymic TCR recombination (Rag2-EGFP), recent thymic emigrants (TRECs), and expansion of peripheral T cells and dendritic cells all support functional immune rejuvenation that results in improved T cellular and humoral vaccine responses (**Fig. 2, Fig. 3, Supplementary Fig. 4**).

We have updated the discussion of our manuscript to reflect this important limitation.

- Have the authors tested how long these effects last after stopping the administration of LNPs? How long these factors keep upregulated in the blood after stopping LNP delivery? Related to this, have they tested whether extending this treatment (maybe with some intermediate resting periods) could exert more potent responses or, opposite, would lead to a loosening of the observed effects?

The reviewer raises several important questions regarding the kinetics and sustainability of the immune effects observed with DFI treatment, which we address below.

1. *Duration of factor expression in vivo:*

We assessed the pharmacokinetics of each DFI-encoded factor following a single LNP injection. As shown in **Fig. 1H** (DLL1), **Fig. 1I** (FLT3-L), and **Fig. 1J** (IL-7), all three proteins exhibit peak expression between 6 and 24 hours, followed by a rapid decline over the subsequent 48-72 hours, consistent with the short half-life of non-replicating mRNA-LNP formulations. This transient expression achieves robust biological effects without persistent systemic cytokine exposure, thereby minimizing the risk of toxicity or desensitization.

2. *Longevity of immunological effects post-treatment:*

While cytokine levels return to baseline within days, we observe that DFI-induced immune changes persist much longer:

- Thymic recombination activity (Rag2-EGFP mice) rises within 24 hours of treatment and is sustained for 72 hours (**Fig. 2F, Extended Data Fig. 9A-C**).
- Recent thymic emigrants (remaining GFP signal in *Nur77*-GFP mice) are also sustained for 72 hours following a single dose of DFI (**Extended Data Fig. 9E-G**)
- Following a 4-week DFI treatment regimen, the increased thymic output gradually declines after treatment cessation, as expected (**Supplementary Fig. 6C**).

- However, the expanded splenic T cell, cDC1, and B cell populations remain stable for at least 4 weeks post-treatment (**Supplementary Fig. 6E-G**), indicating durable reshaping of the peripheral immune landscape even after DFI expression ceases.

3. *Response durability and functional window:*

When vaccination was delayed by 4 weeks after stopping DFI treatment, the enhanced CD8⁺ recall response and tetramer⁺ T cell expansion were no longer detectable (**Supplementary Fig. 6H-K**). This suggests that the window for optimal immunological benefit coincides with or immediately follows DFI treatment, highlighting the importance of synchronization with antigen challenge.

Together, these results indicate that while the encoded factors are transiently expressed, some of the cellular immunological effects of DFI treatment persist for weeks and remain re-inducible with subsequent dosing. While vaccine responsiveness is highest during or shortly after treatment, repeated regimens appear biologically viable and may enable tailored immune support over extended timeframes.

-To firmly demonstrate that self-tolerance is not compromised, it will be ideal to add an additional model of autoimmunity avoiding the use of TCR transgenic mouse models (like NOD), such as EAE.

We thank the reviewer for this important suggestion to evaluate self-tolerance in a polyclonal, inducible autoimmune model. As suggested, we tested DFI in EAE induced by MOG₃₅₋₅₅ peptide vaccination and pertussis toxin administration, and monitored disease by clinical scoring, MHC-tetramer quantification of CNS-infiltrating MOG-specific CD4⁺ T cells, and blinded histopathological assessment by an external CRO pathologist (**Extended Data Fig. 10B-H**).

Consistent with previous studies (Atkinson et al., *JCI Insight* 2022, de la Fuente et al., *Nat Commun* 2024), aged control mice exhibited delayed EAE onset compared to young adults, reflecting diminished T cell priming and reduced CNS infiltration typical of immune senescence. Despite this muted priming, aged mice developed more severe clinical deterioration as disease progressed, likely reflecting physiological frailty rather than uncontrolled immunity (**Extended Data Fig. 10C-D**). Notably, de la Fuente et al. demonstrated that the aged CNS harbors abundant but transcriptionally distinct Tregs, whose altered programs do not inherently breach tissue tolerance unless the local immune environment is profoundly disrupted. In line with this, although DFI-treated aged mice displayed increased frequencies of I-A^b-restricted MOG-specific CD4⁺ T cells in the spleen, consistent with improved peripheral priming (**Extended Data Fig. 10E-F**), they did not exhibit increased autoreactive T cells in the CNS, exacerbated clinical scores, histopathological CNS inflammation, or demyelination (**Extended Data Fig. 10D, G-H**). In this context, viewed apart from the CNS manifestations, we believe that EAE induction essentially functions as a peptide vaccination model, and together with our OVA data, these findings show that DFI enhances peripheral antigen-specific priming without compromising the tissue-resident regulatory networks that preserve CNS immune privilege.

Adult mice treated with DFI showed neither an increase in MOG-reactive T cells nor any change in disease outcomes (**Extended Data Fig. 10C, E-H**), underscoring that DFI corrects age-related immune deficits rather than amplifying pathogenic activation indiscriminately.

Together with our peptide vaccination, mAct-OVA tolerance, and NOD studies, these findings demonstrate that DFI can enhance antigen-specific immunity in aged hosts without breaching self-tolerance or exacerbating tissue-specific autoimmunity. Its effects remain transient and controllable, which we believe underscores the potential of DFI mRNA-LNPs to be further developed as a pre-conditioning or adjuvant strategy for rejuvenating immunity in older individuals.

- Most cancer models used, while valid for the tested hypothesis, are somehow not very representative of the real scenario and incorporate the co-administration of immunotherapy. I would be curious of whether LNPs could also affect the endogenous response and the cancer outcome in any model of typically immunogenic cancer without the co-administration of ICI.

We thank the reviewer for this insightful comment, which prompted us to directly test whether DFI alone can restore endogenous tumor control in an immunogenic setting, independent of immune checkpoint blockade. To address this, we used the MC38-OVA colon carcinoma model, a typically immunogenic tumor type sensitive to T cell-mediated immune surveillance. Consistent with prior reports (Georgiev et al., *Cancer Immunol Res* 2024), MC38-OVA tumors in aged mice grew faster and were rejected less frequently than in adult controls, confirming the contraction of tumor-specific immunity with age.

We then treated aged MC38-bearing mice with DFI LNPs alone for four weeks, without any additional immunotherapy before challenging these animals with subcutaneous tumor injection. This intervention significantly slowed tumor growth, increased rates of spontaneous tumor rejection, and improved overall survival over a 120-day observation period compared to untreated aged controls (**Fig. 4A-D**). Mechanistically, these effects align with the broader immune remodeling we observed in DFI-treated mice, specifically the enhanced abundance of naïve and tumor-specific CD8⁺ T cells with reduced exhaustion (**Fig. 4T-V**). Together, these data suggest that DFI can augment endogenous tumor immunity. We believe this strengthens the applicability of DFI as a standalone intervention, particularly in older patients where bolstering intrinsic tumor surveillance could reduce the need for combination regimens with higher toxicity.

Minor comments

- The abstract states that “Aging leads to a progressive decline in immune function, marked by a reduced T cell diversity...”. While TCR diversity is clearly reduced during aging, I do not think stating that T cells are less diverse in aged mice is correct. Indeed, recent findings suggest that aging is accompanied by the appearance of several new T cell subsets (PMID: 31457092, PMID: 33271118). Therefore, I would recommend editing this particular statement.

We thank the reviewer for this important clarification and for pointing us to the relevant literature. We have revised the first sentence of the abstract. We have also ensured that similar language is used consistently throughout the manuscript to avoid oversimplification.

- There are a couple of mistakes in the text regarding some figures: line 115 should be Fig 1F instead of 1E, line 117 should be Fig 1F-G instead of Fig 1E-F, line 227 should be Fig. 3H instead of 3F and line 231 should be Fig. 3I instead of 3G

| We thank the reviewer for pointing out these errors, which we have now corrected.

Referee #3 (Remarks to the Author):

Recommendations, major revisions.

Review:

T cells are known to be essential to maintain adaptive immunity. Treatments for lymphopenia are lacking for aged individuals or patients undergoing cytoreductive therapies. In this study the authors propose a treatment with mRNAs with known lymphopoietic factors to improve T cell production.

Major comments:

1) The authors claim the delivery of LNPs of well-known thymopoietic factors affects immune function partially through rejuvenation of T cell production and claim this is by boosting thymic function.

We thank the reviewer for highlighting this central claim of our manuscript. Our study builds on the hypothesis that transient hepatic expression of three key thymopoietic ligands - DLL1, FLT3L, and IL-7 - can restore immune competence in aged hosts when synthetically expressed by enhancing thymic T cell development. We support this mechanistically through multiple orthogonal lines of evidence, each demonstrating increased thymic activity and *de novo* T cell output:

1. Increased thymic size and early thymocyte stages

After 28 days of DFI treatment, aged mice exhibited significantly increased thymus weight and cellularity (**Extended Data Fig. 6A-C**), driven by expansion of DN1–DN3 thymocytes, the critical stages of T cell lineage commitment (**Extended Data Fig. 6F**). The relative distribution of double-positive and single-positive thymocytes remained normal (**Extended Data Fig. 6E**), indicating that DFI amplifies early thymopoiesis without skewing later developmental checkpoints.

2. Increased progenitor supply and thymus-homing capacity

Flow cytometric profiling revealed robust expansion of common lymphoid progenitors (CLPs) in bone marrow and peripheral blood (**Extended Data Fig. 7C, 8F**), despite their marked age-associated decline. These CLPs displayed enhanced CCR9 expression and preferential thymus-homing potential (**Extended Data Fig. 7D-G**), a phenotype resembling that in young mice but absent in untreated aged controls.

3. Evidence of active thymopoiesis

Using *Rag2*-EGFP reporter mice, we observed a rapid induction of *Rag2* expression in thymocytes after 72h (**Fig. 2F, Extended Data Fig. 9A-F**). *Nur77*-GFP reporter mice demonstrated increased GFP signal in peripheral naïve T cells (**Extended Data Fig. 9G**), consistent with active positive selection and the recent emigration of thymus-derived cells. Peripheral T cell receptor excision circles (TRECs) were also elevated during treatment (**Fig. 2G**), further confirming increased *de novo* thymic output.

4. Functional diversification of the antigen-specific T cell pool

While bulk splenic TCR β sequencing revealed no global increase in diversity metrics (**Fig. 2E**),

single-cell RNA + TCR sequencing of tumor-infiltrating lymphocytes showed that DFI preconditioning enhanced clonotype diversity, expanded tumor-specific naïve-like CD8⁺ T cells, and reduced terminally exhausted clones (**Fig. 4T-V**). These findings indicate that DFI selectively replenishes functional, antigen-responsive T cell populations, even if their contribution to the total splenic repertoire remains numerically modest.

5. Preserved stromal integrity and tolerance mechanisms

Importantly, these effects occur without structural remodeling of the thymus. Histology and Slide-seq v2 profiling showed no increase in early TEC, cortical TEC, or medullary TEC numbers and no transcriptional changes in Notch ligands or functional markers (**Point-by-point Fig. 3E-H**). Intrathymic dendritic cells and mTECs maintained stable absolute numbers and surface expression of CD80 and MHC-II (intrathymic DCs), and RANK, CD80, AIRE (mTECs), respectively, (**Point-by-point Fig. 4A-B**), further indicating that negative selection and central tolerance remain intact.

Collectively, these findings support our conclusion that DFI augments immune function in aged hosts by boosting thymic T cell production through peripheral amplification and conditioning of CLPs (via systemic FLT3-L, hepatocyte-bound DLL1, and secreted IL-7), which drives enhanced thymopoiesis and *de novo* T cell output. The aged thymus itself serves as a receptive site for these progenitors without requiring stromal reorganization, allowing for functionally relevant but controlled immune rejuvenation.

2) Although the manuscript has multiple orthogonal ways to show immune function the manuscript does describe well known thymopoietic factors. The manuscript would benefit for a better mechanistic understanding of the effect on thymic biology, transient vs long-lasting, bone marrow phenotype, CLP, ETPs etc. vs stromal cells.

We thank the reviewer for emphasizing the importance of clarifying the mechanistic basis of DFI's immune rejuvenation, particularly regarding thymic biology, progenitor dynamics, and stromal stability. In light of the new datasets added to the revised manuscript which together provide a more holistic view of DFI's pleiotropic actions across bone marrow, blood, liver, and thymus. We summarize our findings below:

Transient thymic output increase without stromal remodeling

DFI treatment consistently increased thymus weight and cellularity (**Point-by-point Fig. 3C**), driven by an influx of CCR9⁺ CLPs (**Extended Data Fig. 7D-E**) and expansion of DN1–DN3 thymocytes (**Extended Data Fig. 6E**), rather than structural changes. Histology and Slide-seq profiling of cortical and medullary thymic epithelial cells (TECs) revealed no numerical increase in early TECs, cTECs, or mTECs (**Point-by-point Fig. 3E**) and no transcriptional alterations in key transcription factors, Notch ligands, or functional markers across TEC subsets (**Point-by-point Fig. 3F-H**). Moreover, while intrathymic dendritic cells were more frequent relative to Luc-treated controls, their absolute numbers and surface expression of CD80 and MHC-II, which are both critical for negative selection, remained unchanged (**Point-by-point Fig. 4A**). These data indicate that the thymus functions as a receptive site for expanded progenitors, but its stromal and tolerogenic machinery remain stable, suggesting that the observed enhanced thymopoiesis does not compromise central tolerance.

Bone marrow progenitor effects and systemic FLT3-L

Flow cytometric profiling showed that DFI does not reprogram the hematopoietic stem cell (HSC) pool or correct the well-established age-related shift toward myeloid-biased CD150^{high} HSCs (**Extended Data Fig. 8A-D**). Multipotent progenitors were similarly unaffected apart from a modest rise in the lymphoid-primed MPPc subset (**Extended Data Fig. 8B**). Instead, the dominant effect occurs downstream, with robust expansion of common lymphoid progenitors (CLPs) in bone marrow and peripheral blood, reaching near-adult levels (**Extended Data Fig. 8E-F**). This effect is likely driven by stably elevated systemic FLT3-L following DFI treatment (**Fig. 1I**), which promotes CLP survival and proliferation.

Functional consequences and durability

Together, these coordinated signals explain the restoration of DN1-DN3 thymocytes, increased thymic output (TRECs, Rag2-EGFP, and Nur77 reporters; **Fig. 2F-G, Extended Data Fig. 9**), and replenishment of peripheral naïve T cell pools. These effects are largely transient at the thymic level: TRECs return to baseline within four weeks of stopping treatment (**Supplementary Fig. 6C**), but the expanded splenic T cell, cDC1, and B cell populations persist for weeks (**Supplementary Fig. 6E-G**). Functionally, however, the enhanced vaccine responses observed during active treatment are not sustained: when OVA immunization was delayed until four weeks after the last DFI dose, no significant improvement in antigen-specific CD8⁺ T cell expansion or cytokine production was detected (**Supplementary Fig. 6H-K**). These results indicate that while DFI induces a durable reshaping of peripheral immune compartments, its optimal functional benefit requires temporal coordination with vaccination or immunotherapy. However, these data also suggest the transient and controllable application of DFI LNPs.

Collectively, these data establish that DFI rejuvenates immune function not by reorganizing or regenerating the thymic stroma, nor by globally rebalancing the aged HSC pool, but by leveraging FLT3-L-driven progenitor expansion in bone marrow and DLL1/IL-7-mediated conditioning in the liver to amplify and prime CLPs for efficient thymopoiesis. The thymus serves as a receptive niche, maintaining its stromal integrity and tolerance mechanisms, while the peripheral immune system benefits from a replenished T cell pool and transient IL-7 effects.

3) Is the effect dependent on functional thymic tissue or is there extrathymic T cell production? Is the same phenomenon observed in FOXP1- mice?

[Redacted text]

[Redacted text]

4) Intrathymic function. The authors could do this through a) flow cytometry of thymic epithelial cells, 2) other supporting stroma, 3) FOXP1 expression in cTECs and or mTECs, what is the effect on $\alpha\beta$ vs. $\gamma\delta$ T cells? Intrathymic DCs?

We performed detailed analyses of explanted thymuses from aged mice after four weeks of DFI or Luc treatment, using histology, Slide-seq for transcriptional profiling of thymic epithelial cells (TECs) (**Point-by-point Fig. 3**), and flow cytometry of intrathymic DCs and mTECs (**Point-by-point Fig. 4**). While thymus weight consistently increased in DFI-treated mice, this was attributable to enhanced homing of CCR9⁺ common lymphoid progenitors (**Extended Data Fig. 7D-F**) and expansion of thymocytes (**Extended Data Fig. 6E**), rather than structural remodeling of the thymic stroma (**Point-by-point Fig. 3C**).

Specifically, we observed no numerical increase in early TECs, cortical TECs (cTECs), or medullary TECs (mTECs) (**Point-by-point Fig. 3E**). Slide-seq analysis confirmed stable expression of key transcriptional regulators, including *FOXN1*, Notch ligands, and functional markers across TEC subsets (**Point-by-point Fig. 3F-H**), arguing against compositional or functional reprogramming of the thymic stroma.

Although DFI modestly increased the relative frequency of intrathymic DCs compared to Luc controls, their absolute numbers were unchanged (**Point-by-point Fig. 4A**). Surface expression of CD80 and MHC-II on intrathymic DCs, and RANK, CD80, and AIRE on mTECs, remained stable across age and treatment (**Point-by-point Fig. 4A-B**). These findings indicate that mechanisms central to negative selection remain intact and are not perturbed by DFI.

Finally, flow cytometry of thymocyte populations revealed proportional increases across $\alpha\beta$ T cell precursors (DN, DP, SP, **Extended Data Fig. 6E-F**) without selective skewing toward $\gamma\delta$ T cell development, which, as described is affected by age as well (**Point-by-point Fig. 6E**), supporting that DFI broadly enhances thymopoiesis without altering lineage balance. Together, these results indicate that DFI enhances thymic output by expanding progenitor influx and thymocyte maturation while leaving the stromal and regulatory architecture, including TEC and DC function, largely unaltered.

5) Thymic output: The increase in CD44⁺ CD62L⁺ naïve T cells in the spleen, could be due to peripheral expansion and/or thymic production. Are TRECs increased? What happens in RAG2GFP mice? Nur77GFP or other models to assess recent thymic emigrants? What about TCR repertoire? Is IL7 leading to peripheral expansion or denovo seeding of T cells? Is the exhaustion profile altered in the LNP treated group in B16 melanoma?

We agree that distinguishing peripheral proliferation from *de novo* T cell production is critical for interpreting DFI's mechanism. We addressed this through multiple complementary approaches, which collectively indicate that DFI promotes new T cell generation rather than simple IL-7-driven homeostatic expansion of existing clones.

Direct markers of new thymic emigrants (TRECs, Rag2-EGFP)

Peripheral T cell receptor excision circles (TRECs) were significantly increased in DFI-treated aged mice (**Fig. 2F**), which is incompatible with peripheral proliferation, as TRECs are diluted during cell division. In *Rag2-EGFP* reporter mice, we observed rapid induction of GFP in DN2–DN3 thymocytes beginning 12 hours after DFI dosing, peaking at 72 hours, and returning to baseline after treatment cessation (**Fig. 2G**). Both markers reflect transient but repeated bursts of TCR recombination, a signature of active thymopoiesis.

Positive selection and export (Nur77-GFP)

Nur77-GFP reporter mice demonstrated increased GFP signal in thymocytes and a subset of circulating naïve T cells (**Extended Data Fig. 9E-H**), reflecting TCR engagement during positive selection and recent thymic export.

[Redacted text]

TCR repertoire analysis supports de novo seeding

Bulk TCR β sequencing of splenic T cells revealed no significant change in Simpson clonality (**Fig. 2E**). If IL-7 were primarily driving homeostatic proliferation of existing clones - a well-described effect of IL-7 monotherapy (Schluns et al., *Nat Immunol* 2000; Sportès et al., *J Exp Med* 2008) - we would expect an increase in clonality due to overrepresentation of select preferentially IL-7-responsive expanded clones. The absence of such skewing supports the conclusion that the additional naïve T cells represent new, low-frequency clones entering the repertoire, which remain underrepresented in bulk diversity metrics. Consistent with this, single-cell RNA + TCR sequencing

of tumor-infiltrating lymphocytes (TILs) revealed expanded clonotype diversity, enrichment of tumor-specific naïve-like CD8⁺ T cells, and fewer terminally exhausted clones (**Fig. 4V**), underscoring that DFI diversifies the functional T cell pool.

Exhaustion profiles in B16 melanoma

In the B16-OVA model, DFI preconditioning not only expanded the T cell pool but also improved its quality: frequencies of PD-1⁺TOX⁺ exhausted CD8⁺ cells were reduced, while TCF1⁺ or CD127⁺ naïve-like subsets were enriched (**Fig. 4T**). This suggests that DFI supports less-differentiated, self-renewing populations that sustain effective tumor immunity.

Role of IL-7 within the DFI combination

While IL-7 is critical for survival of developing and peripheral T cells, it does not, by itself, induce Rag2 expression or TRECs, nor does it support Notch-dependent T lineage commitment in human non-canonical extrathymic T cell development (Maillard et al., *Blood* 2006). Our data show that the key hallmarks of new T cell production - TRECs, Rag2 induction, Nur77 expression, expansion of DN1-DN3 thymocytes (**Extended Data Fig. 6F**), and CCR9⁺ CLPs (**Extended Data Fig. 7D-G**) - are observed only when IL-7 is combined with FLT3-L and DLL1. Thus, IL-7 in this context likely functions to enhance survival and thymic recruitment of CLPs, but cannot account for the observed effects through homeostatic proliferation alone.

[Redacted text]

6) If the treatment leads to more Thymic function, is it a lasting effect? Is the thymus still bigger after discontinuation? What happens after stopping treatment? Is there a sustained benefit?

We thank the Reviewer for this important question, which we also received from the other Referees. To address the longevity and stability of DFI's effects, we performed an extended analysis of immune output, immune composition and functional vaccine responses after cessation of treatment, now presented in **Supplementary Fig. 6**. These results distinguish the transient and persistent components of the DFI response.

Thymic output:

DFI treatment significantly increased intrathymic T cell receptor recombination (**Fig. 2F**) and T cell excision circles (TRECs) in the blood, confirming enhanced thymic emigrant production during the dosing period (**Fig. 2G**). However, TRECs returned to baseline within four weeks after the final dose (**Supplementary Fig. 6C**), reflecting the short half-life of the delivered mRNAs and the decline of Rag2 expression in thymocytes within 72 hours after one dose (**Fig. 2F; Extended Data Fig. 9C-D**).

Peripheral immune compartments:

Despite the loss of ongoing thymic export, DFI-induced increases in total splenic T cell and dendritic

cell numbers persisted for at least four weeks post-treatment (**Supplementary Fig. 6G**). STARmap spatial transcriptomics of spleens collected immediately and four weeks after treatment confirmed stable enrichment of mature B cells and DCs within periarteriolar lymphoid sheaths. These findings indicate that, while thymic output is transient, the expanded peripheral compartments generated during treatment remain partially stabilized even in the absence of continued dosing. However, immunological challenges might deplete these compartments again, as outlined below.

Functional vaccine responses:

To determine whether these persistent cellular changes translate into equally persistent functional benefits, we delayed vaccination until four weeks after the last DFI dose (**Supplementary Fig. 6H**). At this point, we no longer observed enhanced antigen-specific CD8⁺ T cell expansion (**Supplementary Fig. 6I**), attenuated exhaustion (**Supplementary Fig. 6J**), or favorable shifts in naïve, effector, and memory T cell pools (**Supplementary Fig. 6K**). This suggests that the window for optimized vaccine or immunotherapy responses coincides with the active treatment period, when both thymic export and peripheral remodeling are maximized.

In summary, DFI induces a reversible thymic activation that drives *de novo* T cell output only during dosing, while concurrently reshaping splenic T, B, and dendritic cell compartments in a manner that persists for weeks. However, to achieve maximal functional benefits such as improved vaccine responses, DFI administration should be synchronized with antigen exposure, as the full breadth of immune enhancement does not persist beyond the treatment window. This transient and controllable profile may however represent a clinical advantage, as it allows for timed immunological boosting without prolonged exposure, thereby minimizing risks of chronic stimulation or autoimmunity.

7) Is the tumor survival due to a larger T cell repertoire? Or higher number of T cells? Hard to assess since the number of T cells in the model are equal. T cell diversity assessment would be helpful.

We thank the Reviewer for raising this key mechanistic question. As noted by the reviewer, in our B16-OVA model, absolute numbers of tumor-infiltrating CD8⁺ T cells are comparable between DFI-treated and control groups, which we believe suggests that improved tumor control cannot be attributed to increased cell numbers alone. Instead, multiple lines of evidence point toward enhanced T cell quality and repertoire diversity as the primary drivers of the effect.

Bulk TCR β repertoire sequencing of splenic T cells showed no change in overall clonality or diversity metrics in DFI-treated aged mice (**Fig. 2E**). While this indicates that DFI does not induce clonal skewing or collapse in the periphery, bulk sequencing is insensitive to low-abundance, newly generated clones, which may comprise only a minority of the total pool.

We therefore assessed T cell diversity and phenotypes specifically in the TIL compartment, as suggested by the Reviewer. Single-cell RNA/V(D)J analysis of tumor-infiltrating lymphocytes (TILs) revealed that DFI treatment indeed led to a more clonally diverse population of tumor-reactive CD8⁺ T cells (**Fig. 4V**). These TILs displayed a higher proportion of naïve-like (CD44⁻CD62L⁺) and progenitor-exhausted (PD-1⁺TCF1⁺) phenotypes (**Fig. 4T-U**), indicating recruitment of new clones and improved functional potential within the tumor microenvironment. Notably, tumor rejection in

both, MC38 and B16 tumor models was observed without an increase in the total number of CD8⁺ T cells, underscoring that T cell quality - reflected in repertoire diversity and differentiation state - rather than sheer abundance underlies the enhanced anti-tumor response.

8) In figure 5, OT mice are used where OVA is expressed as a self-antigen, it would be unexpected for these mice to have a break in tolerance unless the treatment disrupted central or peripheral tolerance mechanisms. Partial AIRE deficiency models or other models of tolerance and medullary thymic function should be used instead or some references to the use of breaking of tolerance in OT mice provided.

We fully appreciate the Reviewer's suggestion to use partial *AIRE*-deficient mice to more directly probe medullary thymic function. While we agree that such models could offer further mechanistic granularity, they were not commercially available and their generation was beyond the scope of the present study. Instead, we relied on well-established, validated models that are widely used to interrogate both central and peripheral tolerance.

The mAct-OVA or RIP-mOVA mouse models, in which OVA is expressed as a self-antigen under the β -actin or rat insulin promoters, respectively, are gold standards for assessing tolerance because OVA-reactive CD8⁺ and CD4⁺ T cells can be sensitively detected using validated MHC tetramers and anti-OVA antibody ELISAs. These assays allow us to detect even subtle breaches of tolerance. These models are widely used to study central and peripheral T cell tolerance and have been validated in numerous studies as stringent systems to detect self-reactivity or tolerance breaches (Zehn and Bevan, *Immunity* 2006; Parish et al., *PNAS* 2009, Koehli et al., *PNAS* 2014, Bosch et al., *Cellular Immunology* 2017, Enouz et al., *J Exp Med* 2012, Miyagawa et al., *Journal of Autoimmunity* 2010). We have now provided these references in the manuscript. We used T cells from OT-I and OT-II TCR-transgenic mice solely as positive controls to benchmark the sensitivity of our flow cytometry and ELISA assays. Likewise, OVA-vaccinated C57BL/6 mice served as positive controls for anti-OVA IgG ELISAs. In all cases, robust antigen-specific signals were readily detected, confirming that our assays were sufficiently sensitive to capture relevant immune responses. The absence of OVA-specific T or B cell responses in DFI-treated mAct-OVA mice, even following OVA protein vaccination and analyzed with these sensitive assays, therefore strongly indicates that central and peripheral tolerance remain intact.

To further address the spirit of the Reviewer's concern, we extended our safety assessment by an additional model. In the polyclonal setting of experimental autoimmune encephalitis (EAE), aged mice treated with DFI and challenged with MOG₃₅₋₅₅ peptide exhibited increased frequencies of MOG-specific CD4⁺ T cells in the spleen but did not develop worsened clinical EAE scores, CNS demyelination, or inflammatory infiltration (**Extended Data Fig. 10B-H**). In this context, viewed apart from the CNS manifestations, EAE induction essentially functions as a peptide vaccination model, and together with our OVA data, these findings show that DFI enhances peripheral antigen-specific priming without compromising the tissue-resident regulatory networks that preserve CNS immune privilege. In young adult mice, DFI neither enhances autoreactivity nor aggravated disease outcomes, demonstrating that DFI does not indiscriminately amplify immune activation in immunocompetent hosts.

Across these three complementary systems i.) mAct-OVA (central and peripheral tolerance), ii.) NOD (spontaneous autoimmunity), and iii.) MOG-induced EAE (polyclonal autoreactivity), we consistently find that DFI restores immune responsiveness in aged hosts without breaching central or peripheral tolerance mechanisms. Combined with our observation that thymic stromal composition, Notch ligand expression, and negative selection machinery, including intrathymic DCs and mTECs, remain stable (**Point-by-point Fig. 3E-H, 4A-B**), these data support DFI as a safe, temporally controlled immune-enhancing strategy, even in the context of self-antigen exposure.

Minor comments:

1) Is there off-target delivery? Is the treatment specific to the liver? The authors might have a sentence about that, the overemphasis on the liver is nice but isn't the delivery mainly going to the liver because it's the organ which is first and foremost targeted by the LNPs.

We thank the reviewer for raising this important point regarding the organ specificity of our mRNA-LNP delivery system. While we focus on the liver as a therapeutic site because of its protein-production capacity, immune-regulatory functions, and accessibility by blood-circulating immune cells, we agree that systemic LNPs preferentially accumulate in the liver primarily due to its anatomical and physiological role as a first-pass filter organ.

To directly assess whether our delivered mRNAs reach other tissues, we performed RIBOmap profiling, which demonstrated that *Dll1*, *Flt3L*, and *Il7* mRNAs are translated exclusively in hepatocytes, with no detectable off-target translation in other organs or cell types (**Fig. 1F-G**). In parallel, *in vivo* bioluminescence imaging following Luc-mRNA LNP administration confirmed strong and selective hepatic signal, with negligible luminescence in heart, thymus, spleen or kidneys (**Supplementary Fig. 1F-G**).

These findings show that while the hepatic tropism of our platform reflects the known biodistribution of systemically delivered LNPs, translation of our therapeutic mRNAs is confined to hepatocytes, with no evidence of off-target expression. We now explicitly state this in the revised manuscript to clarify that the observed liver specificity reflects both the natural pharmacology of LNPs and the absence of unintended tissue transduction (lines 123-127).

2) NASH in the liver is associated with Notch H&E staining's at least of normal liver tissue after LNP is preferable.

We thank the reviewer for raising the important issue of potential Notch-driven liver pathology, particularly in light of reports linking aberrant Notch activity to NASH. To address this, we conducted a comprehensive evaluation of hepatic safety following DFI treatment, which we summarized in **Extended Data Fig. 4:**

Body weight remained stable across all treatment groups throughout the 35-day study (**Extended Data Fig. 4G**). Serum liver function markers (AST, ALT, CK, GGT, and bilirubin) measured longitudinally during the 4-week dosing period showed no elevations or abnormalities (**Extended Data Fig. 4H**).

For histological assessment, we performed H&E and Masson's trichrome staining on liver sections from aged NaCl- and DFI-treated mice (**Extended Data Fig. 4I-J**). Slides were reviewed in a blinded fashion by board-certified veterinary pathologists (**Extended Data Fig. 4K-L**). Pathology scoring encompassed inflammation, hepatocellular necrosis, steatosis, fibrosis, and other features associated with NASH. As summarized in **Extended Data Fig. 4K-L**, no significant differences in cumulative or individual pathology scores were found between groups, including aged DFI-treated animals.

[Redacted text]

Furthermore, it is worth noting that DLL1 is not typically associated with the pro-fibrotic, cholangiopathic, or carcinogenic outcomes reported for DLL4 or Jagged family ligands in the liver (Shen et al., *Am J. Pathol* 2016; Nakano et al., *Commun Biol.* 2022), which we believe further reduces the likelihood of hepatotoxicity by DFI.

Together, we believe findings demonstrate that repeated DFI mRNA-LNP administration is well tolerated in aged mice, with no evidence of Notch-mediated liver injury. We have clarified these results in the revised text to emphasize that liver pathology was systematically assessed and found to be unremarkable, supporting the safety and translational feasibility of hepatic mRNA expression with LNPs (Beck et al., *Cancer Cell* 2024).

3) Why is there an increase in IL6 specifically? Is that due to Liver injury with the LNP?

We thank the reviewer for raising this question regarding the source of the observed IL-6 elevations. To clarify whether IL-6 induction reflects hepatotoxicity, the LNP carrier, or the mRNA cargo, we now performed extended cytokine profiling after four weeks of repeated dosing in aged mice, including Luc mRNA-LNPs as a control alongside each DFI component and the full combination (**Supplementary Fig. 11**).

Across all groups, IL-6 levels were not consistently or significantly elevated, including in animals receiving Luc mRNA-LNPs, indicating that neither the LNP formulation nor the DFI mRNAs induce sustained systemic IL-6. Mild and variable IL-6 increases observed also in Luc-treated animals suggest this effect is not specific to DFI, but rather reflects a low-level innate immune response to repeated exposure to synthetic mRNA, as has been described previously (Karikó et al., *Immunity* 2005; Kim et al., *Nat Commun* 2024).

Importantly, neither liver function parameters (**Extended Data Fig. 4H**) or comprehensive liver histopathology revealed evidence of hepatocellular injury, necrosis, fibrosis, or steatosis in any treatment group (**Extended Data Fig. 4I-L**), strongly arguing against liver damage as the cause of IL-6 elevation. Together, these results indicate that the slight and transient IL-6 signal is not a marker of hepatotoxicity or chronic inflammation, but rather a nonspecific, minor innate response occasionally triggered by synthetic mRNA, and is not sustained under our dosing conditions.

4) Figure 5B: There is no increase in circulating autoreactive T cell, what is the effect of the treatment of LNPs. Is the medullary Thymic epithelial cell function also ameliorated by the treatment?

We thank the reviewer for this important question regarding the impact of DFI on medullary thymic epithelial cell (mTEC) function and central tolerance. To address this, we performed spatial transcriptomics (Slide-seq v2) on whole thymic tissue from DFI- and Luc-treated aged mice, as TECs are highly sensitive to dissociation and therefore challenging to assess using conventional single-cell RNA-seq (**Point-by-point Fig. 3D-H**). These analyses revealed a modest increase in overall mTEC abundance following DFI treatment, consistent with partial restoration of thymic cellularity. However, we detected no significant upregulation of canonical transcriptional regulators or functional markers associated with medullary differentiation, including *Aire*, *Relb*, *Fezf2*, and *Nfkb2*. This indicates that while mTEC numbers may rise slightly, their transcriptional programming remains stable and does not reflect a functional remodeling of the medullary compartment (**Point-by-point Fig. 3E, H**).

To further assess tolerance-related elements, we analyzed intrathymic DCs, which collaborate with mTECs in negative selection. Flow cytometry showed that while their relative frequencies were higher in DFI-treated thymi, absolute DC numbers remained unchanged (**Point-by-point Fig. 4A**). Moreover, surface CD80 and MHC-II expression, both critical for antigen presentation and central tolerance, remained constant across age and treatment (**Point-by-point Fig. 4A**). Confirming the spatial transcriptomics data, we also did not observe any differences in RANK, CD80, or AIRE surface protein levels on mTECs (**Point-by-point Fig. 4B**), suggesting that mTEC function is not affected by the treatment.

Taken together, these findings suggest that DFI does not fundamentally reprogram the thymic stromal niche. Instead, it primarily enhances thymic output by expanding progenitor influx (e.g., CCR9⁺ CLPs) and early thymocyte development (e.g., DN1-3 thymocytes, increased TRECs and Rag2 induction), while leaving mTEC and DC function - and thus central tolerance - largely intact. This is consistent with the absence of increased tissue-specific autoreactive T cells in all three tested animal models (e.g., NOD, mAct-OVA, EAE) and supports the conclusion that DFI enhances immune function without compromising self-tolerance.

5) There is an extensive amount of computational data being presented in supplementary figures. This may be nice but the factors used have been widely studied, the authors should emphasize these previous publications, e.g. of IL7 in both mouse and men.

We thank the reviewer for this valuable suggestion. While our study presents a new combinatorial strategy and delivery context for DLL1, FLT3L, and IL-7, we fully agree that each of these factors, particularly IL-7 and FLT3L, has been extensively studied for their individual immunological roles and therapeutic potential. We have revised the manuscript to better acknowledge and cite foundational and translational work on these molecules in both murine models and human studies. Specifically, we now highlight the essential role of IL-7 in T cell development, homeostasis, and survival, particularly in supporting thymopoiesis, peripheral naïve T cell maintenance, and expansion

in lymphopenic settings, the clinical development of recombinant human IL-7 (rhIL-7) in cancer immunotherapy, HIV infection, and immune reconstitution after chemotherapy, radiotherapy, or bone marrow transplantation and the well-established function of FLT3L as a potent dendritic cell growth factor, particularly in expanding cDC1 and enhancing cross-priming of CD8⁺ T cells, as well as its use in clinical trials to enhance vaccine responses and checkpoint blockade efficacy.

We also clarify in the revised discussion that while the individual activity of these factors is well known, our study leverages their combinatorial and spatially restricted expression in hepatocytes to orchestrate a rejuvenated immune milieu in aged mice, a strategy that differs from systemic recombinant protein administration.

IL-7 references now included in the manuscript:

Fry TJ & Mackall CL. Interleukin-7: from bench to clinic. *Blood*. 2002;99(11):3892–3904. [PMID: 12036849]

Sportès C et al. Administration of rhIL-7 in humans increases in vivo TCR repertoire diversity by preferential expansion of naïve T cell subsets. *J Exp Med*. 2008;205(7):1701–1714. [PMID: 18573909]

Schluns KS, Kieper WC, Jameson SC, Lefrançois L. Interleukin-7 mediates the homeostasis of naïve and memory CD8 T cells in vivo. *Nat Immunol*. 2000 Nov;1(5):426–32. doi: 10.1038/80868. PMID: 11062503.

Sereti I, Dunham RM, Spritzler J, Aga E, Proschan MA, Medvik K, Battaglia CA, Landay AL, Pahwa S, Fischl MA, Asmuth DM, Tenorio AR, Altman JD, Fox L, Moir S, Malaspina A, Morre M, Buffet R, Silvestri G, Lederman MM; ACTG 5214 Study Team. IL-7 administration drives T cell-cycle entry and expansion in HIV-1 infection. *Blood*. 2009 Jun 18;113(25):6304–14. doi: 10.1182/blood-2008-10-186601. Epub 2009 Apr 20. PMID: 19380868; PMCID: PMC2710926.

Perales MA, Goldberg JD, Yuan J, Koehne G, Lechner L, Papadopoulos EB, Young JW, Jakubowski AA, Zaidi B, Gallardo H, Liu C, Rasalan T, Wolchok JD, Croughs T, Morre M, Devlin SM, van den Brink MR. Recombinant human interleukin-7 (CYT107) promotes T-cell recovery after allogeneic stem cell transplantation. *Blood*. 2012 Dec 6;120(24):4882–91. doi: 10.1182/blood-2012-06-437236. Epub 2012 Sep 25. PMID: 23012326; PMCID: PMC3520625.

FLT-3L references now included in the manuscript:

Waskow C et al. The receptor tyrosine kinase Flt3 is required for dendritic cell development in peripheral lymphoid tissues. *Nat Immunol*. 2008;9(6):676–683. [PMID: 18469817]

Maraskovsky E et al. In vivo generation of human dendritic cell subsets by Flt3 ligand. *Blood*. 2000;96(3):878–884. [PMID: 10910910]

Liu K et al. In vivo analysis of dendritic cell development and homeostasis. *Science*. 2009;324(5925):392–397. [PMID: 19286520]

Spranger S et al. Tumor-residing Batf3 dendritic cells are required for effector T cell trafficking and adoptive T cell therapy. *Cancer Cell*. 2017;31(5):711–723.e4. [PMID: 28486107]

[Redacted text and figure]

Point-by-Point Fig. 2 | Comparison of DFI versus LFI (Luc-Flt3l-II7) lipid nanoparticle (LNP) conditioning in a vaccination model.

(A) Experimental design: adult and aged C57BL/6J mice were conditioned with DFI, LFI, or control LNPs for 28 days, followed by immunization with an adjuvanted Ovalbumin (OVA) vaccine.

(B) Spleen weights at day 35 post-vaccination. n = 3 mice per group. Statistical analysis by one-way ANOVA with Tukey's multiple-comparison post hoc test.

(C) Thymus weights at day 35 post-vaccination. n = 3 mice per group. Statistical analysis as in (B).

(D) Relative frequencies of SIINFEKL-H-2K^b tetramer-binding CD8⁺ T cells in spleens. n = 5 mice per group. Data represent mean ± s.e.m. Statistical analysis by one-way ANOVA with Tukey's post hoc test.

(E) Heatmaps of cytokine concentrations (IFN-γ, TNF, IL-17) measured by ELISA following 72-hour *ex vivo* recall stimulation of splenocytes with full-length OVA protein. n = 5 mice per group.

(F) Serum anti-OVA IgG titers at day 35 post-vaccination. n = 10 mice per group. Statistical analysis by one-way ANOVA with Dunnett's post hoc test.

Point-by-Point Fig. 3 | Spatial transcriptomic profiling of thymic epithelial cell (TEC) subsets following DFI treatment.

(A) Experimental design: aged C57BL/6J mice were treated with DFI or Luc mRNA-LNPs for 28 days, followed by explantation of thymuses for histological evaluation, Slide-seq v2 spatial transcriptomics, and parallel flow cytometric profiling of stromal cell populations (see Point-by-Point Fig. 4).

(B) Representative H&E-stained thymic sections from adult, aged Luc-treated, and aged DFI-treated mice, demonstrating cortical and medullary architecture.

(C) Quantification of thymic cellularity ($\times 10^3$ cells/mm²) across experimental groups. n = 3 mice per group. Statistical significance determined by one-way ANOVA with Tukey's post hoc test.

(D) Spatial UMAP visualization of confidently annotated cells across adult, aged Luc-treated, and aged DFI-treated thymic sections by Slide-seq v2.

(E) Absolute counts of early TECs (left), cortical TECs (cTECs; middle), and medullary TECs (mTECs; right) as determined by transcriptional annotation using Slide-seq v2.

(F-H) Heatmaps displaying the relative expression of canonical lineage markers and Notch ligands in early TECs (F), cTECs (G), and mTECs (H) from adult, aged Luc-treated, and aged DFI-treated mice.

Point-by-Point Fig. 4 | Flow cytometric characterization of intrathymic dendritic cells (DCs) and medullary thymic epithelial cells (mTECs).

(A) Quantification of intrathymic DCs (CD45⁺ CD11c⁺ MHC-II⁺) across experimental conditions. From left to right: frequencies of total DCs, absolute DC counts normalized to thymus weight (per mg tissue), mean fluorescence intensity (MFI) of CD80 (B7-1), and MFI of MHC-II on DCs. n = 5 mice per group.

(B) Phenotypic assessment of medullary thymic epithelial cells (mTECs). From left to right: MFI of RANK, CD80 (B7-1), and AIRE expression. n = 4 mice per group.

For all panels, data are presented as mean ± s.e.m. Statistical significance was determined by one-way ANOVA followed by Tukey's multiple-comparison post hoc test.

[Redacted text and figure]

[Redacted text]

Point-by-Point Fig. 6 | Circulating monocyte subsets and intrathymic $\gamma\delta$ T cell frequencies following DFI treatment.

(A) Schematic representation of the hierarchical hematopoietic stem and progenitor cell (HSPC) lineage tree, illustrating the relative changes in subset frequencies in aged DFI-treated compared with aged Luc-treated bone marrow.

(B-D) Quantification of circulating monocyte populations in peripheral blood:

(B) total monocytes (CD45⁺ Lin⁻ [CD3⁻ CD19⁻ NK1.1⁻ Ly6G⁻]),

(C) classical monocytes (Ly6C^{high} CCR2^{high} CX3CR1^{low}), and

(D) non-classical monocytes (Ly6C^{low} CCR2^{low} CX3CR1^{high}).

(E) Frequency of intrathymic $\gamma\delta$ T cells (CD45⁺ CD4⁻ CD8⁻ TCR β ⁻ TCR $\gamma\delta$ ⁺) in explanted thymuses.

For (B-E), quantifications were performed on n = 30 mice total (n = 5 adult, n = 5 aged + Luc, n = 5 aged + DLL1, n = 5 aged + FLT3L, n = 5 aged + IL-7, n = 5 aged + DFI). Data represent mean \pm s.e.m. Statistical comparisons were conducted using one-way ANOVA followed by Dunnett's multiple-comparison post hoc test.

Response to Reviews for 2024-10-20824B

Transient hepatic reconstitution of trophic factors enhances immune function in aged mice

Friedrich *et al.*

We thank the Reviewers for their time and positive feedback. Below we expand on the remaining comments point-by-point.

Referee #1 (Remarks to the Author):

The authors have undertaken extensive additional analyses employing complementary models and orthogonal approaches that not only robustly validate their original conclusions but also meaningfully extend the scope of the initial findings. Collectively, the original and newly generated data provide valuable mechanistic insights that are likely to be of strong interest to Nature's readership and have important implications for the field of immune aging, including potential clinical applications. I was enthusiastic about the initial submission, and I believe the authors' comprehensive and well-reasoned rebuttal has further strengthened the clarity and impact of the work.

The statistical analyses appear to be consistent and appropriate throughout the manuscript and in line with the journal's requirements.

We thank the reviewer for their time and constructive comments.

Referee #2 (Remarks to the Author):

The authors have made a great effort to clarify the mechanisms by which DFI works and have added significant new experiments supporting the beneficial effect of the treatment in different contexts while discarding potential undesirable harmful effects of the treatment. The only concern I have is that in the title and throughout the text these three factors DFI are referred to as thymic factors. While it is true that they are expressed there, some of their effects are also extrathymic, specially for FLT3L which may lead to confusion. I would therefore suggest going back to the original version of the title and softening the statement in order to avoid any potential misunderstanding.

We thank the reviewer for their time and constructive comments. We have duly considered their concern and have now consistently referred to the identified factors and the DFI treatment combination as "trophic" or "immune" factors throughout the manuscript. This approach was adopted as we were unable to categorize them meaningfully otherwise, given that DLL1 is not a cytokine, and both IL7 and FLT3L possess distinct extrathymic origins and effects, as accurately noted by the reviewer. Consequently, we have revised the manuscript's title in accordance while aligning with editorial guidelines.

Referee #3 (Remarks to the Author):

This is a strong, well-executed study with clear novelty and solid methodological rigor. The authors' revisions and responses were adequate with a considerable amount of additional data. I have no outstanding substantive concerns. I recommend acceptance of this manuscript.

We thank the reviewer for their time and constructive comments.